# Optimal Multi-Fidelity Best-Arm Identification

**Riccardo Poiani**[*]
DEIB, Politecnico di Milano, Milan, Italy
`riccardo.poiani@polimi.it`

**Rémy Degenne**
Univ. Lille, Inria, CNRS, Centrale Lille, UMR 9189-CRIStAL, F-59000 Lille, France
`remy.degenne@inria.fr`

**Emilie Kaufmann**
Univ. Lille, CNRS, Inria, Centrale Lille, UMR 9189-CRIStAL, F-59000 Lille, France
`emilie.kaufmann@univ-lille.fr`

**Alberto Maria Metelli**
DEIB, Politecnico di Milano, Milan, Italy
`albertomaria.metelli@polimi.it`

**Marcello Restelli**
DEIB, Politecnico di Milano, Milan, Italy
`marcello.restelli@polimi.it`

## Abstract

In bandit best-arm identification, an algorithm is tasked with finding the arm with highest mean reward with a specified accuracy as fast as possible. We study multi-fidelity best-arm identification, in which the algorithm can choose to sample an arm at a lower fidelity (less accurate mean estimate) for a lower cost. Several methods have been proposed for tackling this problem, but their optimality remain elusive, notably due to loose lower bounds on the total cost needed to identify the best arm. Our first contribution is a tight, instance-dependent lower bound on the cost complexity. The study of the optimization problem featured in the lower bound provides new insights to devise computationally efficient algorithms, and leads us to propose a gradient-based approach with asymptotically optimal cost complexity. We demonstrate the benefits of the new algorithm compared to existing methods in experiments. Our theoretical and empirical findings also shed light on an intriguing concept of optimal fidelity for each arm.

## 1  Introduction

In multi-armed bandits [20], an algorithm chooses at each step one *arm* among $K > 1$ possibilities. It then observes a reward, sampled from a probability distribution on $\mathbb{R}$ corresponding to the arm. Several goals are possible for the algorithm, and we focus on the *best arm identification* task (BAI) in which we aim to identify the arm with the largest mean, using as few samples as possible. This is a well-studied problem [6, 1, 12, 10, 8] with potential applications to, e.g. A/B/n testing [27] or hyper-parameter optimization [11].

In some applications, like physics, parameter studies, or hyper-parameter optimization, getting a sample from the arm distribution might be expensive since it requires evaluating or training a complex model and is computationally demanding. However, it is often the case that cheaper, less accurate sampling methods are available, for instance, by using a coarser model in the physics study example.

---

[*]Work done while at Inria, Lille, France.

38th Conference on Neural Information Processing Systems (NeurIPS 2024).

The *multi-fidelity* bandit framework takes such scenarios into account. When choosing an arm, the algorithm also chooses a fidelity, with a trade-off: a higher fidelity gives a more precise observation but has a higher cost. We assume that the algorithm knows both the cost and the maximal bias of the observations from each fidelity. This is also how the knowledge about the fidelity was modeled in prior work [see, e.g., 16, 15, 25, 31]. The goal is then to find the best arm (i.e., the arm with the highest mean at the highest fidelity) with high probability and minimal cost.

Specifically, the bandit algorithm interacts with the multi-fidelity environment and gathers information to find which arm has the highest mean when pulled at the highest fidelity. In the fixed confidence setting, we want to ensure that the algorithm returns a correct answer with probably at least $1 - \delta$ for a given parameter $\delta \in (0, 1)$. A good algorithm should do that at a minimum cost, and thus, the appropriate quality metric for evaluating an algorithm's performance is the sum of costs paid until it stops, i.e., the cost complexity. Previous work on the multi-fidelity BAI problem [25, 31] provided lower bounds on the cost complexity as well as algorithms with cost upper bounds. Those lower and upper bounds do not match, and the proposed methods require additional prior information [31], or their guarantees are restricted to problems satisfying additional hypotheses [25]. We lift all those requirements and provide an improved lower bound and an algorithm with a matching upper bound.

**Contributions and organization of the paper**   After presenting additional related works, in Section 2, we define fixed-confidence best arm identification in multi-fidelity bandits in more mathematical detail and introduce the notations used throughout the paper. Then, Section 3 contains our first contribution: a tight instance-dependent lower bound on the cost complexity of any algorithm expressed with the maximum of a complex function over all possible *cost* allocations. We also highlight features of that lower bound, like the existence of an optimal fidelity for each arm, which should be chosen exclusively. In Section 4, we propose a computationally efficient procedure for computing gradients of the function featured in the lower bound and describe a gradient-based algorithm whose cost complexity is asymptotically matching the lower bound. Finally, in Section 5, we present the results of numerical experiments which demonstrate the good empirical performance of our new algorithm compared to prior work.

**Additional related works**   The multi-fidelity setting has mostly been studied in the context of Bayesian optimization [9, 24, 17, 26, 14, 21] and black-box function optimization with different structural assumptions [28, 29, 7, 23]. The goal there is to find the minimum of a function by successive queries of that function or of cheaper approximations. The metric for success in these works is most often the simple regret, that is, the difference between the best value found and the true minimum, although other goals were considered like the cumulative regret [16, 15]. Furthermore, we notice that best arm identification with costs has recently been studied in [13] for BAI with only one fidelity. The authors introduce a variant of the Track-and-Stop algorithm [8] and prove its asymptotic optimality. However, we will not be able to adapt this study to the multi-fidelity case because, as we shall see, it requires solving a complex optimization problem for which we have no efficient solution. Finally, our work is related to the vast strand of BAI studies that proposes tight lower bound with asymptotically optimal algorithms [e.g., 8, 4, 22]; nevertheless, as we discuss throughout the text, these studies cannot be directly applied to the multi-fidelity BAI problem.

## 2   Background

In this section, we provide essential background and notation that is used throughout the rest of the paper. A table that summarizes the notation is available in Appendix A.

A multi-fidelity bandit model with $K$ arms and $M$ fidelities is a set of $K \times M$ probability distributions $\boldsymbol{\nu} = (\nu_{a,m})_{a \in [K], m \in [M]}$ where $\nu_{a,m}$ has mean of $\mu_{a,m}$. For each arm $a \in [K]$, $\mu_{a,m}$ represents the mean value of an observation of arm $a$ using fidelity $m$, and let $\boldsymbol{\mu} = (\mu_{a,m})_{a \in [K], m \in [M]}$. An observation at fidelity $m$ is assigned a (known) cost $\lambda_m \geq 0$ with $\lambda_1 < \lambda_2 < \cdots < \lambda_M$. The goal is to identify the arm that has the largest mean at the highest fidelity $M$, $a_\star(\boldsymbol{\mu}) := \operatorname{argmax}_{a \in [K]} \mu_{a,M}$ (sometimes denoted by $\star$ in the sequel to ease notation) with a small total sampling cost, by exploring the arms at different fidelities and using some prior knowledge about their precision. Specifically, we assume that there are some (known) values $\xi_1 > \xi_2 > \cdots > \xi_M = 0$ such that, for all arm $a \in [K]$, the vector $\mu_a := (\mu_{a,m})_{m \in [M]}$ satisfies

$$\forall m \in [M], \quad |\mu_{a,m} - \mu_{a,M}| \leq \xi_m .$$

We write $\mu_a \in \text{MF}$ to indicate that arm $a$ satisfies these multi-fidelity constraints, with these particular parameters $\xi_m$ (although they are not shown in the notation). In this paper, we consider arms that belong to a canonical exponential family [2]. This includes, e.g. arms that have Bernoulli distributions or Gaussian distributions with known variances. Such models are known to be characterized by their means and we refer to such an exponential multi-fidelity bandit model $\boldsymbol{\nu}$ using the means of its arms $\boldsymbol{\mu}$, which belongs to the set $\mathcal{M}_{\text{MF}} := \{\boldsymbol{\mu} \in \Theta^{K \times M} : \forall a \in [K], \mu_a \in \text{MF}\}$, where $\Theta \subseteq \mathbb{R}$ is the interval of possible means.

At each interaction round $t = 1, 2, \ldots$, the agent selects an arm $A_t$ and a fidelity $M_t$, observes a sample $X_t \sim \nu_{A_t, M_t}$ and pays a cost $\lambda_{M_t}$. Letting $\mathcal{F}_t = \sigma(A_1, M_1, X_1, \ldots, A_t, M_t, X_t)$ be the sigma field generated by the observations up to time $t$, a fixed-confidence identification algorithm takes as input a risk parameter $\delta \in (0, 1)$ and is defined by the following ingredients: (i) a *sampling rule* $(A_t, M_t)_t$, where $(A_t, M_t)$ is $\mathcal{F}_{t-1}$-measurable, (ii) a *stopping rule* $\tau_\delta$, which is a stopping time w.r.t. $\mathcal{F}_t$, and (iii) a *decision rule* $\hat{a}_{\tau_\delta} \in [K]$, which is $\mathcal{F}_{\tau_\delta}$-measurable. We want to build strategies that ensure $\mathbb{P}_{\boldsymbol{\mu}}(\hat{a}_{\tau_\delta} \neq a_\star(\boldsymbol{\mu})) \leq \delta$ for all $\boldsymbol{\mu} \in \mathcal{M}_{\text{MF}}$ with a unique optimal arm. Such a strategy is called $\delta$-*correct*. Among $\delta$-correct strategies, we are looking for strategies that minimize the expected identification cost (i.e., *cost complexity*) defined as

$$\mathbb{E}_{\boldsymbol{\mu}}[c_{\tau_\delta}] := \sum_{a \in [K]} \sum_{m \in [M]} \lambda_m \mathbb{E}_{\boldsymbol{\mu}}[N_{a,m}(\tau_\delta)] = \sum_{a \in [K]} \sum_{m \in [M]} \mathbb{E}_{\boldsymbol{\mu}}[C_{a,m}(\tau_\delta)],$$

where $N_{a,m}(t)$ denotes the number of pulls of arm $a$ at fidelity $m$ up to time $t$ and $C_{a,m}(t) = \lambda_m N_{a,m}(t)$ denotes the cost associated to these pulls. In the sequel, we will provide cost complexity guarantees for multi-fidelity instances $\boldsymbol{\mu}$ that belong to the set $\mathcal{M}_{\text{MF}}^*$ of multi-fidelity instances with a unique optimal arm, i.e., for which $|a_\star(\boldsymbol{\mu})| = 1$. We remark that for $M = 1$ and $\lambda_m = 1$ we recover the best arm identification problem in a classical bandit model, for which the cost complexity coincides with the sample complexity, $\mathbb{E}_{\boldsymbol{\mu}}[\tau_\delta]$.

**Additional notation**   Given an integer $n \in \mathbb{N}$, we denote by $\Delta_n$ the $n$-dimensional simplex. Furthermore, given $x, y \in (0, 1)$, we define $\text{kl}(x, y) = x \log(x/y) + (1-x) \log((1-x)/(1-y))$. Given $(p, q) \in \Theta^2$, we denote by $d(p, q)$ the Kullback-Leibler (KL) divergence between the distribution in the exponential family with mean $p$ and that with mean $q$. We also write $d^-(x, y) = d(x, y)\mathbf{1}\{x \geq y\}$ and $d^+(x, y) = d(x, y)\mathbf{1}\{x \leq y\}$. Finally, we denote by $v(p)$ the variance of the distribution with mean $p$.

## 3   On the cost complexity of multi-fidelity best-arm identification

In this section, we discuss the statistical complexity of identifying the best-arm in MF-BAI problems. Formal proofs of the claims of this section are presented in Appendix B.

### 3.1   Lower bound on the cost complexity

We present an instance-dependent lower bound on the expected cost-complexity. The lower bound uses the solution to an optimization problem, where the functions optimized quantify the trade-off between the information gained by pulling an arm at some fidelity and the cost of that fidelity. Since those functions also appear in our algorithm, we will now introduce notation for them. For all $\boldsymbol{\omega} \in \Delta_{K \times M}$ and $\boldsymbol{\mu} \in \Theta^{K \times M}$, we define

$$f_{i,j}(\boldsymbol{\omega}, \boldsymbol{\mu}) := \inf_{\substack{\theta_i \in \text{MF}, \, \theta_j \in \text{MF} \\ \theta_{j,M} \geq \theta_{i,M}}} \sum_{a \in \{i,j\}} \sum_{m \in [M]} \omega_{a,m} \frac{d(\mu_{a,m}, \theta_{a,m})}{\lambda_m} \, , \tag{1}$$

$$F(\boldsymbol{\omega}, \boldsymbol{\mu}) := \max_{i \in [K]} \min_{j \neq i} f_{i,j}(\boldsymbol{\omega}, \boldsymbol{\mu}) \, . \tag{2}$$

The quantity $f_{i,j}(\boldsymbol{\omega}, \boldsymbol{\mu})$ is the dissimilarity according to a KL weighted by the costs between $\boldsymbol{\mu}$ and the closest $\boldsymbol{\theta} \in \Theta^{K \times M}$ such that arms $i$ and $j$ satisfy the multi-fidelity constraints and $\theta_k = \mu_k$ for $k \notin \{i, j\}$, with arm $j$ better than arm $i$. If $\boldsymbol{\mu} \in \mathcal{M}_{\text{MF}}$ then that closest $\boldsymbol{\theta}$ is also in $\mathcal{M}_{\text{MF}}$ but otherwise it might not be the case: if an arm $k \notin \{i, j\}$ is not in MF for $\boldsymbol{\mu}$, then it is equally not in MF for $\boldsymbol{\theta}$. For $\boldsymbol{\mu} \in \mathcal{M}_{\text{MF}}$ the maximum in the definition of $F$ is realized at the best arm $\star$, as $\min_{a \neq i} f_{i,a}(\boldsymbol{\omega}, \boldsymbol{\mu})$ is zero for $i \neq \star$. That is, $F(\boldsymbol{\omega}, \boldsymbol{\mu}) = \min_{j \neq \star} f_{\star,j}(\boldsymbol{\omega}, \boldsymbol{\mu})$. We define $F$ with a maximum over $i$ and

not with that last expression because we want to define it for all points in $\Theta^{K \times M}$, even the points which are not in $\mathcal{M}_{\text{MF}}$. For those points, we could imagine different notions of best arm, for example, $\arg\max_k \mu_{k,M}$, but the right one for our algorithm is the arm for which we have the most evidence (weighted by cost) to say that all other arms are not better. That arm is the argmax in our definition of $F$. Given these definitions, we now introduce our new lower bound.

**Theorem 3.1.** *Let $\delta \in (0, 1)$. For any $\delta$-correct strategy, and any multi-fidelity bandit model $\boldsymbol{\mu} \in \mathcal{M}^*_{\text{MF}}$, it holds that:*

$$\mathbb{E}_{\boldsymbol{\mu}}[c_{\tau_\delta}] \geq C^*(\boldsymbol{\mu}) \log\left(\tfrac{1}{2.4\,\delta}\right), \tag{3}$$

*where $C^*(\boldsymbol{\mu})^{-1} := \sup_{\boldsymbol{\omega} \in \Delta_{K \times M}} F(\boldsymbol{\omega}, \boldsymbol{\mu}) = \sup_{\boldsymbol{\omega} \in \Delta_{K \times M}} \min_{a \neq \star} f_{\star,a}(\boldsymbol{\omega}, \boldsymbol{\mu})$.*

The quantity $C^*(\boldsymbol{\mu})$ describes the statistical complexity of an MF problem $\boldsymbol{\mu}$ as the typical max-min game that appears in lower bounds for BAI problems [see, e.g., 8, 3]. Specifically, first, the max-player chooses a vector $\boldsymbol{\omega} \in \Delta_{K \times M}$, and then the min-player chooses a bandit model $\boldsymbol{\theta} \in \mathcal{M}_{\text{MF}}$ in which the optimal arm is different, with the goal of minimizing the function $F(\boldsymbol{\omega}, \boldsymbol{\mu})$. Following the methods from previous work, the objective value for $\boldsymbol{\omega}$ and $\boldsymbol{\theta}$ should be $\sum_{a \in [K]} \sum_{m \in [M]} \omega_{a,m} \frac{d(\mu_{a,m}, \theta_{a,m})}{\lambda_m}$, featuring a sum over all arms and fidelities. However in the definition of $f_{i,a}(\boldsymbol{\omega}, \boldsymbol{\mu})$ we restrict $\boldsymbol{\theta}$ to be different from $\boldsymbol{\mu}$ on only two arms. We can prove that if $\boldsymbol{\mu} \in \mathcal{M}_{\text{MF}}$, this gives the same objective value at the minimizing $\boldsymbol{\theta}$ as the full sum. The difference will be important in our algorithm, which will compute that minimizer for points $\hat{\boldsymbol{\mu}}$ that do not belong to $\mathcal{M}_{\text{MF}}$.

A difference with standard BAI settings is that in Equation (1) each $\boldsymbol{\omega} \in \Delta_{K \times M}$ should be interpreted as a vector of *cost proportions* that the max-player is investing (in expectation) in each arm-fidelity pair to identify the optimal arm $\mu_{\star,M}$.[2] We can interpret the *oracle weights* $\boldsymbol{\omega}^* \in \arg\max_{\Delta_{K \times M}} F(\boldsymbol{\omega}, \boldsymbol{\mu})$ as the optimal cost proportions that the agent should follow in order to identify $\mu_{\star,M}$ while minimizing the identification cost. To clarify the difference and the relationship between cost and pull proportions we notice that, given a cost proportion $\boldsymbol{\omega}$, it is always possible to compute the pull proportions $\boldsymbol{\pi}(\boldsymbol{\omega}) \in \Delta_{K \times M}$ that the agent should play in order to incur the costs proportions specified by $\boldsymbol{\omega}$, and vice versa. More specifically, these relationships are described for each arm-fidelity pair by the following equations for every $a \in [K]$ and $m \in [M]$:

$$\pi_{a,m}(\boldsymbol{\omega}) = \frac{\omega_{a,m}}{\lambda_m} \frac{1}{\sum_{i \in [K]} \sum_{j \in [M]} \frac{\omega_{i,j}}{\lambda_j}} \qquad \omega_{a,m}(\boldsymbol{\pi}) = \frac{\lambda_m \pi_{a,m}}{\sum_{i \in [K]} \sum_{j \in [M]} \lambda_j \pi_{i,j}}. \tag{4}$$

As a direct consequence, it is possible to rewrite $C^*(\boldsymbol{\mu})^{-1}$ as a function of $\boldsymbol{\pi}$, the pull proportions. Doing so reveals that the minimizer $\boldsymbol{\theta}$ in $f_{\star,j}$ does not depend on the costs: it is also the minimizer of $\sum_{a \in \{i,j\}, m \in [M]} \pi_{a,m} d(\mu_{a,m}, \theta_{a,m})$. While the agent optimizes the cost proportions $\boldsymbol{\omega}$ to get the best possible information/cost ratio, the min-player minimizes only the information available to the algorithm to tell $\boldsymbol{\mu}$ and $\boldsymbol{\theta}$ apart. Finally, we notice that $F(\boldsymbol{\omega}, \boldsymbol{\mu})$ is concave in $\boldsymbol{\omega}$[3] but $F(\boldsymbol{\omega}(\boldsymbol{\pi}), \boldsymbol{\mu})$ is not concave in $\boldsymbol{\pi}$. As we shall see in Section 4, this difference will play a crucial role in constructing an asymptotically optimal algorithm.

The formulation of the lower bound as a game where one player maximizes an information/cost ratio while the other player minimizes information makes our result close to lower bounds for regret minimization like the one of [5], where the (unknown) gap of an arm plays the role of the cost.

**Comparison to previous work**  The only known lower bound for the multi-fidelity BAI problem is the one presented in [25]. That same bound was then shown in [31]. The bound from those previous works is looser than Theorem 3.1. For example, in a two-arms bandit with a single fidelity (denoted by $M$) and Gaussian rewards with variance 1, the bound from previous work is $\lambda_M(\mu_{1,M} - \mu_{2,M})^{-2} \log(1/2.4\delta)$, while our lower bound is $8\lambda_M(\mu_{1,M} - \mu_{2,M})^{-2} \log(1/2.4\delta)$. Furthermore, on particular instances with 2 arms and 2 fidelity, we can prove that our lower bound improves by a factor $\lambda_M/\lambda_1$, which can be arbitrarily large (See Appendix B.2). More generally, the proof of the previous lower bounds exhibits a particular point in the alternative, which makes it always looser than our bound which features an infimum over all points. Theorem 3.1 is also optimal in the regime $\delta \to 0$ since it is matched by the algorithm we introduce in the next section.

---

[2]This claim is evident when looking at the proof of Theorem 3.1.

[3]This is a consequence of the fact $F(\boldsymbol{\omega}, \boldsymbol{\mu})$ is an infimum over linear functions of $\boldsymbol{\omega}$.

## 3.2 Sparsity of the oracle weights: a tight concept of optimal fidelity

We conclude our study of the lower bound by further analyzing the optimal allocation $\boldsymbol{\omega}^*$. Unlike in the standard best arm identification problem, we did not find an efficient algorithm to compute it, which prevents us from using a Track-and-Stop-like approach [8]. Nevertheless, we will explain in the next section how to efficiently compute the $f_{i,j}$ functions and their gradient. These computations are crucial for our algorithm but also allow us to prove our next result about the possible sparsity of $\boldsymbol{\omega}^*$. For each arm $a \in [K]$, it is not difficult to show that there must exist some fidelity $m \in [M]$ for which $\omega^*_{a,m} > 0$ (Lemma B.2). However, as the following result highlights, in most cases, only one fidelity per arm has non-zero weight.

**Theorem 3.2.** *Let* $\Delta^*_{K \times M}(\boldsymbol{\mu}) := \operatorname{argmax}_{\boldsymbol{\omega} \in \Delta_{K \times M}} F(\boldsymbol{\omega}, \boldsymbol{\mu})$ *and*

$$\widetilde{\mathcal{M}}_{\mathrm{MF}} := \left\{ \boldsymbol{\mu} \in \mathcal{M}^*_{\mathrm{MF}} : \exists i \in [K], \exists m_1, m_2 \in [M]^2, \exists \boldsymbol{\omega}^* \in \Delta^*_{K \times M}(\boldsymbol{\mu}) : \omega^*_{i,m_1} > 0, \omega^*_{i,m_2} > 0 \right\}.$$

*The set* $\widetilde{\mathcal{M}}_{\mathrm{MF}}$ *is a subset of* $\mathbb{R}^{K \times M}$ *whose Lebesgue measure is zero.*

Theorem 3.2 implies that in almost all multi-fidelity bandits, for any $\boldsymbol{\omega}^* \in \Delta^*_{K \times M}(\boldsymbol{\mu})$ and each arm $a \in [K]$, there exists a single fidelity $m^*_a \in [M]$ for which $\omega^*_{a,m^*_a} > 0$ holds. However, we note that this result does not offer an easy way to compute these optimal arm-dependent fidelities.[4] Nevertheless, as we shall see in the next section, our algorithm does not actually require identifying these optimal fidelity levels to enjoy optimality guarantees.

Finally, we remark that existing MF-BAI works [25, 31] already proposed notions of optimal, arm-dependent fidelity that the agent should employ to identify the optimal arm $\star$. Nevertheless, as we verify in Appendix B.5, these concepts do not comply with the concept of optimal fidelity that arises from the tight lower bound of Theorem 3.1. In other words, there exist bandit models $\boldsymbol{\mu}$ in which following these alternative concepts of optimal fidelity leads to sub-optimal performance.

## 4 The multi-fidelity sub-gradient ascent algorithm

We present our solution for solving MF-BAI problems, an algorithm called Multi-Fidelity Sub-Gradient Ascent (MF-GRAD). Its pseudocode can be found in Algorithm 1. All proofs for this section are presented in Appendix C.

A reader familiar with the literature on BAI algorithms inspired from lower bounds like Theorem 3.1 may have the natural idea of simply using the Track-and-Stop algorithm [8] or the related game-based algorithm of [4]. Those algorithms can't be directly applied here, first because of the costs: we want to bound the cost complexity, not the stopping time, and adapting those methods to costs is not trivial. Furthermore, Track-and-Stop (even in the cost-aware variant of [13]) would require the computation of the optimal cost proportions at $\hat{\boldsymbol{\mu}}(t)$, which is a max-min problem for which we don't have an efficient algorithm. Our solution is inspired by the gradient ascent algorithm of [22], which requires computing gradients of $F$ (hence only a minimization problem and not a max-min). The same innovations required to extend this method to the multi-fidelity case could likely allow us to adapt the algorithm of [4], or the exploration part of the regret-minimizing algorithm of [5].

Let us introduce some auxiliary notation. Let $\overline{\boldsymbol{\omega}} \in \Delta_{K \times M}$ be the uniform vector $\left( \frac{1}{KM}, \ldots, \frac{1}{KM} \right)$. For all $t \in \mathbb{N}$, we define $\mathrm{Clip}_t(x) = \left( \min\{x_{a,m}, G\sqrt{t}\} \right)_{a \in [K], m \in [M]}$ for an arbitrary constant $G > 0$. We also define $\alpha_t = \frac{1}{\sqrt{t}}$ and $\gamma_t = \frac{1}{4\sqrt{t}}$. Finally, for all $t \in \mathbb{N}$, we denote by $\boldsymbol{C}(t) \in \mathbb{R}^{KM}$ the vector whose $(a,m)$-th dimension is given by $C_{a,m}(t)$. We now present Algorithm 1.

**Sampling rule** After a first initialization phase in which the algorithm pulls each arm at each fidelity once (Line 1), the agent starts its sub-gradient ascent routine. More specifically at each iteration $t \in \mathbb{N}$, the agent first computes the vector $\tilde{\boldsymbol{\omega}}(t+1)$ using the Exponential Weights algorithm on the sequence of gain functions $\{g_s\}^t_{s=1} := \{\mathrm{Clip}_s \left( \left( \sum_{a,m} \lambda_m \tilde{\pi}_{a,m}(s) \right) \nabla F(\tilde{\boldsymbol{\omega}}(s), \hat{\boldsymbol{\mu}}(s)) \right)\}^t_{s=1}$, where $\tilde{\boldsymbol{\pi}}(t) := \boldsymbol{\pi}(\tilde{\boldsymbol{\omega}}(t))$ represents the pull-proportions induced by $\tilde{\boldsymbol{\omega}}(t)$ and $\nabla F(\tilde{\boldsymbol{\omega}}(s), \hat{\boldsymbol{\mu}}(s))$ denotes a sub-gradient of $F(\boldsymbol{\omega}, \boldsymbol{\mu})$ w.r.t $\boldsymbol{\omega}$ (Line 3). Neglecting for a moment the clipping function and the term

---

[4]We provide insights on cases in which it is possible to compute the optimal fidelity in Appendix B.4.

---

**Algorithm 1** Multi-Fidelity Sub-Gradient Ascent

---

1: **Initialization.** Pull each arm at each fidelity once, and set $\tilde{\boldsymbol{\omega}}(t) = \bar{\boldsymbol{\omega}}$ for all $t \in \{1, \ldots, KM\}$
2: **Sampling Rule** for $t \geq KM$
3: Sub-gradient Ascent

$$\tilde{\boldsymbol{\omega}}(t+1) \in \operatorname*{argmax}_{\boldsymbol{\omega} \in \Delta_{K \times M}} \alpha_{t+1} \sum_{s=KM}^{t} \boldsymbol{\omega} \cdot \mathrm{Clip}_s \left( \left( \sum_{a,m} \lambda_m \tilde{\pi}_{a,m}(s) \right) \nabla F(\tilde{\boldsymbol{\omega}}(s), \hat{\boldsymbol{\mu}}(s)) \right) - \mathrm{kl}(\boldsymbol{\omega}, \bar{\boldsymbol{\omega}})$$

4: From Costs to Pulls

$$\tilde{\pi}_{a,m}(t+1) = \frac{\tilde{\omega}_{a,m}(t+1)}{\lambda_m} \frac{1}{\sum_{i \in [K]} \sum_{j \in [M]} \frac{\tilde{\omega}_{i,j}(t+1)}{\lambda_j}} \quad \forall a \in [K], m \in [M]$$

5: Forced Exploration $\quad \boldsymbol{\pi}'(t+1) = (1 - \gamma_t)\tilde{\boldsymbol{\pi}}(t+1) + \gamma_t \bar{\boldsymbol{\omega}}$
6: Cumulative Tracking $\quad (A_{t+1}, M_{t+1}) \in \operatorname{argmax}_{(a,m) \in [K] \times [M]} \sum_{s=1}^{t} \pi'_{a,m}(s) - N_{a,m}(t)$
7: **Stopping Rule** $\quad \tau_\delta = \inf \left\{ t \geq KM : \max_{i \in [K]} \min_{j \neq i} f_{i,j}(\boldsymbol{C}(t), \hat{\boldsymbol{\mu}}(t)) \geq \beta_{t,\delta} \right\}$
8: **Decision Rule** $\quad \hat{a}_{\tau_\delta} \in \operatorname{argmax}_{i \in [K]} \min_{j \neq i} f_{i,j}(\boldsymbol{C}(t), \hat{\boldsymbol{\mu}}(t))$

---

$\tilde{c}(s) := \left( \sum_{a,m} \lambda_m \tilde{\pi}_{a,m}(s) \right)$ (these terms are present mainly for technical reasons), this step can be interpreted, from an intuitive perspective, as finding a sequence of weights $\{\tilde{\boldsymbol{\omega}}(t)\}_t$ that minimizes the regret on the sequence of empirical losses $F(\boldsymbol{\omega}^*, \hat{\boldsymbol{\mu}}(s)) - F(\tilde{\boldsymbol{\omega}}(s), \hat{\boldsymbol{\mu}}(s))$. [5] At this point, once $\tilde{\boldsymbol{\omega}}(t+1)$ is computed, Algorithm 1 will convert these cost proportions into pull proportions while adding some forced exploration (Line 4-5), and then, it applies a standard cumulative tracking procedure [8] in the pull-proportion space so to ensure that $N_{a,m}(t) \approx \sum_{s=1}^{t} \pi'_{a,m}(s)$ (Line 6).

**Stopping and decision rule** Finally, the algorithm applies a generalized likelihood ratio (GLR) test to decide when to stop (Line 7). For $i, j \in [K]$, $f_{i,j}(\boldsymbol{C}(t), \hat{\boldsymbol{\mu}}(t))$ can be interpreted a GLR statistics for comparing two classes: $\Theta^{KM}$ versus $\{\boldsymbol{\theta} \mid \theta_i \in \mathrm{MF}, \theta_j \in \mathrm{MF}, \theta_{j,M} \geq \theta_{i,M}\}$. If that GLR is large enough (if it exceeds a threshold $\beta_{t,\delta}$), we can reject the hypothesis that $\boldsymbol{\mu}$ belongs to the second class. If there is an arm $i$ for which we can reject the alternative class for all $j \neq i$, we have rejected all $\boldsymbol{\theta} \in \mathcal{M}_{\mathrm{MF}}$ where $i$ is not the best arm and we can safely stop and return the answer $\hat{a}_{\tau_\delta} = i$. Since each $f_{i,j}$ is expressed as a sum of only two arms and $M$ fidelities, it is possible to show that choosing $\beta_{t,\delta} \approx \log(K/\delta) + 2M \log(\log(t) + 1)$ (see its exact expression in (31)) guarantees the correctness of the test, namely that $\mathbb{P}_{\boldsymbol{\mu}}(\hat{a}_{\tau_\delta} \neq \star) \leq \delta$ holds (Proposition C.13).

## 4.1 Theoretical guarantees

At this point, we are ready to state the main theoretical result on the performance of our algorithm.

**Theorem 4.1.** *For any multi-fidelity bandit model* $\boldsymbol{\mu} \in \mathcal{M}_{\mathrm{MF}}$*, Algorithm 1 using the threshold* $\beta_{t,\delta}$ *given in* (31) *is* $\delta$*-correct and satisfies*

$$\limsup_{\delta \to 0} \frac{\mathbb{E}_{\boldsymbol{\mu}}[c_{\tau_\delta}]}{\log(1/\delta)} \leq C^*(\boldsymbol{\mu}). \tag{5}$$

As we can see from Theorem 4.1, Algorithm 1 is asymptotically optimal, meaning that it matches the lower bound we presented in Theorem 3.1 for the asymptotic regime of $\delta \to 0$.

**Comparison with existing MF-BAI algorithms** We conclude this section by comparing our results with the literature [25, 31]. First, [25] and [31] rely on *additional assumptions* that play a crucial role both for the algorithm design and the resulting theoretical guarantees. In [25], the authors enforce an additional and intricate structural assumption on the relationships between $\lambda$'s and $\xi$'s (see Assumption 1 in [25]). In [31], instead, the authors assume additional knowledge expressed as

---

[5]Whenever $\hat{\boldsymbol{\mu}}(s)$ is sufficiently close to $\boldsymbol{\mu}$, this implicitly generates a sequence of weights that provide values of $F(\cdot, \boldsymbol{\mu})$ "close" to the one of the oracle weights $\boldsymbol{\omega}^*$.

an upper bound on $\mu_{\star,M}$ and a lower bound on $\text{argmax}_{i \neq \star} \mu_{i,M}$. For both works, whenever these assumptions are not satisfied (i.e., $\lambda$'s and $\xi$'s do not respect Assumption 1 in [25], and the knowledge on $\mu_{\star,M}$, $\text{argmax}_{i \neq \star} \mu_{i,M}$ is imprecise/not available), the theoretical guarantees offered by existing algorithms are arbitrarily sub-optimal. On the other hand, our algorithm requires no additional assumptions and is the only one that matches exactly the cost complexity lower bound. Indeed, neither the cost upper bound of [25] nor the one of [31] matches the lower bound of Theorem 3.1, even when their additional hypotheses are satisfied.

## 4.2 Computing the gradient of $F(\omega, \mu)$

Algorithm 1 requires computing a sub-gradient of $F(\boldsymbol{\omega}, \boldsymbol{\mu})$. Notably, we remark that this is needed for a generic $\boldsymbol{\mu} \in \Theta^{KM}$, as $\hat{\boldsymbol{\mu}}(t)$ might violate the fidelity constraints due to inaccurate estimations or degenerate cases in which the multi-fidelity constraints are attained with equality. In this section, we provide an efficient algorithm for the computation of the sub-gradient that arises from a more in-depth study of the function $F(\boldsymbol{\omega}, \boldsymbol{\mu})$. To this end, we begin by presenting some intermediate characterization of the functions $f_{i,j}(\boldsymbol{\omega}, \boldsymbol{\mu})$ that define $F(\boldsymbol{\omega}, \boldsymbol{\mu})$.

**Lemma 4.2.** *Consider $\boldsymbol{\mu} \in \Theta^{KM}$ and $\boldsymbol{\omega} \in \Delta_{K \times M}$. Define for $k \in [K]$,*

$$\psi_k^* := \underset{\psi \in \mathbb{R}}{\text{argmin}} \sum_{m=1}^{M} \omega_{k,m} \frac{d^-(\mu_{k,m}, \psi + \xi_m) + d^+(\mu_{k,m}, \psi - \xi_m)}{\lambda_m}$$

*Then, the following holds:*

$$f_{i,j}(\boldsymbol{\omega}, \boldsymbol{\mu}) = \sum_{k \in \{i,j\}} \sum_{m=1}^{M} \omega_{k,m} \frac{d^-(\mu_{k,m}, \psi_k^* + \xi_m) + d^+(\mu_{k,m}, \psi_k^* - \xi_m)}{\lambda_m} \quad \text{if } \psi_j^* > \psi_i^* \quad (6)$$

$$f_{i,j}(\boldsymbol{\omega}, \boldsymbol{\mu}) = \inf_{\eta \in \mathbb{R}} \sum_{k \in \{i,j\}} \sum_{m=1}^{M} \omega_{k,m} \frac{d^-(\mu_{k,m}, \eta + \xi_m) + d^+(\mu_{k,m}, \eta - \xi_m)}{\lambda_m} \quad \text{otherwise.} \quad (7)$$

We further introduce $\eta_{i,j}^*$ as the minimizer in the expression in (7) [6]. When $\boldsymbol{\mu} \in \mathcal{M}_{\text{MF}}$, we can show that $\psi_k^* = \mu_{k,M}$ for all $k$ and due to the multi-fidelity constraints the expression in (6) is always equal to zero. Hence in both cases $f_{i,j}(\boldsymbol{\omega}, \boldsymbol{\mu})$ is equal to the expression in (7), which can be rewritten

$$f_{i,j}(\boldsymbol{\omega}, \boldsymbol{\mu}) = \mathbf{1}\left(\mu_{i,M} \geq \mu_{j,m}\right) \sum_{k \in \{i,j\}} \sum_{m=1}^{M} \omega_{k,m} \frac{d^-(\mu_{k,m}, \eta_{i,j}^* + \xi_m) + d^+(\mu_{k,m}, \eta_{i,j}^* - \xi_m)}{\lambda_m} .$$

This quantity can be interpreted as the transportation cost for making $\mu_{j,M}$ larger than $\mu_{i,M}$. When $\mu_i \notin \text{MF}$ or $\mu_j \notin \text{MF}$, if $\psi_j^* > \psi_i^*$, $f_{i,j}(\boldsymbol{\omega}, \boldsymbol{\mu})$ is equal to the expression (6) that can be interpreted as a transportation cost with an alternative in which $i$ and $j$ satisfy the multi-fidelity constraints.

Using this preliminary result, we provide a precise expression for the sub-gradient of $F(\boldsymbol{\omega}, \boldsymbol{\mu})$.

**Theorem 4.3.** *Consider $\boldsymbol{\mu} \in \Theta^{KM}$ and $\boldsymbol{\omega} \in \Delta_{K \times M}$ such that $F(\boldsymbol{\omega}, \boldsymbol{\mu}) > 0$ holds. Let $(i, a) \in [K]^2$ be a pair of arms that attains the max-min value in Equation (2). Then a sub-gradient $\nabla F(\boldsymbol{\omega}, \boldsymbol{\mu})$ of $F(\boldsymbol{\omega}, \boldsymbol{\mu})$ w.r.t. to $\boldsymbol{\omega}$ is given by one of the two following expressions: for $j \in \{a, i\}$ and $m \in [M]$,*

$$\nabla F(\boldsymbol{\omega}, \boldsymbol{\mu})_{j,m} = \frac{d^+(\mu_{j,m}, \eta_{i,a}^* - \xi_m) + d^-(\mu_{j,m}, \eta_{i,a}^* + \xi_m)}{\lambda_m} \quad \text{if } \psi_i^* \geq \psi_a^*, \quad (8)$$

$$\nabla F(\boldsymbol{\omega}, \boldsymbol{\mu})_{j,m} = \frac{d^+(\mu_{j,m}, \psi_j^* - \xi_m) + d^-(\mu_{j,m}, \psi_j^* + \xi_m)}{\lambda_m} \quad \text{otherwise.} \quad (9)$$

*That sub-gradient $\nabla F(\boldsymbol{\omega}, \boldsymbol{\mu})$ is 0 in all the remaining $KM - 2M$ dimensions.*

Theorem 4.3 shows how to compute a sub-gradient of $F(\boldsymbol{\omega}, \boldsymbol{\mu})$ under the mild assumption that $F(\boldsymbol{\omega}, \boldsymbol{\mu}) > 0$.[7] More specifically, it is sufficient to consider the pair $(i, a)$ that attains the max-min

---

[6]To ease the notation, we omit (most of the time) the dependence of $\psi_k^*$ and $\eta_{i,j}^*$ in $\boldsymbol{\omega}$ and $\boldsymbol{\mu}$.

[7]As we discuss in Remark C.8, $F(\tilde{\boldsymbol{\omega}}(t), \hat{\boldsymbol{\mu}}(t)) = 0$ is a rare condition, and, whenever it happens, it is possible to alter Algorithm 1 slightly without affecting its theoretical guarantees.

value in Equation (2), and then test whether $\psi_i^* \geq \psi_a^*$ holds to choose which expression to use among Equations (8) and (9). An interesting interpretation of the sub-gradient expression is that, whenever $\psi_i^* \geq \psi_a^*$, the sub-gradient is pointing toward the direction of the space that aims at increasing the information to discriminate the eventual optimality of arm $a$ against $i$. On the other hand, whenever $\psi_a^* > \psi_i^*$ holds, the sub-gradient points towards the direction of minimizing errors in the multi-fidelity constraints for arm $i$ and arm $a$ (if any).

**Computing the sub-gradient efficiently** To conclude, we notice that to compute a sub-gradient, it is required to compute $\psi_k^*$ for all arm $k$ and $\eta_{i,j}^*$ for all pairs of arms such that $\psi_i^* \geq \psi_j^*$. Using their definitions, this will require solving $\mathcal{O}(K^2)$ one-dimensional optimization problems of functions that involve $\mathcal{O}(M)$ variables, which leads to a computational complexity which is roughly $\mathcal{O}(K^2 M n)$, where $n$ is the number of iterations of the convex solver. In the following, we show that it is possible to exploit the structure of the $f_{i,j}$'s to obtain an algorithm whose total complexity is $\mathcal{O}(K^2 M^2)$ and that does not suffer from any approximation error due to the optimization procedure. Specifically, we now present a result that shows how to compute $\eta_{i,j}^*$. A similar result holds also for $\psi_k^*$ and is deferred to Appendix C.

**Lemma 4.4.** *Consider $\boldsymbol{\mu} \in \Theta^{KM}$ and $\boldsymbol{\omega} \in \Delta_{K \times M}$ such that $f_{i,j}(\boldsymbol{\omega}, \boldsymbol{\mu}) > 0$ . Suppose that $\psi_i^* \geq \psi_j^*$ holds. Then, there exists a unique minimizer $\eta_{i,j}^*(\boldsymbol{\omega})$ of Equation (7) which is the unique solution of the following equation of $\eta$:*

$$\eta = \frac{\sum_{a \in \{i,j\}} \sum_m \frac{\omega_{a,m}}{\lambda_m} \left( \overline{k}_{a,m}(\eta) \frac{\mu_{a,m} + \xi_m}{v(\eta - \xi_m)} + \underline{k}_{a,m}(\eta) \frac{\mu_{a,m} - \xi_m}{v(\eta + \xi_m)} \right)}{\sum_{a \in \{i,j\}} \sum_m \frac{\omega_{a,m}}{\lambda_m} \left( \overline{k}_{a,m}(\eta) \frac{1}{v(\eta - \xi_m)} + \underline{k}_{a,m}(\eta) \frac{1}{v(\eta + \xi_m)} \right)}, \tag{10}$$

*where $\overline{k}_{a,m}(x) = \mathbf{1}\{x \geq \mu_{a,m} + \xi_m\}$ and $\underline{k}_{a,m}(x) = \mathbf{1}\{x \leq \mu_{i,m} - \xi_m\}$.*

From Lemma 4.4, to compute $\eta_{i,j}^*$ it is sufficient to find the unique solution to the fixed point equation given in (10). To do this efficiently, we observe that the right hand side of Equation (10) depends on $\eta$ only for the presence of the indicator functions $\overline{k}_{a,m}(\eta)$ and $\underline{k}_{a,m}(\eta)$, which can only take a finite number of values. Hence, it is sufficient to evaluate the right-hand side at an arbitrary point within a given interval where the values of the indicator functions do not change. If the resulting value is within the considered interval, then this value is our fixed point. Since there are at most $\mathcal{O}(M)$ candidate fixed points, this procedure takes at most $\mathcal{O}(M^2)$ steps.

**Computational complexity remark** It follows that the per-iteration computational complexity of Algorithm 1 is $\mathcal{O}\left(K^2 M^2\right)$. The computationally efficient technique explained above indeed applies not only to the sampling rule but also to the stopping and the decision rules.[8]

## 5  Numerical experiments

We conclude this work by presenting numerical simulations whose goal is to show the empirical benefits of our approach. We compare MF-GRAD against IISE [25], and the gradient approach of [22] that simply does BAI using samples collected at fidelity $M$. We will refer to this additional baseline as GRAD. In the following, we avoided the comparison with the multi-fidelity algorithms in [31] as we ran into issues when doing experiments. We elaborate more on this point in Appendix D.6, where we provide numerical evidence of the fact those algorithms might fail at stopping, together with an argument that shows a mistake in the proofs of [31].

Given this setup, first, we test all methods on a $4 \times 5$ multi-fidelity bandit with Gaussian arms that have been randomly generated, using a risk parameter $\delta = 0.01$. Due to space constraints and for the sake of exposition, we refer the reader to Appendix D.1 for the value of $\boldsymbol{\mu}$, $\xi$'s and $\lambda$'s and details on the stopping rules calibration. We report the empirical distribution of the resulting cost complexities in Figure 1. As one can verify, MF-GRAD obtains the most competitive performance. Experiments on additional $4 \times 5$ bandits that are reported in Appendix D.3 provide a similar conclusion.

Furthermore, to illustrate the sub-optimality of IISE and GRAD from an intuitive perspective, we test our algorithm on a simple $5 \times 2$ instance that allows to easily understand why existing methods underperform MF-GRAD. Specifically, we consider $\mu_i = [0.4, 0.5]$ for all $i \in [4]$, $\mu_5 = [0.5, 0.6]$,

---

[8]Given the definition of $f_{i,j}$, it is sufficient to replace $\omega_{i,m}/\lambda_m$ in Equation (10) with $N_{a,m}(t)$.

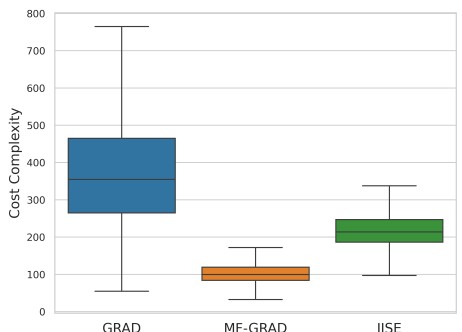

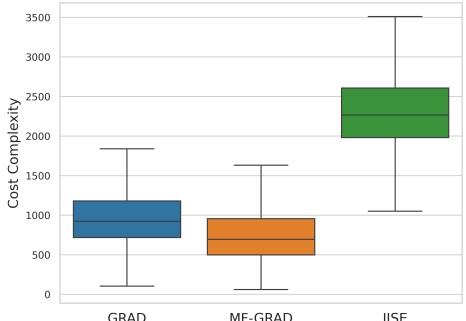

**Figure 1:** Empirical cost complexity for 1000 runs times with $\delta = 0.01$ on the $4 \times 5$ multi-fidelity bandit.

**Figure 2:** Empirical cost complexity for 1000 runs times with $\delta = 0.01$ on the $5 \times 2$ multi-fidelity bandit.

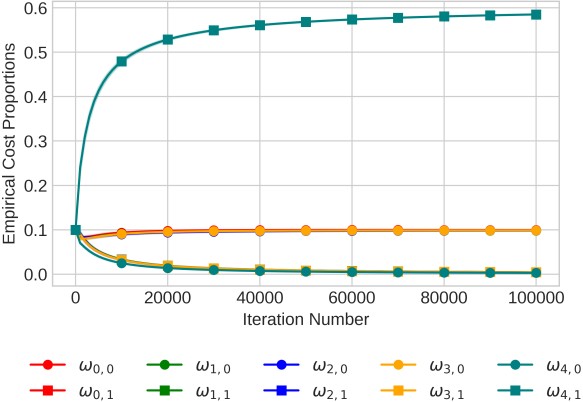

**Figure 3:** Empirical cost proportions of MF-GRAD for 100000 iterations on the $5 \times 2$ bandit model. Results are average over 100 runs and shaded area report 95% confidence intervals. Empirical cost proportions of a certain arm are plotted with the same color. Cost proportions at fidelity 1 and 2 are visualized with a circle and a squared respectively.

$\lambda = [0.5, 5]$, $\xi = [0.1, 0]$ and we report the cost complexity of the three algorithms in Figure 2. In this case, we can prove that the optimal fidelity is sparse on fidelity $m = 1$ for $i \in [4]$, and on fidelity $m = 2$ for arm 5. Furthermore, thanks to the symmetry of the problem, it is possible to show that $\omega_i^* = [0.09621, 0]$ for all $i \in [4]$, and $\omega_5^* = [0, 0.61516]$ (see Appendix D.1). As one can see, IISE obtains the worst performance in this domain. The reason is that the concept of optimal fidelity on which IISE relies is sub-optimal (i.e., according to the design principle of IISE, the optimal fidelity is $m = 2$ for all arms), and the algorithm, in practice, will discard sub-optimal arms using samples that have been collected only at fidelity $m = 2$. Nevertheless, this will only happen after a first period in which IISE tries to exploit (unsuccessfully) data at fidelity $m = 1$. GRAD, on the other hand, obtains sub-optimal performances since although most of the budget should be spent on fidelity 2 (as $\omega^*_{5,2} = 0.61516$), it never pulls the cheapest (and optimal) fidelity for arms $i \in [4]$. Finally, MF-GRAD, on the other hand, obtains the most competitive performance since, as learning progresses, its empirical cost proportions eventually approach the one prescribed by $\omega^*$. To verify this behavior, we removed the stopping rule from MF-GRAD, and let the algorithm run for $10^5$ iterations. In Figure 3, we report the entire evolution of the cost proportions during learning. As one can appreciate, at the end of this process, the empirical cost proportions of MF-GRAD are approaching the one described by $\omega^*$. [9]. Finally, we also refer the reader to Appendix D for additional results (e.g., additional domains, smaller regimes of $\delta$) and further insights.

---

[9]Furthermore, at the end of this period, we measured the distance between $\omega^*$ and the empirical cost proportions $\omega(10^5)$; it holds that $||\omega^* - \omega(10^5)||_2 \approx 0.031 \pm 0.0006$. The error has been estimated with 100 independent runs, and 0.0006 reports the 95% confidence intervals.

# 6 Conclusions

For fixed-confidence best arm identification in multi-fidelity bandits, we presented a lower bound on the cost complexity and an algorithm with a matching upper bound in the regime of high confidence. The algorithm uses features of the lower bound optimization problem in order to compute its updates efficiently. Unlike prior work, it does not require any assumption or prior knowledge on the bandit instance. Our work also confirmed the existence in most cases of an "optimal fidelity" to explore each arm in the asymptotic regime, and revealed that the intuitive such notions proposed in prior work were inaccurate. Yet, our algorithm does not need to identify these optimal fidelities in order to be asymptotically optimal.

This raises the following question: could the performance of the algorithm be enhanced by exploiting the sparsity pattern? We conjecture that estimating the optimal fidelities accurately may actually be harder than identifying the best arm. However, leveraging some sufficient conditions for $w_{a,m}^* = 0$ (such as the ones given in Proposition B.6) to eliminate some fidelities and reduce the support of the forced exploration component of the algorithm seems a promising idea. A limitation of our current analysis is that it only provides asymptotic guarantees in the high confidence regime, although our experiments reveal good performance for moderate values of $\delta$. In future work, we will seek a better understanding of the moderate confidence regime [30]. To this end, we may leverage some proof techniques from other works using online optimization that obtain finite-time bounds [4, 5]. On the lower bound side, while $C^*(\boldsymbol{\mu})$ essentially scales with $K$ due to the sparsity pattern, an interesting open question is whether there is a worse case $\mathcal{O}(KM)$ scaling in the moderate confidence regime, indicating that all fidelities do need to be explored at least a constant amount of times.

## Acknowledgments and Disclosure of Funding

This work was done while Riccardo Poiani was visiting the Scool team of Inria Lille. He acknowledges the funding of a MOB-LIL-EX grant from the University of Lille. Rémy Degenne and Emilie Kaufmann acknowledge the funding of the French National Research Agency under the project FATE (ANR22-CE23-0016-01) and the PEPR IA FOUNDRY project (ANR-23-PEIA-0003). Alberto Maria Metelli and Marcello Restelli acknowledge the funding of the European Union – Next Generation EU within the project NRPP M4C2, Investment 1.,3 DD. 341 - 15 march 2022 – FAIR – Future Artificial Intelligence Research – Spoke 4 - PE00000013 - D53C22002380006.

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

# A Table of Symbols

Table 1 reports a summary on the main symbols and the notation used throughout the paper.

**Table 1:** Notation

| Symbol | Meaning |
|---|---|
| $K, M$ | Number of arms and number of fidelity |
| $\delta \in (0,1)$ | Maximum risk parameter |
| $\tau_\delta$ | Stopping time of an algorithm |
| $c_{\tau_\delta}$ | Cost incurred at the stopping time $\tau_\delta$ |
| $a_\star(\boldsymbol{\mu})$ | $\operatorname{argmax}_{a \in [K]} \mu_{a,M}$. Often denoted simply by $\star$ |
| $\hat{a}_{\tau_\delta}$ | Arm recommended by the algorithm when it stops |
| $\boldsymbol{\mu}$ | Bandit model |
| $\mu_{i,m}$ | Mean of the $i$-th arm at fidelity $m$ within bandit model $\boldsymbol{\mu}$ |
| $\xi_m$ | Precision of fidelity $m$, i.e., $\max_{i \in [K]} \lvert \mu_{i,m} - \mu_{i,M} \rvert \le \xi_m$ |
| $\lambda_m$ | Cost incurred for gathering samples at fidelity $m$ |
| $\Theta$ | Set of possible means in the exponential family |
| $\mu_a \in \mathrm{MF}$ | Arm $a$ satisfies the multi-fidelity constraints |
| $\hat{\boldsymbol{\mu}}(t)$ | Empirical bandit model at time $t$ |
| $\mathcal{M}_{\mathrm{MF}}$ | Set of multi-fidelity bandit models |
| $\mathcal{M}_{\mathrm{MF}}^*$ | Set of multi-fidelity bandit models with a unique optimal arm |
| $d(p,q)$ | KL divergence between two distributions with means $p,q$ in the exponential family |
| $d^+(p,q), d^-(p,q)$ | $d^+(p,q) = d(p,q)\mathbf{1}\{p \le q\}, d^-(x,y) = d(p,q)\mathbf{1}\{p \ge q\}$ |
| $v(y)$ | Variance of the distribution in the exponential family with mean parameter $y$ |
| $C^*(\boldsymbol{\mu})^{-1}$ | Expression that characterizes the lower-bound on the cost-complexity |
| $\boldsymbol{\omega}, \boldsymbol{\pi}$ | Vector of cost and pull proportions respectively |
| $\boldsymbol{\omega}^*$ | $\boldsymbol{\omega}^* \in \operatorname{argmax}_{\boldsymbol{\omega} \in \Delta_{K \times M}} F(\boldsymbol{\omega}, \boldsymbol{\mu})$ |
| $f_{i,j}(\boldsymbol{\omega}, \boldsymbol{\mu})$ | Dissimilarity between arms $i$ and $j$ defined in Equation (1) |
| $F(\boldsymbol{\omega}, \boldsymbol{\mu})$ | $\max_{i \in [K]} \min_{j \ne i} f_{i,j}(\boldsymbol{\omega}, \boldsymbol{\mu})$ |
| $\Delta_{K \times M}^*(\boldsymbol{\mu})$ | Set of optimal oracle weights $\boldsymbol{\omega}^*$ for the multi-fidelity bandit model $\boldsymbol{\mu}$ |
| $\overline{\mathcal{M}}_{\mathrm{MF}}$ | Subset of multi-fidelity bandit models for which there exists a non-sparse optimal allocation $\boldsymbol{\omega}^*$ |
| $\overline{\boldsymbol{\omega}}$ | Uniform $KM$-dimensional vector $((KM)^{-1}, \ldots, (KM)^{-1})$ |
| $G > 0$ | Clipping constant in Algorithm 1 |
| $\alpha_t, \gamma_t$ | Learning rate and forced exploration rate respectively |
| $C(t)$ | Vector whose $(a,m)$-th dimension is $C_{a,m}(t)$ |
| $\boldsymbol{\omega}(t)$ | Vector of empirical cost proportions, namely $\omega_{a,m}(t) = C_{a,m}(t)(\sum_{i \in [K]} \sum_{j \in [M]} C_{i,j}(t))^{-1}$ |
| $\psi_i^*$ | Minimizer of Equation (6) |
| $\eta_{i,j}^*$ | Minimizer of Equation (7) |
| $\overline{k}_{i,m}(\eta)$ | $\mathbf{1}\{\eta \ge \mu_{i,m} + \xi_m\}$ |
| $\underline{k}_{i,m}(\eta)$ | $\mathbf{1}\{\eta \le \mu_{i,m} - \xi_m\}$ |

# B Cost complexity lower bound: proofs and derivations

## B.1 Proof of Theorem 3.1

**Theorem 3.1.** *Let $\delta \in (0,1)$. For any $\delta$-correct strategy, and any multi-fidelity bandit model $\boldsymbol{\mu} \in \mathcal{M}_{\mathrm{MF}}^*$, it holds that:*

$$\mathbb{E}_{\boldsymbol{\mu}}[c_{\tau_\delta}] \ge C^*(\boldsymbol{\mu}) \log\left(\tfrac{1}{2.4\,\delta}\right), \qquad (3)$$

*where $C^*(\boldsymbol{\mu})^{-1} := \sup_{\boldsymbol{\omega} \in \Delta_{K \times M}} F(\boldsymbol{\omega}, \boldsymbol{\mu}) = \sup_{\boldsymbol{\omega} \in \Delta_{K \times M}} \min_{a \ne \star} f_{\star,a}(\boldsymbol{\omega}, \boldsymbol{\mu})$.*

*Proof.* Consider $\delta \in (0,1)$, a multi-fidelity bandit model $\boldsymbol{\mu}$ and an alternative instance $\boldsymbol{\theta} \in \mathrm{Alt}(\boldsymbol{\mu})$ where $\mathrm{Alt}(\boldsymbol{\mu}) = \bigcup_{i \ne \star} \{\boldsymbol{\theta} \in \mathcal{M}_{\mathrm{MF}} : \theta_{a,M} > \theta_{\star,M}\}$. Then, by applying Lemma 1 in [18], we can

directly connect the expected number of draws of each arm to the KL divergence of the two multi-fidelity bandit models. More specifically, we have that:

$$\sum_{a\in[K]}\sum_{m\in[M]}\mathbb{E}_{\boldsymbol{\mu}}[N_{a,m}(\tau_\delta)]d(\mu_{a,m},\theta_{a,m}) \geq \mathrm{kl}(\delta, 1-\delta). \tag{11}$$

Then, similarly to Theorem 1 in [8], we now proceed by applying Equation (11) with all the alternative models $\boldsymbol{\theta} \in \mathrm{Alt}(\boldsymbol{\mu})$. Specifically, we have that:

$$\mathrm{kl}(\delta, 1-\delta) \leq \inf_{\boldsymbol{\theta}\in\mathrm{Alt}(\boldsymbol{\mu})}\sum_{a\in[K]}\sum_{m\in[M]}\mathbb{E}_{\boldsymbol{\mu}}[N_{a,m}(\tau_\delta)]d(\mu_{a,m},\theta_{a,m})$$

$$= \mathbb{E}_{\boldsymbol{\mu}}[c_{\tau_\delta}]\inf_{\boldsymbol{\theta}\in\mathrm{Alt}(\boldsymbol{\mu})}\sum_{a\in[K]}\sum_{m\in[M]}\frac{\mathbb{E}_{\boldsymbol{\mu}}[\lambda_m N_{a,m}(\tau_\delta)]}{\mathbb{E}_{\boldsymbol{\mu}}[c_{\tau_\delta}]}\frac{d(\mu_{a,m},\theta_{a,m})}{\lambda_m}$$

$$\leq \mathbb{E}_{\boldsymbol{\mu}}[c_{\tau_\delta}]\sup_{\boldsymbol{\omega}\in\Delta_{K\times M}}\inf_{\boldsymbol{\theta}\in\mathrm{Alt}(\boldsymbol{\mu})}\sum_{a\in[K]}\sum_{m\in[M]}\omega_{a,m}\frac{d(\mu_{a,m},\theta_{a,m})}{\lambda_m}$$

$$= \mathbb{E}_{\boldsymbol{\mu}}[c_{\tau_\delta}]\sup_{\boldsymbol{\omega}\in\Delta_{K\times M}}\min_{a\neq\star}\inf_{\substack{\boldsymbol{\theta}\in\mathcal{M}_{\mathrm{MF}}:\\\theta_{a,M}>\theta_{\star,M}}}\sum_{i,m}\omega_{i,m}\frac{d(\mu_{i,m},\theta_{i,m})}{\lambda_m}$$

$$\overset{(a)}{=} \mathbb{E}_{\boldsymbol{\mu}}[c_{\tau_\delta}]\sup_{\boldsymbol{\omega}\in\Delta_{K\times M}}\min_{a\neq\star}\inf_{\substack{\theta_a\in\mathrm{MF},\theta_\star\in\mathrm{MF}:\\\theta_{a,M}>\theta_{\star,M}}}\sum_{i\in\{\star,a\},m\in[M]}\omega_{i,m}\frac{d(\mu_{i,m},\theta_{i,m})}{\lambda_m}$$

$$= \mathbb{E}_{\boldsymbol{\mu}}[c_{\tau_\delta}]\sup_{\boldsymbol{\omega}\in\Delta_{K\times M}}\min_{a\neq\star}f_{\star,a}(\boldsymbol{\omega},\boldsymbol{\mu})$$

$$= \mathbb{E}_{\boldsymbol{\mu}}[c_{\tau_\delta}]C^*(\boldsymbol{\mu})^{-1},$$

where in $(a)$ we use that as $\boldsymbol{\mu} \in \mathcal{M}_{\mathrm{MF}}$, the minimum in $\boldsymbol{\theta}$ does not change any arm $i \notin \{\star, a\}$. Finally, we lower bound $\mathrm{kl}(\delta, 1-\delta)$ with $\log\left(\frac{1}{2.4\,\delta}\right)$, thus concluding the proof. $\qquad\square$

## B.2 Comparison with existing lower bound

In this section, we provide a comparison with the existing lower bound for the MF-BAI setting. In the following, we restrict our attention to bandits with Gaussian arms with variance $1/2$. We assume for simplicity of the exposition that $\star = 1$ and that $\mu_{1,M} > \mu_{2,M} \geq \cdots \geq \mu_{K,M}$. Given this setup, we begin by recalling Theorem 1 in [25].

**Theorem B.1** (Theorem 1 in [25]). *Consider any multi-fidelity bandit model $\boldsymbol{\mu}$ with Gaussian arms with variance $1/2$. Then, for any $\delta$-correct algorithm and $\delta \leq 0.15$ it holds that $\mathbb{E}_{\boldsymbol{\mu}}[c_{\tau_\delta}]/\log\left((2.4\delta)^{-1}\right)$ is lower bounded by:*

$$\min_{\substack{m\in[M]:\\\mu_{1,m}>\mu_{2,M}+\xi_m}}\frac{\lambda_m}{(\mu_{1,m}-(\mu_{2,M}+\xi_m))^2} + \sum_{i=1}^{K}\min_{\substack{m\in[M]:\\\mu_{1,M}-\xi_m>\mu_{i,m}}}\frac{\lambda_m}{(\mu_{i,m}-(\mu_{1,M}-\xi_m))^2}.$$

At this point, focus, for simplicity on the following $2 \times 2$ bandit model (but a trivial generalization holds for $K \times M$ bandits). We consider $\mu_{1,m} = \mu_{1,M} + \frac{\xi_m}{2}$ and $\mu_{2,m} = \mu_{2,M} - \frac{\xi_m}{2}$. Furthermore, suppose that $\mu_{1,M} = -\mu_{2,M}$. Then, let $\Delta := \mu_{1,M} - \mu_{2,M}$. Suppose that $\Delta = \xi_m$, which yields $\mu_{1,M} = \frac{\xi_m}{2}$ and $\mu_{2,M} = -\frac{\xi_m}{2}$, and that

$$\frac{\Delta}{\Delta - \frac{\xi_m}{2}} \leq \sqrt{\frac{\lambda_M}{\lambda_m}}.$$

Since $\Delta = \xi_m$, this condition actually simplifies to $\lambda_M \geq 4\lambda_m$.

Under these conditions it is possible to verify that the lower bound of [25] is given by

$$\mathbb{E}_{\boldsymbol{\mu}}[c_{\tau_\delta}] \geq \frac{2\lambda_m}{\left(\Delta - \frac{\xi_m}{2}\right)^2}\log\left(\frac{1}{2.4\delta}\right) = \frac{8\lambda_m}{\Delta^2}\log\left(\frac{1}{2.4\delta}\right).$$

At this point, consider, instead, the result that we presented in Theorem 3.1 and consider a generic weight proportion $\boldsymbol{\omega}$. From Corollary C.2, we know that:

$$F(\boldsymbol{\omega}, \boldsymbol{\mu}) = f_{1,2}(\boldsymbol{\omega}, \boldsymbol{\mu}) = \inf_{\eta \in [\mu_{2,M}, \mu_{1,M}]} \sum_{m \in [M]} \omega_{1,m} \frac{d^-(\mu_{1,m}, \eta + \xi_m)}{\lambda_m} + \omega_{2,m} \frac{d^+(\mu_{2,m}, \eta - \xi_m)}{\lambda_m}. \tag{12}$$

Then, let $\boldsymbol{\omega}^{*,M}$ be the optimal weights restricted on the portion of the simplex in which $\omega_{1,m} = \omega_{2,m} = 0$. Then, let $\eta^{*,M}$ be the optimal solution of Equation (12) when considering $\boldsymbol{\omega}^{*,M}$. Using the symmetry of the KL divergence for Gaussian distributions, it holds that $\eta^{*,M} = 0$ and $\omega_{1,2}^{*,M} = \omega_{2,2}^{*,M} = 0.5$. Then, for any $\boldsymbol{\omega}$ it holds that:

$$\begin{aligned}
F(\boldsymbol{\omega}, \boldsymbol{\mu}) &\leq \sum_{m \in [M]} \omega_{1,m} \frac{d^-(\mu_{1,m}, \eta^{*,M} + \xi_m)}{\lambda_m} + \omega_{2,m} \frac{d^+(\mu_{2,m}, \eta^{*,M} - \xi_m)}{\lambda_m} \\
&= \omega_{1,M} \frac{d(\mu_{1,M}, \eta^{*,M})}{\lambda_M} + \omega_{2,M} \frac{d(\mu_{2,M}, \eta^{*,M})}{\lambda_M} \\
&\leq F(\boldsymbol{\omega}^{*,M}, \boldsymbol{\mu}),
\end{aligned}$$

where in the second step, we have used the fact that $d(\mu_{2,m}, \eta^{*,M} - \xi_m) = d(\mu_{2,M} - \frac{\xi_m}{2}, -\xi_m) = 0$ and $d(\mu_{1,m}, \eta^{*,M} + \xi_m) = d(\mu_{1,M} + \frac{\xi_m}{2}, \xi_m) = 0$. In other words, we have shown that in this example the optimal allocation is sparse and on fidelity $M$. To conclude, we have that:

$$F(\boldsymbol{\omega}^{*,M}, \boldsymbol{\mu}) = 0.5 * \frac{d(\frac{\xi_m}{2}, 0)}{\lambda_M} + 0.5 * \frac{d(-\frac{\xi_m}{2}, 0)}{\lambda_M} = \frac{d(\frac{\xi_m}{2}, 0)}{\lambda_M} = \frac{(\Delta/2)^2}{\lambda_M} = \frac{\Delta^2}{4\lambda_M},$$

which leads to:

$$\mathbb{E}_{\boldsymbol{\mu}}[c_{\tau_\delta}] \geq \frac{4\lambda_M}{\Delta^2} \log\left((2.4\delta)^{-1}\right).$$

Under the assumptions on the problem, $\frac{4\lambda_M}{\Delta^2}$ is always larger than $\frac{8\lambda_m}{\Delta^2}$. This result says that the ratio among the lower bounds can be of order $\lambda_M / \lambda_m$, which is arbitrarily large.

### B.3 Proof of Theorem 3.2

In this section, we provide a formal proof on the sparsity of the optimal oracle allocation $\boldsymbol{\omega}^*$. The proofs given in this section rely on results that are explained in Appendix C.1.

At this point, in order to prove Theorem 3.2, we first introduce some intermediate results that will be used in the proving the theorem. Specifically, we begin by showing that, for each arm $a$, there always exists a fidelity $m$ such that $\omega_{a,m}^* > 0$ holds.

**Lemma B.2.** *Consider $\boldsymbol{\mu} \in \mathcal{M}_{\mathrm{MF}}$ and $\boldsymbol{\omega}^* \in \Delta_{K \times M}^*(\boldsymbol{\mu})$. Then, for all $a \in [K]$, there exists $m \in [M]$ such that $\omega_{a,m}^* > 0$.*

*Proof.* We split the proof into two cases. First we consider $a \neq \star$, and proceed by contradiction. Consider $\boldsymbol{\omega}^* \in \Delta_{K \times M}^*(\boldsymbol{\mu})$, and suppose there exists $a \neq \star$ such that $\omega_{\star,m}^* = 0$ for all $m \in [M]$. In this case, however, we have that:

$$F(\boldsymbol{\omega}^*, \boldsymbol{\mu}) \leq f_{\star,a}(\boldsymbol{\omega}^*, \boldsymbol{\mu}) = \inf_{\substack{\theta_a \in \mathrm{MF}, \theta_\star \in \mathrm{MF}: \\ \theta_{a,M} \geq \theta_{\star,M}}} \sum_{m=1}^M \omega_{\star,m}^* \frac{d(\mu_{\star,m}, \theta_{\star,m})}{\lambda_m} = 0, \tag{13}$$

where, in the first step we have used the definition of $F(\boldsymbol{\omega}^*, \boldsymbol{\mu})$, in the second one the fact that $\omega_{a,m}^* = 0$ for all $m \in [M]$, and in the last one we selected $\theta_{\star,m} = \mu_{\star,m}$ for all $m \in [M]$. Nevertheless, from Lemma C.6, we know that, whenever $\omega_{i,M} > 0$ holds for all $i \in [K]$, then $F(\boldsymbol{\omega}, \boldsymbol{\mu}) > 0$ holds as well. Therefore, $\boldsymbol{\omega}^* \notin \Delta_{K \times M}^*$.

The proof for the case in which $i = \star$ follows identical reasoning. $\qquad\square$

We then continue by proving that, at any optimal allocation $\boldsymbol{\omega}^*$, all the transportation costs $f_a(\boldsymbol{\omega}^*, \boldsymbol{\mu})$ are equal.

**Lemma B.3.** *Consider $\boldsymbol{\mu} \in \mathcal{M}_{\mathrm{MF}}$ and $\boldsymbol{\omega}^* \in \Delta^*_{K \times M}(\boldsymbol{\mu})$. Then, for all $a, b$ such that $a \neq \star$ and $b \neq \star$ the following holds:*

$$f_{\star,a}(\boldsymbol{\omega}^*, \boldsymbol{\mu}) = f_{\star,b}(\boldsymbol{\omega}^*, \boldsymbol{\mu}).$$

*Proof.* We introduce the following notation:

$$\mathcal{A} = \left\{ a \in [K] \setminus \{\star\} : a \in \underset{b \neq \star}{\mathrm{argmin}}\, f_b(\boldsymbol{\omega}^*, \boldsymbol{\mu}) \right\}$$

$$\mathcal{B} = ([K] \setminus \{\star\}) \setminus \mathcal{A}.$$

At this point, we proceed by contradiction. Suppose that $\mathcal{B} \neq \emptyset$. Then, for some sufficiently small $\epsilon > 0$, we define $\tilde{\boldsymbol{\omega}} \in \Delta_{K \times M}$ in the following way. For all $a \in \mathcal{A}$:

$$\tilde{\omega}_{a,M} = \omega^*_{a,M} + \epsilon/|\mathcal{A}|$$

$$\tilde{\omega}_{a,m} = \omega^*_{a,m} \quad \forall m < M.$$

For all $b \in \mathcal{B}$, instead:

$$\tilde{\omega}_{b,m_b} = \omega^*_{b,m_b} - \epsilon/|\mathcal{B}|$$

$$\tilde{\omega}_{b,m} = \omega^*_{b,m} \quad \forall m \neq m_b,$$

where $m_b \in [M]$ is any fidelity such that $\omega^*_{b,m_b} > 0$ (which exists by Lemma B.2).

Given this definition of $\tilde{\boldsymbol{\omega}}$, it is easy to see that $f_{\star,a}(\tilde{\boldsymbol{\omega}}, \boldsymbol{\mu}) > f_{\star,a}(\boldsymbol{\omega}^*, \boldsymbol{\mu})$ for all $a \in \mathcal{A}$. This is a direct consequence of the fact that $f_{\star,a}(\cdot, \boldsymbol{\mu})$ is a strictly increasing function of $\omega_{a,M}$, which is apparent from its expression from its expression for $\boldsymbol{\mu} \in \mathcal{M}_{\mathrm{MF}}$ given in Corollary C.2 and the computation of its gradient (Lemma C.3). Moreover, due to similar arguments, it also holds that $f_{\star,b}(\tilde{\boldsymbol{\omega}}, \boldsymbol{\mu}) \leq f_{\star,b}(\boldsymbol{\omega}^*, \boldsymbol{\mu})$ for all $b \in \mathcal{B}$. Using the continuity of the functions $f$, for $\varepsilon$ small enough we further have $f_{\star,a}(\tilde{\boldsymbol{\omega}}, \boldsymbol{\mu}) < f_{\star,b}(\tilde{\boldsymbol{\omega}}, \boldsymbol{\mu})$ for all $a \in \mathcal{A}$ and $b \in \mathcal{B}$. This leads to $\min_{a \neq \star} f_{\star,a}(\boldsymbol{\omega}^*, \boldsymbol{\mu}) < \min_{a \neq \star} f_{\star,a}(\tilde{\boldsymbol{\omega}}, \boldsymbol{\mu})$ which contradicts the optimality of $\boldsymbol{\omega}^*$. □

We now continue by providing necessary conditions that characterize some key properties of the oracle weights $\boldsymbol{\omega}$, which follows from the expression of the gradient of $f_{\star,a}(\boldsymbol{\omega}, \boldsymbol{\mu})$ with respect to $\boldsymbol{\omega}$ for $\boldsymbol{\mu} \in \mathcal{M}_{\mathrm{MF}}$ (Lemma C.3).

**Lemma B.4.** *Consider $\boldsymbol{\mu} \in \mathcal{M}$ and $\boldsymbol{\omega}^* \in \Delta_{K \times M}(\boldsymbol{\mu})$. Then, for all $a \neq \star$ the following conditions holds:*

$$\frac{d^+(\mu_{a,m_1}, \eta^*_{\star,a}(\boldsymbol{\omega}^*) - \xi_{m_1})}{\lambda_{m_1}} = \frac{d^+(\mu_{a,m_2}, \eta^*_{\star,a}(\boldsymbol{\omega}^*) - \xi_{m_2})}{\lambda_{m_2}} \quad \forall m_1, m_2 : \omega^*_{a,m_1}, \omega^*_{a,m_2} > 0$$

$$\frac{d^-(\mu_{\star,m_1}, \eta^*_{\star,a}(\boldsymbol{\omega}^*) + \xi_{m_1})}{\lambda_{m_1}} = \frac{d^-(\mu_{\star,m_2}, \eta^*_{\star,a}(\boldsymbol{\omega}^*) + \xi_{m_2})}{\lambda_{m_2}} \quad \forall m_1, m_2 : \omega^*_{\star,m_1}, \omega^*_{\star,m_2} > 0$$

$$\frac{d^+(\mu_{a,m_1}, \eta^*_{\star,a}(\boldsymbol{\omega}^*) - \xi_{m_1})}{\lambda_{m_1}} = \frac{d^-(\mu_{\star,m_2}, \eta^*_{\star,a}(\boldsymbol{\omega}^*) + \xi_{m_2})}{\lambda_{m_2}} \quad \forall m_1, m_2 : \omega^*_{a,m_1}, \omega^*_{\star,m_2} > 0$$

*Proof.* We begin by recalling the definition of $C^*(\boldsymbol{\mu})^{-1}$:

$$C^*(\boldsymbol{\mu})^{-1} = \sup_{\boldsymbol{\omega} \in \Delta_{K \times M}} \min_{a \neq \star} f_{\star,a}(\boldsymbol{\omega}, \boldsymbol{\mu}),$$

which, is a concave optimization problem with a non-empty feasible region. Therefore, we can apply the KKT conditions to study the properties of each local optimal point $\boldsymbol{\omega}^*$ for which the sub-derivatives exist, i.e., from Theorem 4.3 and Corollary C.2, the ones for which the following condition hold:[10]

$$\min_{a \neq \star} f_{\star,a}(\boldsymbol{\omega}) > 0. \tag{14}$$

---

[10]We notice that a global optimum point clearly satisfies this condition.

At this point, fix any arm $a$ that attains the minimum in Equation (14). Then, from the KKT conditions, we obtain the following system of inequalities:

$$\begin{cases} -\frac{\partial}{\partial \omega_{a,m}} f_{\star,a}(\boldsymbol{\omega}, \boldsymbol{\mu}) + c - b_{a,m} = 0 & \forall m \in [M] \\ -\frac{\partial}{\partial \omega_{\star,m}} f_{\star,a}(\boldsymbol{\omega}, \boldsymbol{\mu}) = 0 & \forall m \in [M] \\ b_{i,m} \omega_{i,m}^* = 0 & \forall i \in \{\star, a\}, \forall m \in [M] \\ b_{i,m} \geq 0 & \forall i \in [K], \forall m \in [M] \\ \omega_{i,m}^* \geq 0 & \forall i \in [K], \forall m \in [M] \\ \sum_{i,m} \omega_{i,m}^* = 1 \end{cases}.$$

At this point, suppose that $\omega_{i_1,m_1} > 0$, $\omega_{i_2,m_2} > 0$ for some $i_1, i_2 \in \{\star, a\}$ and some $m_1, m_2 \in [M]$, then $b_{i_1,m_1} = 0$, $b_{i_2,m_2} = 0$. As a consequence, by applying Lemma C.3 and the fact that $\boldsymbol{\mu} \in \mathcal{M}_{\mathrm{MF}}$ (i.e., see Corollary C.2), the following equations holds:

$$\frac{d^+(\mu_{a,m_1}, \eta_{\star,a}^*(\boldsymbol{\omega}^*) - \xi_{m_1})}{\lambda_{m_1}} = \frac{d^+(\mu_{a,m_2}, \eta_{\star,a}^*(\boldsymbol{\omega}^*) - \xi_{m_2})}{\lambda_{m_2}} \quad \forall m_1, m_2 : \omega_{a,m_1}^*, \omega_{a,m_2}^* > 0$$

$$\frac{d^-(\mu_{\star,m_1}, \eta_{\star,a}^*(\boldsymbol{\omega}^*) + \xi_{m_1})}{\lambda_{m_1}} = \frac{d^-(\mu_{\star,m_2}, \eta_{\star,a}^*(\boldsymbol{\omega}^*) + \xi_{m_2})}{\lambda_{m_2}} \quad \forall m_1, m_2 : \omega_{\star,m_1}^*, \omega_{\star,m_2}^* > 0$$

$$\frac{d^+(\mu_{a,m_1}, \eta_{\star,a}^*(\boldsymbol{\omega}^*) - \xi_{m_1})}{\lambda_{m_1}} = \frac{d^-(\mu_{\star,m_2}, \eta_{\star,a}^*(\boldsymbol{\omega}^*) + \xi_{m_2})}{\lambda_{m_2}} \quad \forall m_1, m_2 : \omega_{a,m_1}^*, \omega_{\star,m_2}^* > 0$$

Finally, to conclude the proof, it is sufficient to iterate these arguments for all $a \neq \star$. Indeed, from Lemma B.3, we know that all sub-optimal arms will attain the minimum in Equation (14) at a global optimum $\boldsymbol{\omega}^*$. □

At this point we are ready to prove our main result.

**Theorem 3.2.** *Let* $\Delta_{K \times M}^*(\boldsymbol{\mu}) \coloneqq \mathrm{argmax}_{\boldsymbol{\omega} \in \Delta_{K \times M}} F(\boldsymbol{\omega}, \boldsymbol{\mu})$ *and*

$$\widetilde{\mathcal{M}}_{\mathrm{MF}} \coloneqq \left\{ \boldsymbol{\mu} \in \mathcal{M}_{\mathrm{MF}}^* : \exists i \in [K], \exists m_1, m_2 \in [M]^2, \exists \boldsymbol{\omega}^* \in \Delta_{K \times M}^*(\boldsymbol{\mu}) : \omega_{i,m_1}^* > 0, \omega_{i,m_2}^* > 0 \right\}.$$

*The set* $\widetilde{\mathcal{M}}_{\mathrm{MF}}$ *is a subset of* $\mathbb{R}^{K \times M}$ *whose Lebesgue measure is zero.*

*Proof.* Let us introduce some additional notation. Consider a subset of arm-fidelity pairs $\mathcal{X} \subseteq [K] \times [M]$, and define $\mathcal{G}(\mathcal{X}) \subseteq \mathcal{M}$ as the subset of multi-fidelity bandit models $\boldsymbol{\mu}$ for which there exists $\boldsymbol{\omega}^* \in \Delta_{K \times M}^*(\boldsymbol{\mu})$ such that, for all $(i, m) \in \mathcal{X}$, $\omega_{i,m}^* > 0$ holds.

Then, fix an arm $i \neq \star$, and any three fidelity $m_1, m_2, m_3 \in [M]$, and consider $\boldsymbol{\mu} \in \mathcal{G}(\{(i, m_1), (i, m_2), (\star, m_3)\})$.[11] Then, from Lemma B.4, we know that the following condition holds:

$$\frac{d^+(\mu_{i,m_1}, \eta_{\star,i}^*(\boldsymbol{\omega}^*) - \xi_{m_1})}{\lambda_{m_1}} = \frac{d^-(\mu_{\star,m_3}, \eta_{\star,i}^*(\boldsymbol{\omega}^*) + \xi_{m_3})}{\lambda_{m_3}}. \tag{15}$$

This, in turn, implies that $\eta_{\star,i}^*(\boldsymbol{\omega}^*)$ is uniquely identified as a function of $\mu_{i,m_1}, \mu_{\star,m_3}, \xi_{m_1}, \xi_{m_3}, \lambda_{m_1}$ and $\lambda_{m_3}$. Indeed, $d^+(\mu_{i,m_1}, \eta_{\star,i}^*(\boldsymbol{\omega}^*) - \xi_{m_1})$ is a strictly increasing function of $\eta_{\star,i}^*(\boldsymbol{\omega}^*)$, while $d^-(\mu_{\star,m_3}, \eta_{\star,i}^*(\boldsymbol{\omega}^*) + \xi_{m_3})$ is a strictly decreasing function of $\eta_{\star,i}^*(\boldsymbol{\omega}^*)$. Let $c_1 = \eta_{\star,i}^*(\boldsymbol{\omega}^*)$, and let $c_2 = \frac{d^+(\mu_{i,m_1}, \eta_{\star,i}^*(\boldsymbol{\omega}^*) - \xi_{m_1})}{\lambda_{m_1}}$. At this point, since $\omega_{i,m_2}^* > 0$ holds by definition, we also know, from Lemma B.4, that the following condition has to be satisfied:

$$\frac{d^+(\mu_{i,m_2}, c_1 - \xi_{m_2})}{\lambda_{m_2}} = c_2.$$

Therefore, the value of $\mu_{i,m_2}$ is uniquely identified as a function of $\mu_{i,m_1}, \mu_{\star,m_3}, \xi_{m_1}, \xi_{m_3}, \lambda_{m_1}$ and $\lambda_{m_3}$. That function is measurable (it's a combination of $d^+$, $d^-$ and their inverses), hence $\boldsymbol{\mu}$ lies on the graph of a measurable function, and such a graph has Lebesgue measure 0. Therefore,

---

[11]Similar arguments hold also for $i = \star$.

$\mathcal{G}(\{(i, m_1), (i, m_2), (\star, m_3)\})$ has measure 0. At this point, thanks to Lemma B.2, we know that, for arm $\star$, there always exists at least a fidelity $m_3$ such that $\omega^*_{\star, m_3} > 0$. We thus have that

$$\mathcal{G}((i, m_1), (i, m_2)) \subseteq \bigcup_{m_3 \in [M]} \mathcal{G}((i, m_1), (i, m_2), (\star, m_3)) .$$

Since $\mathcal{G}((i, m_1), (i, m_2))$ is contained in a set which is a countable union of null measure sets, it has null measure.

To conclude the proof, we notice that:

$$\widetilde{\mathcal{M}}_{\mathrm{MF}} \subseteq \bigcup_{i=1}^{K} \bigcup_{m_1, m_2 \in [M]^2} \mathcal{G}(\{(i, m_1), (i, m_2)\}) := \mathcal{Y}.$$

The proof follows from the fact that (i) $\mathcal{Y}$ is a countable union of set of null measure (and, consequently, has null measure), and (ii) $\widetilde{\mathcal{M}}_{\mathrm{MF}} \subseteq \mathcal{Y}$. $\qquad\square$

## B.4 Additional results on the sparsity of the oracle weights

In this section, we present additional results on the sparsity of the oracle weights. Specifically:

(i) We identify a specific class of multi-fidelity bandit models in which the optimal allocation is sparse. In particular, within this class of MF-bandit models, the optimal allocation have non-zero values only at the cheapest fidelity.

(ii) We then provide sufficient conditions to determine whether some fidelity have zero weights at any optimal weight vector $\boldsymbol{\omega}^*$

We now proceed by constructing the class of multi-fidelity bandits that we mentioned in point (i) above. In this construction, we will consider Gaussian multi-fidelity bandits with variance $\frac{1}{2}$. Then, for any number of arms $K$ and fidelity $M$, we will denote with $\mathcal{A}_{KM}$, the set of Gaussian multi-fidelity bandits that satisfy the following construction. We start by building the means of the arms at the highest fidelity $M$. Specifically, we consider a generic $\mu_{\star, m} > 0$, and let $\mu_{a, M} = -\mu_{\star, m}$ for all $a \neq \star$. Then, for each fidelity $m < M$, and any values of $\lambda_m$ and $\xi_m$, we let $\mu_{i, m} = \mu_{i, M} - \xi_m$ for all $i \neq \star$, and $\mu_{\star, m} = \mu_{\star, M} + \xi_m$. Finally, to simplify some computations, we set $\sigma^2$ of each Gaussian distribution to $\frac{1}{2}$.

**Proposition B.5.** *For all $\boldsymbol{\mu} \in \mathcal{A}_{KM}$, and any $\boldsymbol{\omega}^* \in \Delta^*_{K \times M}$, it holds that, for all $a \in [K]$ and all $m > 1$, $\omega^*_{a, m} = 0$.*

*Proof.* To prove the result, starting from Corollary C.2, it is sufficient to notice that, for all $\boldsymbol{\mu} \in \mathcal{A}_{KM}$ and all $a \neq \star$, $f_{\star, a}(\boldsymbol{\omega}, \boldsymbol{\mu})$ can be rewritten as:

$$f_{\star, a}(\boldsymbol{\omega}, \boldsymbol{\mu}) = \inf_{\eta \in [\mu_{a, M}, \mu_{\star, m}]} \sum_{i \in \{\star, a\}} \sum_{m=1}^{M} \omega_{i, m} \frac{d(\mu_{i, M}, \eta)}{\lambda_m}. \tag{16}$$

Specifically, Equation (16) follows directly from the symmetric property of KL divergence for Gaussian distributions, and by the construction of $\boldsymbol{\mu}$. The proof then continue by contradiction. Suppose there exists $\boldsymbol{\omega}^*$ such that there exists $(i, m)$ (with $m > 1$) such that $\omega^*_{i, m} > 0$. By defining $\tilde{\boldsymbol{\omega}}$ as the vector which is equal to $\boldsymbol{\omega}^*$ except in the components $(i, m)$ and $(a, 1)$ for all $a \in [K]$. More specifically, for a sufficiently small $\epsilon > 0$, we define $\tilde{\omega}_{i, m} = \omega^*_{i, m} - \epsilon$ and $\tilde{\omega}_{a, 1} = \omega^*_{a, 1} + \epsilon/(K)$ for all $a \in [K]$. Then, it is easy to see that $f_{\star, a}(\tilde{\boldsymbol{\omega}}, \boldsymbol{\mu}) > f_{\star, a}(\boldsymbol{\omega}^*, \boldsymbol{\mu})$ holds for all $a \neq \star$, thus contradicting the optimality of $\boldsymbol{\omega}^*$. $\qquad\square$

Finally, we now provide sufficient conditions to determine whether some fidelity have zero weights at any optimal weight vector $\boldsymbol{\omega}^*$

**Proposition B.6.** *Fix $a \neq \star$. Then, if $\mu_{a, m} + \xi_m \geq \mu_{\star, m}$, then it holds that $\omega^*_{a, m} = 0$. Furthermore, if $\mu_{\star, m} - \xi_m \leq \mu_{j, M}$ for all $j \neq *$, then it holds that $\omega^*_{\star, m} = 0$.*

*Proof.* Consider $a \neq \star$, and let us analyze $f_{\star,a}(\boldsymbol{\omega}, \boldsymbol{\mu})$ for any $\boldsymbol{\omega} \in \Delta_{K \times M}$. More specifically, we recall from Corollary C.2, that the only term in which $\omega_{a,m}$ plays a role is the following one:

$$\omega_{a,m} \frac{d(\mu_{a,m}, \eta^*_{\star,a}(\boldsymbol{\omega}) - \xi_m)}{\lambda_m} \mathbf{1} \left\{ \eta^*_{\star,a}(\boldsymbol{\omega}) \geq \mu_{a,m} + \xi_m \right\}. \tag{17}$$

Nevertheless, since $\eta^*_{\star,a}(\boldsymbol{\omega}) \leq \mu_{\star,m} \leq \mu_{a,m} + \xi_m$, we have that Equation (17) is always equal to $0$ for all $\boldsymbol{\omega} \in \Delta_{K \times M}$. To prove the result we now proceed by contradiction. Suppose that $\boldsymbol{\omega}^*$ is such that $\omega^*_{a,m} > 0$. Then, consider $\tilde{\boldsymbol{\omega}}$ as a vector which is equal to $\boldsymbol{\omega}^*$ except in the components $(a, m)$ and $(i, M)$ for all $i \neq \star$. More specifically, for a sufficiently small $\epsilon > 0$, we define $\tilde{\omega}_{a,m} = \omega^*_{a,m} - \epsilon$ and $\tilde{\omega}_{i,M} = \omega^*_{i,M} + \epsilon/(K-1)$ for all $i \neq \star$. At this point, by noticing that $f_{\star,i}(\boldsymbol{\omega}, \boldsymbol{\mu})$ is strictly increasing in $\omega_{i,M}$ (i.e., due to Theorem 4.3 and the fact that $\boldsymbol{\mu} \in \mathcal{M}_{\mathrm{MF}}$), and since $f_{\star,a}(\boldsymbol{\omega}, \boldsymbol{\mu})$ is not affected by the value of $\omega_{a,m}$ (i.e., Equation (17)), we have that $f_{\star,i}(\tilde{\boldsymbol{\omega}}, \boldsymbol{\mu}) > f_{\star,i}(\boldsymbol{\omega}^*, \boldsymbol{\mu})$ for all $i \neq *$, thus contradicting the optimality of $\boldsymbol{\omega}^*$.

To show that if $\mu_{\star,m} - \xi_m \leq \mu_{j,M}$ for all $j \neq *$, then it holds that $\omega^*_{\star,m} = 0$, it is possible to follow identical reasonings. The only difference is that the term $\omega_{\star,m}$ plays a role in each of the $(K-1)$-equations defining $F(\boldsymbol{\omega}, \boldsymbol{\mu})$, namely:

$$\omega_{\star,m} \frac{d(\mu_{\star,m}, \eta^*_{\star,a}(\boldsymbol{\omega}) + \xi_m)}{\lambda_m} \mathbf{1} \left\{ \eta^*_{\star,a}(\boldsymbol{\omega}) \leq \mu_{1,m} - \xi_m \right\} \quad \forall a \neq \star. \tag{18}$$

Nevertheless, Equation (18) is equal to $0$ for all $a \neq \star$ since $\eta^*_{\star,a}(\boldsymbol{\omega}) \geq \mu_{i,m} - \xi_m \geq \mu_{a,M}$ holds for all $\boldsymbol{\omega}$ and all $a \neq \star$. The proof then follows by an identical construction of an alternative weight vector $\tilde{\boldsymbol{\omega}}$ which increases the objective function. $\qquad\square$

### B.5 Sub-optimality of "optimal" fidelity of previous works

In this section, we discuss how the concept of "optimal" fidelity of previous works (i.e., [25] and [31]) fails to satisfy the notion of optimal fidelity that arises from the tighter lower bound that we presented in Section 3. In this section, we consider as example $2 \times 2$ multi-fidelity bandit models with Gaussian distributions. To ease the notation, we will consider $\mu_{1,M} > \mu_{2,M}$.

#### B.5.1 Case 1

We notice that [25] provided the two concepts of optimal fidelity. The first one is from their Theorem 1. This same concept was then considered later in [31]. A fidelity $m$ is optimal for a certain arm $a \in [K]$ if it satisfies the following condition:

$$m^*_a \in \underset{m \in [M]}{\mathrm{argmax}} \frac{\mu_{1,M} - (\mu_{a,m} + \xi_m)}{\sqrt{\lambda_m}} \quad \text{if } a \neq 1 \tag{19}$$

$$m^*_a \in \underset{m \in [M]}{\mathrm{argmax}} \frac{(\mu_{a,m} - \xi_m) - \mu_{2,M}}{\sqrt{\lambda_m}} \quad \text{if } a = 1 \tag{20}$$

Then, consider the following $2 \times 2$ example of multi-fidelity BAI problem. Let $\xi_1 = 0.1$, $\mu_{1,M} = 0.6$, $\mu_{1,m} = 0.65$, $\mu_{2,M} = 0.5$, $\mu_{2,m} = 0.45$ (where we use the notation $M = 2$ for the maximal fidelity and $m = 1$). Suppose, furthermore, that all distributions are Gaussian. In this case, from Equation (19)-(20), we have that $m^*_1 = 1$ and $m^*_2 = 1$ whenever the following conditions are satisfied:

$$\frac{\mu_{1,M} - (\mu_{2,m} + \xi_m)}{\sqrt{\lambda_m}} > \frac{\mu_{1,M} - \mu_{2,M}}{\sqrt{\lambda_M}}$$

$$\frac{\mu_{1,m} - \xi_m - \mu_{2,M}}{\sqrt{\lambda_m}} > \frac{\mu_{1,M} - \mu_{2,M}}{\sqrt{\lambda_M}}.$$

Plugging in the numerical values, we obtain in both cases

$$\frac{0.05}{\sqrt{\lambda_m}} > \frac{0.1}{\sqrt{\lambda_M}},$$

thus showing that, according to [31], the optimal fidelity for both arms is $m = 1$ whenever $\sqrt{\frac{\lambda_M}{\lambda_m}} > \frac{0.1}{0.05}$.

At this point, consider the expression of $F(\boldsymbol{\omega}, \boldsymbol{\mu}) = f_{1,2}(\boldsymbol{\omega}, \boldsymbol{\mu})$ in this particular example. Then, it is possible to show that, for any $\boldsymbol{\omega} \in \Delta_{2 \times 2}$ such that $\omega_{1,M} = \omega_{2,M} = 0$, then, $f_{1,2}(\boldsymbol{\omega}, \boldsymbol{\mu}) = 0$. Specifically, we have that $F(\boldsymbol{\omega}, \boldsymbol{\mu})$ is given by:

$$\inf_{\eta \in [\mu_{2,M}, \mu_{1,M}]} \omega_{1,m} \frac{d(\mu_{1,m}, \eta + \xi_m) \mathbf{1}\{\eta \leq \mu_{1,m} - \xi_m\}}{\lambda_m} + \omega_{a,m} \frac{d(\mu_{2,m}, \eta - \xi_m) \mathbf{1}\{\eta \geq \mu_{a,m} + \xi_m\}}{\lambda_m}$$

In turn, this is equal to:

$$\inf_{\eta \in [0.5, 0.6]} \omega_{1,m} \frac{d(0.55, \eta) \mathbf{1}\{\eta \leq 0.55\}}{\lambda_m} + \omega_{a,m} \frac{d(0.55, \eta) \mathbf{1}\{\eta \geq 0.55\}}{\lambda_m},$$

which is always 0 for $\eta = 0.55$.

On the other hand, Lemma C.6, shows that any strategy that gives positive value to weights at fidelity $M = 2$ obtains $F(\boldsymbol{\omega}, \boldsymbol{\mu}) > 0$.

### B.5.2 Case 2

Furthermore, [25] provided also the following concept of optimal fidelity which only holds for sub-optimal arms (see Definition 1 in [25]). A fidelity $m$ such that $\mu_{1,M} - \mu_{2,M} > 4\xi_m$ holds is said to be optimal for arm $a \neq 1$ if the following holds:

$$\frac{\lambda_m}{(\mu_{1,M} - \mu_{a,M} - 4\xi_m)^2} \leq \min_{\bar{m} > m} \frac{\lambda_{\bar{m}}}{(\mu_{1,M} - \mu_{a,M} - 4\xi_m)^2}. \tag{21}$$

At this point, consider the following classes of multi-fidelity bandit models: $\mu_{2,m} = \mu_{2,M} - \xi_m$, $\mu_{1,m} = \mu_{1,M} + \xi_m$, $\mu_{1,M} - \mu_{2,M} \leq 4\xi_m$. In this case, from Equation (20) it follows that the optimal fidelity for arm 2 is always $M$. Nevertheless, since $\mu_{2,m} = \mu_{2,M} - \xi_m$, $\mu_{1,m} = \mu_{1,M} + \xi_m$, we know from Proposition B.5 $\omega_{1,M} = \omega_{2,M} = 0$.

## C  Algorithm analysis

### C.1  Gradient computation

We start by analyzing a salient feature of $f_{i,j}(\boldsymbol{\omega}, \boldsymbol{\mu})$ that holds for any $\boldsymbol{\mu} \in \Theta^{KM}$.

**Lemma C.1.** *Consider $\boldsymbol{\mu} \in \Theta^{KM}$. Fix any $\boldsymbol{\omega} \in \Delta_{K \times M}$ and $i, j \in [K]$. Let $\boldsymbol{\theta}^*$ be the solution of the following optimization problem:*

$$\boldsymbol{\theta}^* \in \underset{\substack{\theta_i \in \mathrm{MF}, \theta_j \in \mathrm{MF}: \\ \theta_{j,M} \geq \theta_{i,M}}}{\operatorname{argmin}} \sum_{m \in [M]} \omega_{j,m} \frac{d(\mu_{j,m}, \theta_{j,m})}{\lambda_m} + \sum_{m \in [M]} \omega_{i,m} \frac{d(\mu_{i,m}, \theta_{i,m})}{\lambda_m}.$$

*Furthermore, define for $k \in \{i, j\}$:*

$$\overline{M}_k(\boldsymbol{\omega}, \boldsymbol{\mu}, \boldsymbol{\theta}^*) := \{m \in [M-1] : \theta^*_{k,M} > \mu_{k,m} + \xi_m\}$$

$$\underline{M}_k(\boldsymbol{\omega}, \boldsymbol{\mu}, \boldsymbol{\theta}^*) := \{m \in [M-1] : \theta^*_{k,M} < \mu_{k,m} - \xi_m\}.$$

*Then, for $k \in \{i, j\}$ we have that*

$$\theta^*_{k,m} = \mu_{k,m} \qquad \forall m \in [M] \setminus \left( \overline{M}_k(\boldsymbol{\omega}, \boldsymbol{\mu}, \boldsymbol{\theta}^*) \cup \underline{M}_k(\boldsymbol{\omega}, \boldsymbol{\mu}, \boldsymbol{\theta}^*) \right) \tag{22}$$

$$\theta^*_{k,m} = \theta^*_{k,M} - \xi_m \qquad \forall m \in \overline{M}_k(\boldsymbol{\omega}, \boldsymbol{\mu}, \boldsymbol{\theta}^*) \tag{23}$$

$$\theta^*_{k,m} = \theta^*_{k,M} + \xi_m \qquad \forall m \in \underline{M}_k(\boldsymbol{\omega}, \boldsymbol{\mu}, \boldsymbol{\theta}^*) \tag{24}$$

*In particular,*

$$f_{i,j}(\boldsymbol{w}, \boldsymbol{\mu}) = \sum_{k \in \{i,j\}} \sum_{m \in [M]} \omega_{k,m} \frac{d^-(\mu_{k,m}, \theta^*_{k,M} + \xi_m) + d^+(\mu_{k,m}, \theta^*_{k,M} - \xi_m)}{\lambda_m}$$

$$= \min_{\theta_{j,M} \geq \theta_{i,M}} \sum_{k \in \{i,j\}} \sum_{m \in [M]} \omega_{k,m} \frac{d^-(\mu_{k,m}, \theta_{k,M} + \xi_m) + d^+(\mu_{k,m}, \theta_{k,M} - \xi_m)}{\lambda_m}$$

*Proof.* We begin by proving Equation (22). To this end, it is sufficient to notice that, given a fixed $\theta^*_{k,M}$, it is possible to set $\theta^*_{k,m} := \mu_{k,m}$, whenever the following condition is satisfied:

$$|\theta^*_{k,M} - \mu_{k,m}| \leq \xi_m. \tag{25}$$

The condition is Equation (25) is equivalent to requiring $m \in [M] \setminus \left( \overline{\mathcal{M}}_k(\boldsymbol{\omega}, \boldsymbol{\mu}, \boldsymbol{\theta}^*) \cup \underline{\mathcal{M}}_k(\boldsymbol{\omega}, \boldsymbol{\mu}, \boldsymbol{\theta}^*) \right)$, which concludes the first part of the proof.

We continue by proving Equation (23). Consider $m \in \overline{M}_k(\boldsymbol{\omega}, \boldsymbol{\mu}, \boldsymbol{\theta}^*)$, that is $\theta^*_{k,m} > \mu_{k,m} + \xi_m$. From this condition, it directly follows that $\theta^*_{k,m} > \mu_{k,m}$; therefore, since $d(\mu_{k,m}, x)$ is increasing in $x$, it follows that, in order to attain the argmin, we need to pick the smallest value of $\theta_{k,m}$ that satisfies the multi-fidelity constraint $|\theta^*_{k,M} - \theta_{k,m}|$, that is $\theta^*_{k,M} - \xi_m$.

The proof of Equation (23) follows is almost identical to the one of Equation (24); it is sufficient to replace the definition of $\overline{M}_k(\boldsymbol{\omega}, \boldsymbol{\mu}, \boldsymbol{\theta}^*)$ with $\underline{M}_k(\boldsymbol{\omega}, \boldsymbol{\mu}, \boldsymbol{\theta}^*)$. □

At this point, we continue by analyzing in more detail the function $f_{i,j}(\boldsymbol{\omega}, \boldsymbol{\mu})$.

**Lemma 4.2.** *Consider $\boldsymbol{\mu} \in \Theta^{KM}$ and $\boldsymbol{\omega} \in \Delta_{K \times M}$. Define for $k \in [K]$,*

$$\psi^*_k := \underset{\psi \in \mathbb{R}}{\arg\min} \sum_{m=1}^M \omega_{k,m} \frac{d^-(\mu_{k,m}, \psi + \xi_m) + d^+(\mu_{k,m}, \psi - \xi_m)}{\lambda_m}$$

*Then, the following holds:*

$$f_{i,j}(\boldsymbol{\omega}, \boldsymbol{\mu}) = \sum_{k \in \{i,j\}} \sum_{m=1}^M \omega_{k,m} \frac{d^-(\mu_{k,m}, \psi^*_k + \xi_m) + d^+(\mu_{k,m}, \psi^*_k - \xi_m)}{\lambda_m} \quad \text{if } \psi^*_j > \psi^*_i \tag{6}$$

$$f_{i,j}(\boldsymbol{\omega}, \boldsymbol{\mu}) = \inf_{\eta \in \mathbb{R}} \sum_{k \in \{i,j\}} \sum_{m=1}^M \omega_{k,m} \frac{d^-(\mu_{k,m}, \eta + \xi_m) + d^+(\mu_{k,m}, \eta - \xi_m)}{\lambda_m} \quad \text{otherwise.} \tag{7}$$

*Proof.* The proof follows by analyzing the definition of $f_{i,j}(\boldsymbol{\omega}, \boldsymbol{\mu})$. Consider $\boldsymbol{\theta}^*_i, \boldsymbol{\theta}^*_j$ that attaines the minimum in Equation (1). Then, there are two possibilities: either $\theta^*_{i,M} = \theta^*_{j,M}$ or $\theta^*_{j,M} > \theta^*_{i,M}$.

Suppose that $\theta^*_{j,M} > \theta^*_{i,M}$, then we notice that the optimization problem in $f_{i,j}(\boldsymbol{\omega}, \boldsymbol{\mu})$ is a 2D-convex optimization problem in the variables $\theta_{j,M}, \theta_{i,M}$ (thanks to Lemma C.1). Therefore, since the minimum of the constrained problem is such that $\theta_{j,M} > \theta_{i,M}$, than, by the convexity of the problem, this is also a minimum for the unconstrained problem, thus leading to:

$$f_{i,j}(\boldsymbol{\omega}, \boldsymbol{\mu}) = \inf_{\theta_i \in \text{MF}, \theta_j \in \text{MF}} \sum_{k \in \{i,j\}} \sum_{m \in [M]} \omega_{k,m} \frac{d(\mu_{k,m}, \theta_{k,m})}{\lambda_m}.$$

At this point, we notice that the constraints in the previous optimization problem are only intra-arm. Therefore, we can rewrite $f_{i,j}(\boldsymbol{\omega}, \boldsymbol{\mu})$ as:

$$f_{i,j}(\boldsymbol{\omega}, \boldsymbol{\mu}) = \sum_{k \in \{i,j\}} \inf_{\theta_k \in \text{MF}} \sum_{m \in [M]} \omega_{k,m} \frac{d(\mu_{k,m}, \theta_{k,m})}{\lambda_m}.$$

Furthermore, applying the same reasoning as in the proof of Lemma C.1, we can further rewrite $f_{i,j}(\boldsymbol{\omega}, \boldsymbol{\mu})$ as follows:

$$f_{i,j}(\boldsymbol{\omega}, \boldsymbol{\mu}) = \sum_{k \in \{i,j\}} \inf_{\psi \in \mathbb{R}} \sum_{m \in [M]} \omega_{k,m} \frac{d^+(\mu_{k,m}, \psi - \xi_m) + d^-(\mu_{k,m}, \psi + \xi_m)}{\lambda_m}$$

$$= \sum_{k \in \{i,j\}} \sum_{m \in [M]} \omega_{k,m} \frac{d^-(\mu_{k,m}, \psi^*_k + \xi_m) + d^+(\mu_{k,m}, \psi^*_k - \xi_m)}{\lambda_m}.$$

At this point, we notice that due to Lemma C.1 we know that $\theta^*_{j,M} = \psi^*_j$ and $\theta^*_{i,M} = \psi^*_i$, thus concluding the first part of the proof.

Consider now the case in which $\theta^*_{j,M} = \theta^*_{i,M}$ holds. Then, applying Lemma C.1, and using $\theta^*_{i,M} = \theta^*_{j,M}$, we can rewrite $f_{i,j}(\boldsymbol{\omega}, \boldsymbol{\mu})$ as follows:

$$f_{i,j}(\boldsymbol{\omega}, \boldsymbol{\mu}) = \inf_{\eta \in \mathbb{R}} \sum_{k \in \{i,j\}} \sum_{m=1}^{M} \frac{\omega_{k,m}}{\lambda_m} \left( d^+(\mu_{k,m}, \eta - \xi_m) + d^-(\mu_{k,m}, \eta + \xi_m) \right),$$

thus concluding the proof. $\qquad \square$

Given this result, we recall that the definitions of

$$\psi^*_i = \operatorname*{argmin}_{\psi \in \mathbb{R}} \sum_{m \in [M]} \omega_{i,m} \frac{d^-(\mu_{i,m}, \psi + \xi_m) + d^+(\mu_{i,m}, \psi - \xi_m)}{\lambda_m} \qquad (26)$$

$$\eta^*_{i,j} = \operatorname*{argmin}_{\eta \in \mathbb{R}} \sum_{k \in \{i,j\}} \sum_{m \in [M]} \omega_{k,m} \frac{d^-(\mu_{k,m}, \eta + \xi_m) + d^+(\mu_{k,m}, \eta - \xi_m)}{\lambda_m}. \qquad (27)$$

**Corollary C.2.** *Consider* $\boldsymbol{\mu} \in \mathcal{M}_{\mathrm{MF}}$, *and* $a \in [K]$ *such that* $a \neq \star$. *Then it holds that:*

$$
\begin{aligned}
f_{\star,a}(\boldsymbol{\omega}, \boldsymbol{\mu}) =& \inf_{\eta \in [\mu_{a,M}, \mu_{\star,M}]} \sum_{i \in \{\star,a\}} \sum_{m=1}^{M} \omega_{i,m} \frac{d^-(\mu_{i,m}, \eta + \xi_m) + d^+(\mu_{i,m}, \eta - \xi_m)}{\lambda_m} \\
=& \inf_{\eta \in [\mu_{a,M}, \mu_{\star,M}]} \sum_{m=1}^{M} \omega_{\star,m} \frac{d^-(\mu_{\star,m}, \eta + \xi_m)}{\lambda_m} + \omega_{a,m} \frac{d^+(\mu_{a,m}, \eta - \xi_m)}{\lambda_m}
\end{aligned}
$$

*Proof.* At this point, we notice that whenever $\boldsymbol{\mu} \in \mathcal{M}_{\mathrm{MF}}$ it holds that $f_{\star,a}$ can always be expressed as Equation (7). This is direct by the condition on $\psi$'s in Lemma 4.2. Furthermore, it also holds at $\eta^*_{\star,a}$ that $d^-(\mu_{a,m}, \eta^*_{\star,a} + \xi_m) = 0$, and $d^+(\mu_{\star,m}, \eta^*_{\star,a} - \xi_m) = 0$. This is a consequence of the fact that $\eta^*_{\star,a} \in [\mu_{a,M}, \mu_{\star,M}]$ for all weights $\boldsymbol{\omega}$. Indeed, $\eta^*_{\star,a} \in [\mu_{a,M}, \mu_{\star,M}]$ holds due to monotonicity property of the KL divergence. $\qquad \square$

We now analyze in more detail Equations (26) and (27). In particular, we begin by focusing on Equation (26). Taking the gradient in Equation (27) w.r.t. the optimization variable $\eta$, and setting it equal to 0, we obtain that any optimal point $\eta^*_{i,j}(\boldsymbol{\omega})$ needs to satisfy the following equation:

$$\eta \left( \sum_{a \in \{i,j\}} \sum_{m=1}^{M} \frac{\omega_{a,m}}{\lambda_m} \left( \overline{k}_{a,m} \frac{1}{v(\eta - \xi_m)} + \underline{k}_{a,m} \frac{1}{v(\eta + \xi_m)} \right) \right) = \qquad (28)$$

$$\sum_{a \in \{i,j\}} \sum_{m=1}^{M} \frac{\omega_{a,m}}{\lambda_m} \left( \overline{k}_{a,m} \frac{\mu_{a,m} + \xi_m}{v(\eta - \xi_m)} + \underline{k}_{a,m} \frac{\mu_{a,m} - \xi_m}{v(\eta + \xi_m)} \right).$$

where we recall that $\overline{k}_{a,m}(\eta)$ and $\underline{k}_{a,m}(\eta)$ are given by:

$$\overline{k}_{a,m}(\eta) = \mathbf{1}\{\eta \geq \mu_{a,m} + \xi_m\}$$
$$\underline{k}_{a,m}(\eta) = \mathbf{1}\{\eta \leq \mu_{a,m} - \xi_m\}.$$

Given this intermediate result, we now investigate in more depth the solution of Equation (28).

**Lemma 4.4.** *Consider* $\boldsymbol{\mu} \in \Theta^{KM}$ *and* $\boldsymbol{\omega} \in \Delta_{K \times M}$ *such that* $f_{i,j}(\boldsymbol{\omega}, \boldsymbol{\mu}) > 0$. *Suppose that* $\psi^*_i \geq \psi^*_j$ *holds. Then, there exists a unique minimizer* $\eta^*_{i,j}(\boldsymbol{\omega})$ *of Equation* (7) *which is the unique solution of the following equation of* $\eta$:

$$\eta = \frac{\sum_{a \in \{i,j\}} \sum_m \frac{\omega_{a,m}}{\lambda_m} \left( \overline{k}_{a,m}(\eta) \frac{\mu_{a,m} + \xi_m}{v(\eta - \xi_m)} + \underline{k}_{a,m}(\eta) \frac{\mu_{a,m} - \xi_m}{v(\eta + \xi_m)} \right)}{\sum_{a \in \{i,j\}} \sum_m \frac{\omega_{a,m}}{\lambda_m} \left( \overline{k}_{a,m}(\eta) \frac{1}{v(\eta - \xi_m)} + \underline{k}_{a,m}(\eta) \frac{1}{v(\eta + \xi_m)} \right)}, \qquad (10)$$

*where* $\overline{k}_{a,m}(x) = \mathbf{1}\{x \geq \mu_{a,m} + \xi_m\}$ *and* $\underline{k}_{a,m}(x) = \mathbf{1}\{x \leq \mu_{i,m} - \xi_m\}$.

*Proof.* Let us analyze:

$$f_{i,j}(\boldsymbol{\omega}, \boldsymbol{\mu}) = \inf_{\eta \in \mathbb{R}} \sum_{a \in \{i,j\}} \sum_{m \in M} \omega_{a,m} \frac{d^-(\mu_{a,m}, \eta + \xi_m)}{\lambda_m} + \omega_{a,m} \frac{d^+(\mu_{a,m}, \eta - \xi_m)}{\lambda_m}$$

$$:= \inf_{\eta \in \mathbb{R}} g_{i,j}(\boldsymbol{\omega}, \boldsymbol{\mu}, \eta).$$

At this point, we proceed by contradiction. Suppose that there exists $x_1, x_2 \in$ $\mathrm{argmin}_{\eta \in \mathbb{R}} g_{i,j}(\boldsymbol{\omega}, \boldsymbol{\mu}, \eta)$ such that $x_1 \neq x_2$. From the convexity of $g_{i,j}(\boldsymbol{\omega}, \boldsymbol{\mu}, \eta)$ w.r.t. $\eta$, we know that any $x \in [x_1, x_2]$ belongs to the argmin set as well. Furthermore, for all $x \in [x_1, x_2]$, since $f_{i,j}(\boldsymbol{\omega}, \boldsymbol{\mu}) > 0$, at least one of the following condition is satisfied:

- $\frac{\omega_{a,M}}{\lambda_M} d(\mu_{a,M}, x) > 0$ holds for some $a \in \{i, j\}$

- $\frac{\omega_{a,m}}{\lambda_m} d(\mu_{a,M}, x + \xi_m) \underline{k}_{a,m}(x) > 0$ holds for some $a \in \{i, j\}$ and some fidelity $m < M$

- $\frac{\omega_{a,m}}{\lambda_m} d(\mu_{a,M}, x - \xi_m) \overline{k}_{a,m}(x) > 0$ holds for some $a \in \{i, j\}$ and some fidelity $m < M$

Therefore, from Equation (28), we obtain that all $x \in [x_1, x_2]$ are fixed points of the following Equation:

$$x = \frac{\sum_{a \in \{i,j\}} \sum_{m=1}^{M} \frac{\omega_{a,m}}{\lambda_m} \left( \overline{k}_{a,m}(x) \frac{\mu_{a,m} + \xi_m}{v(x - \xi_m)} + \underline{k}_{a,m}(x) \frac{\mu_{a,m} - \xi_m}{v(x + \xi_m)} \right)}{\left( \sum_{a \in \{i,j\}} \sum_{m=1}^{M} \frac{\omega_{a,m}}{\lambda_m} \left( \overline{k}_{a,m}(x) \frac{1}{v(x - \xi_m)} + \underline{k}_{a,m}(x) \frac{1}{v(x + \xi_m)} \right) \right)}, \tag{29}$$

At this point, we notice that for any couple of different $\tilde{x}_1, \tilde{x}_2$ that satisfies Equation (29), there exists at least one arm $a \in \{i, j\}$ and one fidelity $m < M$ such that at least one of the following two conditions hold:

- $\overline{k}_{a,m}(\tilde{x}_1) \neq \overline{k}_{a,m}(\tilde{x}_2)$

- $\underline{k}_{a,m}(\tilde{x}_2) \neq \underline{k}_{a,m}(\tilde{x}_2)$

This however, is possible only for a finite number of points, while the interval $[x_1, x_2]$ contains infinitely many optimal points. Therefore, there exists a unique solution $\eta_{i,j}^*(\boldsymbol{\omega}) \in$ $\mathrm{argmin}_{\eta \in \mathbb{R}} g_{i,j}(\boldsymbol{\omega}, \boldsymbol{\mu}, \eta)$, and, furthermore, it is a solution of Equation (29), thus concluding the proof. $\square$

Given this result, we continue by providing a result on how to compute the derivative of $f_{i,j}(\boldsymbol{\omega}, \boldsymbol{\mu})$ whenever $f_{i,j}(\boldsymbol{\omega}, \boldsymbol{\mu})$ is given by Equation (7).

**Lemma C.3.** *Consider $\boldsymbol{\mu} \in \Theta^{KM}$ and $\boldsymbol{\omega} \in \Delta_{K \times M}$ such that $f_{i,j}(\boldsymbol{\omega}, \boldsymbol{\mu}) > 0$. Furthermore, suppose that $f_{i,j}(\boldsymbol{\omega}, \boldsymbol{\mu})$ is given by Equation (7). Then, for all $a \in \{i, j\}$ and all $m \in [M]$:*

$$\frac{\partial f_{i,j}(\boldsymbol{\omega}, \boldsymbol{\mu})}{\partial \omega_{a,m}} = \frac{d^+(\mu_{a,m}, \eta_{i,j}^* + \xi_m) + d^-(\mu_{i,m}, \eta_{i,j}^* - \xi_m)}{\lambda_m}.$$

*Proof.* First of all, we notice that, since $f_{i,j}(\boldsymbol{\omega}, \boldsymbol{\mu}) > 0$ holds, and since $f_{i,j}(\boldsymbol{\omega}, \boldsymbol{\mu})$ is expressed as in Equation (7), then, thanks to Lemma 4.4, we know that $\eta_{i,j}^*(\boldsymbol{\omega})$ is the unique optimum of the Equation (7). In the rest of this proof, we will explicit the relationship between $f_{i,j}$ and $\eta_{i,j}^*(\boldsymbol{\omega})$ by writing $f_{i,j}(\boldsymbol{\omega}, \boldsymbol{\mu}, \eta_{i,j}^*(\boldsymbol{\omega}))$. At this point, fix $a \in \{i, j\}$ and $m \in [M]$. Then, it is easy to verify from Equation (10) that both the right and left derivative of $\eta_{i,j}^*(\boldsymbol{\omega})$ w.r.t. $\omega_{a,m}$ exists. Suppose for a

moment that they are equal, then we have that $\frac{\partial \eta_{i,j}^*(\boldsymbol{\omega})}{\partial \omega_{a,m}}$ exists and it continuous. Therefore, we obtain

$$
\frac{\partial}{\partial \omega_{a,m}} f_{i,j}(\boldsymbol{\omega}, \boldsymbol{\mu}, \eta_{i,j}^*(\boldsymbol{\omega})) = \frac{\partial f_{i,j}}{\partial \omega_{a,m}}(\boldsymbol{\omega}, \boldsymbol{\mu}, \eta_{i,j}^*(\boldsymbol{\omega})) + \frac{\partial f_{i,j}}{\partial \eta_{i,j}^*(\boldsymbol{\omega})}(\boldsymbol{\omega}, \boldsymbol{\mu}, \eta_{i,j}^*(\boldsymbol{\omega})) \frac{\partial \eta_{i,j}^*(\boldsymbol{\omega})}{\partial \omega_{a,m}}
$$

$$
= \frac{\partial f_{i,j}}{\partial \omega_{a,m}}(\boldsymbol{\omega}, \boldsymbol{\mu}, \eta_{i,j}^*(\boldsymbol{\omega}))
$$

$$
= \frac{d^+(\mu_{a,m}, \eta_{i,j}^* + \xi_m) + d^-(\mu_{a,m}, \eta_{i,j}^* - \xi_m)}{\lambda_m},
$$

where in the second step we have used that $\frac{\partial f_{i,j}}{\partial \eta_{i,j}^*(\boldsymbol{\omega})}(\boldsymbol{\omega}, \boldsymbol{\mu}, \eta_{i,j}^*(\boldsymbol{\omega})) = 0$ since $\eta_{i,j}^*(\boldsymbol{\omega})$ is a minimizer of Equation (7).

Similarly, whenever, the right and the left derivatives of $\eta_{i,j}^*(\boldsymbol{\omega})$ are different[12], we can follow similar arguments, but analyzing left and right derivatives, and we will obtain an identical result. Indeed, this does not introduce discontinuity issue in the derivatives of $f_{i,j}$ thanks to the fact that $\eta_{i,j}^*(\boldsymbol{\omega})$ is a minimizer of Equation (7). $\qquad \square$

At this point, it remains to analyze in more detail the case in which we have that $f_{i,j}(\boldsymbol{\omega})$ is expressed as in Equation (6).

**Lemma C.4.** *Consider $\boldsymbol{\mu} \in \Theta^{KM}$ and $\boldsymbol{\omega} \in \Delta_{K \times M}$ such that $f_{i,j}(\boldsymbol{\omega}, \boldsymbol{\mu}) > 0$ holds. Furthermore, suppose that $\psi_j^* > \psi_i^*$. Then, for each $a \in \{i, j\}$, there exists a unique minimizer $\psi_a^*$ of Equation (6) which is the unique solution of the following equation of $\psi$:*

$$
\psi = \frac{\sum_{m=1}^M \frac{\omega_{a,m}}{\lambda_m} \left( \overline{k}_{a,m}(\psi) \frac{\mu_{a,m} + \xi_m}{v(\psi - \xi_m)} + \underline{k}_{a,m}(\psi) \frac{\mu_{a,m} - \xi_m}{v(\psi + \xi_m)} \right)}{\left( \sum_{m=1}^M \frac{\omega_{a,m}}{\lambda_m} \left( \overline{k}_{a,m}(\psi) \frac{1}{v(\psi - \xi_m)} + \underline{k}_{a,m}(\psi) \frac{1}{v(\psi + \xi_m)} \right) \right)}. \tag{30}
$$

*Proof.* The proof follows by noticing that, for each $a \in \{i, j\}$, the optimization problem in Equation (6) is an unconstrained convex optimization problem in $\psi$. Taking the derivative and setting it equal to 0 yields the desired result. $\qquad \square$

At this point, we proceed by showing how to compute the partial derivatives of $f_{i,j}(\boldsymbol{\omega})$ whenever it is expressed as in Equation (6).

**Lemma C.5.** *Consider $\boldsymbol{\mu} \in \Theta^{KM}$ and $\boldsymbol{\omega} \in \Delta_{K \times M}$ such that $f_{i,j}(\boldsymbol{\omega}, \boldsymbol{\mu}) > 0$. Furthermore, suppose that $\psi_j^* > \psi_i^*$. Then, for all $m \in [M]$ it holds that:*

$$
\frac{\partial f_{i,j}(\boldsymbol{\omega}, \boldsymbol{\mu})}{\partial \omega_{a,m}} = \frac{d^+(\mu_{a,m}, \psi_a^* + \xi_m) + d^-(\mu_{a,m}, \psi_a^* - \xi_m)}{\lambda_m}.
$$

*Proof.* The proof is a straightforward adaptation of the proof of Lemma C.3. $\qquad \square$

Finally, we are now ready to prove our result on the sub-gradient of $F(\boldsymbol{\omega}, \boldsymbol{\mu})$.

**Theorem 4.3.** *Consider $\boldsymbol{\mu} \in \Theta^{KM}$ and $\boldsymbol{\omega} \in \Delta_{K \times M}$ such that $F(\boldsymbol{\omega}, \boldsymbol{\mu}) > 0$ holds. Let $(i, a) \in [K]^2$ be a pair of arms that attains the max-min value in Equation (2). Then a sub-gradient $\nabla F(\boldsymbol{\omega}, \boldsymbol{\mu})$ of $F(\boldsymbol{\omega}, \boldsymbol{\mu})$ w.r.t. to $\boldsymbol{\omega}$ is given by one of the two following expressions: for $j \in \{a, i\}$ and $m \in [M]$,*

$$
\nabla F(\boldsymbol{\omega}, \boldsymbol{\mu})_{j,m} = \frac{d^+(\mu_{j,m}, \eta_{i,a}^* - \xi_m) + d^-(\mu_{j,m}, \eta_{i,a}^* + \xi_m)}{\lambda_m} \quad \text{if } \psi_i^* \geq \psi_a^*, \tag{8}
$$

$$
\nabla F(\boldsymbol{\omega}, \boldsymbol{\mu})_{j,m} = \frac{d^+(\mu_{j,m}, \psi_j^* - \xi_m) + d^-(\mu_{j,m}, \psi_j^* + \xi_m)}{\lambda_m} \quad \text{otherwise.} \tag{9}
$$

*That sub-gradient $\nabla F(\boldsymbol{\omega}, \boldsymbol{\mu})$ is 0 in all the remaining $KM - 2M$ dimensions.*

*Proof.* The proof follows by the definition of $F(\boldsymbol{\omega}, \boldsymbol{\mu})$ together with Lemma 4.2, Lemma C.3, and Lemma C.5. $\qquad \square$

---

[12]This can happen, for instance, whenever $\eta_{i,j}^*(\boldsymbol{\omega}) = \mu_{a,m} \pm \xi_m$.

We now show a sufficient condition for $F(\boldsymbol{\omega}, \boldsymbol{\mu}) > 0$ to hold when $\boldsymbol{\mu} \in \Theta^{KM}$.

**Lemma C.6.** *Consider $\boldsymbol{\mu} \in \Theta^{KM}$ such that there exists $\star$ for which $\mu_{\star,M} > \max_{a \neq \star} \mu_{a,M}$. Furthermore, consider $\boldsymbol{\omega} \in \Delta_{K \times M}$ such that $\omega_{i,M} > 0$ holds for all $i \in [K]$. Then, we have that $F(\boldsymbol{\omega}, \boldsymbol{\mu}) > 0$.*

*Proof.* From the definition of $F$, and the definition of $\star$, we have that:

$$F(\boldsymbol{\omega}, \boldsymbol{\mu}) \geq \min_{a \neq \star} \inf_{\substack{\theta_a \in \text{MF}, \ \theta_\star \in \text{MF} \\ \theta_{a,M} \geq \theta_{\star,M}}} \sum_{j \in \{\star, a\}} \sum_{m \in [M]} \omega_{j,m} \frac{d(\mu_{j,m}, \theta_{j,m})}{\lambda_m}$$

$$\geq \min_{a \neq \star} \inf_{y \geq x} \omega_{\star,M} \frac{d(\mu_{\star,M}, x)}{\lambda_M} + \omega_{a,M} \frac{d(\mu_{a,M}, y)}{\lambda_M}$$

$$= \min_{a \neq \star} \inf_{\eta \in [\mu_{a,M}, \mu_{\star,M}]} \omega_{\star,M} \frac{d(\mu_{\star,M}, \eta)}{\lambda_M} + \omega_{a,M} \frac{d(\mu_{a,M}, \eta)}{\lambda_M}$$

$$> 0,$$

where in the last step we have used the fact that $\mu_{\star,M} > \mu_{i,M}$ for all $i \neq \star$, together with $\omega_{i,M} > 0$ for all $i \in [K]$. $\square$

Furthermore, we show that the sequence of weights generated by Algorithm 1 satisfy $\omega_{i,M} > 0$ for all $i \in [K]$

**Lemma C.7.** *The sequence of weights $\{\tilde{\boldsymbol{\omega}}(t)\}_t$ satisfy $\omega_{i,M}(t) > 0$ for all $i \in [K]$ and for all $t$.*

*Proof.* We begin by recalling the definition of $\tilde{\boldsymbol{\omega}}(t)$:

$$\tilde{\boldsymbol{\omega}}(t+1) \in \underset{\boldsymbol{\omega} \in \Delta_{K \times M}}{\arg\max} \ \alpha_{t+1} \sum_{s=KM}^{t} \boldsymbol{\omega} \cdot \text{Clip}_s \left( \nabla F(\tilde{\boldsymbol{\omega}}(s), \hat{\boldsymbol{\mu}}(s)) - \text{kl}(\boldsymbol{\omega}, \overline{\boldsymbol{\omega}}) \right)$$

From this definition, thanks to the property of kl, we have that $\tilde{\omega}_{a,m}(t) > 0$ for all $a \in [K]$, and all $m \in [M]$. $\square$

*Remark C.8.* It follows by combining Lemma C.6 and Lemma C.7, that $F(\boldsymbol{\omega}(t), \hat{\boldsymbol{\mu}}(t)) = 0$ might happen only when there are multiple best arms at fidelity $M$. Whenever this condition is encountered, it is possible to project the bandit model $\hat{\boldsymbol{\mu}}(t)$ to have a unique optimal arm (e.g., by adding a small $\epsilon > 0$ to one of the optimal arms). When looking at the proof of Theorem 4.1, we can see that this does not impact its theoretical guarantees as (i) on the good event $\mathcal{E}_T$ this does not happen, and, Lemma C.17 holds unchanged.

### C.2 Smoothness of $F(\omega, \mu)$

**Lemma C.9.** *For any set $S \subseteq \Theta^{KM}$ and any subset of arms $A \subseteq [K]$, the function $(\boldsymbol{\omega}, \boldsymbol{\mu}) \mapsto \inf_{\theta \in S} \sum_{a \in A, m \in [M]} \omega_{a,m} \frac{d(\mu_{a,m}, \theta_{a,m})}{\lambda_m}$ is jointly continuous on $\Delta_{K \times M} \times \Theta^{KM}$.*

*Proof.* We apply (a trivial generalization of) Lemma 27 of [3]. The lemma in that paper is stated for a set of alternative models, but the proof actually works for any set $S$. Likewise, it is stated for the case of $\lambda_m = 1$, but since it works for an arbitrary Bregman divergence $d$ it applies to a rescaled version as well. To deal with the restriction to a subset of arms $A$ instead of all arms, we can view the function as a function of $(\boldsymbol{\omega}_{A \times [M]}, \boldsymbol{\mu}_A)$, where we restrict the vectors to the arms in $A$, and continuity of the original function is equivalent to continuity of the restricted version. $\square$

**Lemma C.10.** *The function $(\boldsymbol{\omega}, \boldsymbol{\mu}) \mapsto F(\boldsymbol{\omega}, \boldsymbol{\mu})$ is jointly continuous on $\Delta_{K \times M} \times \Theta^{KM}$.*

*Proof.* By definition, $F(\boldsymbol{\omega}, \boldsymbol{\mu}) = \max_i \min_{a \neq i} f_{i,a}(\boldsymbol{\omega}, \boldsymbol{\mu})$ with $f_{i,a}(\boldsymbol{\omega}, \boldsymbol{\mu}) = \inf_{\theta \in S_{i,a}} \sum_{a \in A_{i,a}, m \in [M]} \omega_{a,m} \frac{d(\mu_{a,m}, \theta_{a,m})}{\lambda_m}$ for $S_{i,a} = \{\theta \in \mathcal{M}_{MF} \mid \theta_{a,M} \geq \theta_{i,M}\}$ and $A_{i,a} = \{i, a\}$. Since a minimum of finitely many continuous functions is continuous and likewise for the maximum, it suffices to show that each $f_{i,a}$ is jointly continuous. This is true by Lemma C.9. $\square$

**Corollary C.11.** *Let $C \subseteq \Theta^{KM}$ be a compact set. Then $F$ is uniformly continuous on $\triangle_{K \times M} \times C$.*

*Proof.* The set $\triangle_{K \times M} \times C$ is compact and $F$ is continuous, hence it is uniformly continuous on that set. $\qquad\square$

**Lemma C.12.** *Let $\boldsymbol{\mu} \in \Theta^{KM}$. For all $\varepsilon > 0$, there exists $\kappa_\epsilon > 0$ such that for all $\boldsymbol{\omega} \in \triangle_{K \times M}$ and all $\mu' \in \Theta^{KM}$,*

$$\left\| \boldsymbol{\mu} - \boldsymbol{\mu}' \right\|_\infty \leq \kappa_\epsilon \implies \left| F(\boldsymbol{\omega}, \boldsymbol{\mu}') - F(\boldsymbol{\omega}, \boldsymbol{\mu}) \right| \leq \epsilon.$$

*Proof.* Take any compact ball $\mathcal{B}(\boldsymbol{\mu}, \kappa)$ for the norm $\| \cdot \|_\infty$ with $\kappa > 0$ centered at $\boldsymbol{\mu}$. Then $F$ is uniformly continuous on $\triangle_{K \times M} \times \mathcal{B}(\boldsymbol{\mu}, \kappa)$ by Corollary C.11. This means that for any $\varepsilon > 0$, there exists $\kappa'_\varepsilon > 0$ such that for all $(\boldsymbol{\omega}', \boldsymbol{\mu}') \in \triangle_{K \times M} \times \mathcal{B}(\boldsymbol{\mu}, \kappa)$,

$$\|\boldsymbol{\omega} - \boldsymbol{\omega}'\|_\infty \leq \kappa'_\varepsilon \ \wedge \ \|\boldsymbol{\mu} - \boldsymbol{\mu}'\|_\infty \leq \kappa'_\varepsilon \implies |F(\boldsymbol{\omega}', \boldsymbol{\mu}') - F(\boldsymbol{\omega}, \boldsymbol{\mu})| \leq \varepsilon \, .$$

We can take $\kappa_\varepsilon = \min\{\kappa, \kappa'_\varepsilon\}$ to remove the condition $\boldsymbol{\mu}' \in \mathcal{B}(\boldsymbol{\mu}, \kappa)$. The result of the Lemma is this for the special case $\boldsymbol{\omega}' = \boldsymbol{\omega}$. $\qquad\square$

## C.3 Correctness

In the following, we propose an analysis on the correctness which is based on the concentration results provided in [22]. We notice that these results are based on Gaussian distributions. Nevertheless, at the cost of a more involved notation, it is possible to extend all the results of this work for canonical exponential families using, e.g., Theorem 7 in [19].

At this point, let us consider the following value of $\beta_{t,\delta}$:

$$\beta_{t,\delta} = \log\left(\frac{K}{\delta}\right) + 2M \log\left(4 \log\left(\frac{K}{\delta}\right) + 1\right) + 12M \log\left(\log(t) + 3\right) + 2M\tilde{C}, \quad (31)$$

where $\tilde{C}$ is a universal constant (see Proposition 1 in [22]). Then, we can show the following result.

**Proposition C.13.** *Let $\delta > 0$, then it holds that $\mathbb{P}_\mu(\hat{a}_{\tau_\delta} \neq *) \leq \delta$.*

*Proof.* With probabilistic arguments we have that:

$$\mathbb{P}_{\boldsymbol{\mu}}(\hat{a}_{\tau_\delta} \neq \star) \leq \mathbb{P}_{\boldsymbol{\mu}}\left(\exists t \geq KM, \ \exists i \neq \star, \ \min_{j \neq i} f_{i,j}(\boldsymbol{C}(t), \hat{\boldsymbol{\mu}}(t)) \geq \beta_{t,\delta}\right)$$

$$\leq \sum_{i \neq \star} \mathbb{P}_{\boldsymbol{\mu}}\left(\exists t \geq KM, \ \min_{j \neq i} f_{i,j}(\boldsymbol{C}(t), \hat{\boldsymbol{\mu}}(t)) \geq \beta_{t,\delta}\right)$$

$$\leq \sum_{i \neq \star} \mathbb{P}_{\boldsymbol{\mu}}\left(\exists t \geq KM, \ f_{i,\star}(\boldsymbol{C}(t), \hat{\boldsymbol{\mu}}(t)) \geq \beta_{t,\delta}\right)$$

$$\leq \sum_{i \neq \star} \mathbb{P}_{\boldsymbol{\mu}}\left(\exists t \geq KM, \ \sum_{k \in \{i,\star\}} \sum_{m \in [M]} N_{k,m} d(\hat{\mu}_{a,m}(t), \mu_{a,m}) \geq \beta_{t,\delta}\right)$$

$$\leq \delta,$$

where in fourth step we have used the definition of $f_{i,\star}$, and in the last one Proposition 1 in [22] together with a union bound on $K$. $\qquad\square$

## C.4 Auxiliary lemmas

This section contains auxiliary lemmas that will be used in the analysis of Algorithm 1.

**Lemma C.14.** *For all $a \in [K]$, $m \in [M]$, and for all $t \geq 1$, it holds that $N_{a,m}(t) \geq \frac{\sqrt{t}}{4KM} - \ln(KM)$. Furthermore, it holds that:*

$$\left\| \sum_{s=0}^{t} \tilde{\boldsymbol{\pi}}(s) - N(t) \right\|_\infty \leq 2 \ln(KM) \sqrt{t}. \quad (32)$$

*Proof.* This lemma is a simple combination of Lemma 3 in [22] with the tracking result of [5]. Using algebraic manipulations, we have that:

$$
\begin{aligned}
N_{a,m}(t) &\geq \sum_{s=1}^{t} \pi'_{a,m}(s) - \left| N_{a,m}(t) - \sum_{s=1}^{t} \pi'_{a,m}(s) \right| \\
&\geq \sum_{s=1}^{t} \pi'_{a,m}(s) - \ln(KM) \\
&\geq \sum_{s=1}^{t} \frac{\gamma_s}{KM} - \ln(KM) \\
&= \frac{1}{KM} \sum_{s=1}^{t} \frac{1}{4\sqrt{s}} - \ln(KM) \\
&\geq \frac{\sqrt{t}}{4KM} - \ln(KM),
\end{aligned}
$$

where, in the second step we have used Theorem 6 in [5], together with the fact that $\ln(KM) \geq \ln(4) \geq 1$.

For the second part of the proof, we have that:

$$
\begin{aligned}
\left| \sum_{s=1}^{t} \tilde{\pi}_{a,m}(s) - N_{a,m}(t) \right| &\leq \left| \sum_{s=1}^{t} \pi'_{a,m}(s) - N_{a,m}(t) \right| + 2 \sum_{s=1}^{t} \gamma_s \\
&\leq \ln(KM) + \sqrt{t} \\
&\leq 2\ln(KM)\sqrt{t}.
\end{aligned}
$$

$\square$

**Lemma C.15.** *Consider $\epsilon > 0$ and $B \in \mathbb{R}$ such that $C^*(\boldsymbol{\mu})^{-1} - B - \epsilon > 0$. Then, there exists a constant $C_\epsilon$ such that, for*

$$
\sum_{a,m} C_{a,m}(T) \geq \max \left\{ \lambda_M C_\epsilon, \frac{\log\left(\frac{K}{\delta}\right) + 2M \log\left(4\log\left(\frac{K}{\delta}\right) + 1\right)}{C^*(\boldsymbol{\mu})^{-1} - B - \epsilon} \right\} := C_0(\epsilon, \delta), \quad (33)
$$

*it holds that:*

$$
C^*(\boldsymbol{\mu})^{-1} - B \geq \frac{\beta_{T,\delta}}{\sum_{a,m} C_{a,m}(T)}. \quad (34)
$$

*Proof.* Let $C_\epsilon$ be a constant that depends on $\epsilon$ such that, for $T \geq C_\epsilon$ it holds that:

$$
\frac{12M \log(\log(T) + 3) + 2M\tilde{C}}{\lambda_{\min}} \leq \epsilon T.
$$

Then, for $\sum_{a,m} C_{a,m}(T) \geq C_0(\epsilon, \delta)$, we have that:

$$
\begin{aligned}
\frac{\beta_{T,\delta}}{\sum_{a,m} C_{a,m}(T)} &= \frac{\log\left(\frac{K}{\delta}\right) + 2M \log\left(4\log\left(\frac{K}{\delta}\right) + 1\right) + 12M \log(\log(T) + 3) + 2M\tilde{C}}{\sum_{a,m} C_{a,m}(T)} \\
&\leq \frac{\log\left(\frac{K}{\delta}\right) + 2M \log\left(4\log\left(\frac{K}{\delta}\right) + 1\right)}{\sum_{a,m} C_{a,m}(T)} + \epsilon \\
&\leq C^*(\boldsymbol{\mu})^{-1} - B,
\end{aligned}
$$

which concludes the proof. $\square$

## C.5 Proof of Theorem 4.1

Before diving into the proof of Theorem 4.1, we introduce some additional notation. We denote with $\mathcal{B}_\infty(x, \kappa)$ the ball of radius $\kappa$ centered at $x$. Then, for all $T$, and $\epsilon > 0$, we introduce the following event:

$$\mathcal{E}_\epsilon(T) = \bigcap_{t \geq h(T)} \{\hat{\boldsymbol{\mu}}(t) \in \mathcal{B}_\infty(\boldsymbol{\mu}, \kappa_\epsilon)\},$$

where $h(T) \approx T^{1/4}$. At this point, we present our result. Furthermore, we denote with $\boldsymbol{\omega}(t)$ the vector of empirical cost proportions, namely, for all $(a, m)$, $\omega_{a,m}(t) = \frac{C_{a,m}(t)}{\sum_{i \in [K]} \sum_{j \in [M]} C_{i,j}(t)}$.

First of all, we introduce an initial result that controls the expectation of the stopping cost.

**Lemma C.16.** *Consider $B$ such that $C^*(\boldsymbol{\mu})^{-1} - B - \epsilon > 0$, and suppose that there exists a constant $T_\epsilon$ such that, for all $T \geq T_\epsilon$, it holds that $F(\boldsymbol{\omega}(T), \hat{\boldsymbol{\mu}}(T)) \geq F(\boldsymbol{\omega}^*(\boldsymbol{\mu})) - B$ on the good event $\mathcal{E}_\epsilon(T)$. Then, it holds that:*

$$\mathbb{E}_{\boldsymbol{\mu}}[c_{\tau_\delta}] \leq \lambda_M T_\epsilon + C_0(\epsilon, \delta) + 1 + \sum_{t=0}^{+\infty} \mathbb{P}_{\boldsymbol{\mu}}(\mathcal{E}(T)^c). \tag{35}$$

*Proof.* Using probabilistic arguments, we have that:

$$\begin{aligned}
\mathbb{E}_{\boldsymbol{\mu}}[c_{\tau_\delta}] &= \int_0^{+\infty} \mathbb{P}_{\boldsymbol{\mu}}(c_{\tau_\delta} > x) dx \\
&\leq \lambda_M T_\epsilon + C_0(\epsilon, \delta) + \int_{\lambda_M T_\epsilon + C_0(\epsilon, \delta)} \mathbb{P}_{\boldsymbol{\mu}}(c_{\tau_\delta} > x, c_{\tau_\delta} \geq C_0(\epsilon, \delta)) dx \\
&\leq \lambda_M T_\epsilon + C_0(\epsilon, \delta) + \int_{\lambda_M T_\epsilon + C_0(\epsilon, \delta)} \mathbb{P}_{\boldsymbol{\mu}}\left(\tau_\delta > \frac{x}{\lambda_M}, c_{\tau_\delta} \geq C_0(\epsilon, \delta)\right) dx \\
&\leq \lambda_M T_\epsilon + C_0(\epsilon, \delta) + 1 + \sum_{T = \lfloor T_\epsilon + \frac{C_0(\epsilon, \delta)}{\lambda_M} \rfloor}^{+\infty} \mathbb{P}_{\boldsymbol{\mu}}(\tau_\delta > T, c_{\tau_\delta} \geq C_0(\epsilon, \delta)) \\
&\leq \lambda_M T_\epsilon + C_0(\epsilon, \delta) + 1 + \sum_{T=0}^{+\infty} \mathbb{P}_{\boldsymbol{\mu}}(\mathcal{E}(T)^c),
\end{aligned}$$

where (i) in the second inequality, we have upper bounded $c_{\tau_\delta} \leq \lambda_M \tau_\delta$, (ii) in the third inequality, we have used the fact that $\tau_\delta$ is an integer variable, and (iii) in the last inequality we have used that for all $T \geq T_\epsilon$ such that $\sum_{a,m} C_{a,m}(T) \geq C_0(\epsilon, \delta)$, then we have that $\mathcal{E}(T) \subseteq \{\tau_\delta < T\}$. Indeed, combining Lemma C.15 with the definition $T_\epsilon$, we obtain that for all $T \geq T_\epsilon$ such that $\sum_{a,m} C_{a,m}(T) \geq C_0(\epsilon, \delta)$, then we have that $\mathcal{E}(T) \subseteq \{\tau_\delta < T\}$. This last step is direct by noticing that $F(\boldsymbol{\omega}(T), \hat{\boldsymbol{\mu}}(T)) \geq \frac{\beta_{T,\delta}}{\sum_{a,m} C_{a,m}(T)}$ implies stopping. $\square$

Lemma C.16 shows how to upper-bound the expected cost complexity. Notice that this result requires different arguments w.r.t. the usual ones that appears while controlling the expected sample complexity (see, e.g., [8]).

Then, we report a basic property of the sub-gradient ascent routing that is employed in our algorithm. Before doing that, we recall that, on the good-event $\mathcal{E}_T$, it holds that there exists a constant $L$, that depends on $\boldsymbol{\mu}$, such that the empirical sub-gradients are uniformly-bounded by $L$.

**Lemma C.17.** *Let $\tilde{c}(t) = \sum_{a,m} \lambda_m \tilde{\pi}_{a,m}(s)$. Define $C_r := \log(KM) + KMG + 4(L\lambda_M)^2 + 2G^2$, and consider the sequence of weights $\{\tilde{\boldsymbol{\omega}}(t)\}_t$ generated by Algorithm 1. Then, on the good event $\mathcal{E}_\epsilon(T)$ it holds that:*

$$\sum_{t=h(t)}^{T} \tilde{c}(t) \nabla F(\boldsymbol{\omega}^*, \hat{\boldsymbol{\mu}}(t)) \cdot (\boldsymbol{\omega}^* - \tilde{\boldsymbol{\omega}}(t)) \leq C_r \sqrt{T}.$$

*Proof.* The proof is identical to the one of Proposition 3 in [22]. The only difference is that, in our case, we need to multiply the scale of the sub-gradient $L$ by $\lambda_M$, to the additional presence of $\tilde{c}(t)$ in the sequence of gains that we use in our sub-gradient ascent algorithm. $\qquad\square$

Finally, we show that there exists an additional problem dependent constant $C_{\boldsymbol{\mu}}$ that will be useful in performing some upper-bound reasoning in the proof of the final result.

**Lemma C.18.** *Consider the following quantity:*

$$\frac{\min_{a\neq\star}\inf_{\boldsymbol{\theta}_a,\boldsymbol{\theta}_\star\in\mathrm{MF}}\sum_{s=1}^{T}\sum_{i\in\{\star,a\}}\sum_m\tilde{\pi}_{i,m}(s)d(\mu_{i,m},\theta_{i,m})}{\sum_{a,m}C_{a,m}(T)}.$$

*There exists a problem dependent constant $C_{\boldsymbol{\mu}}$ such that the previous equation can be upper bounded by:*

$$F(\boldsymbol{\omega}(T),\boldsymbol{\mu})+\frac{4\ln(KM)M\lambda_M C_{\boldsymbol{\mu}}}{\lambda_{\min}\sqrt{T}}.$$

*Proof.* Let us begin by analyzing $F(\boldsymbol{\omega}(T),\boldsymbol{\mu})$. Fix $\bar{a}$ such that $F(\boldsymbol{\omega}(T),\boldsymbol{\mu})=f_{\star,\bar{a}}(\boldsymbol{\omega}(T),\boldsymbol{\mu})$. Moreover, consider $\boldsymbol{\theta}_\star,\boldsymbol{\theta}_{\bar{a}}\in\arg\min\sum_{i\in\{\star,\bar{a}\}}\sum_m\omega_{a,m}(T)\frac{d(\mu_{a,m},\theta_{a,m})}{\lambda_m}$. Then, consider the following difference:

$$H:=\frac{\min_{a\neq\star}\inf_{\boldsymbol{\theta}_a,\boldsymbol{\theta}_\star\in\mathrm{MF}}\sum_{s=1}^{T}\sum_{i\in\{\star,a\}}\sum_m\tilde{\pi}_{i,m}(s)d(\mu_{i,m},\theta_{i,m})}{\sum_{a,m}C_{a,m}(T)}-F(\boldsymbol{\omega}(T),\boldsymbol{\mu}).$$

Then, the previous Equation can be upper bounded by:

$$
\begin{aligned}
H &\leq \frac{\sum_{i\in\{\star,\bar{a}\}}\sum_m\left(\sum_{s=1}^{T}\tilde{\pi}_{i,m}(s)-N_{i,m}(T)\right)d(\mu_{i,m},\theta_{i,m}^*)}{\sum_{a,m}C_{a,m}(T)}\\
&\leq 2\ln(KM)\sqrt{T}\frac{\sum_{i\in\{\star,\bar{a}\}}\sum_m d(\mu_{i,m},\theta_{i,m}^*)}{\sum_{a,m}C_{a,m}(T)}\\
&\leq \frac{4\ln(KM)M\sqrt{T}C_{\boldsymbol{\mu}}}{\sum_{a,m}C_{a,m}(T)},
\end{aligned}
$$

where in the first step, we have used the definition of $\boldsymbol{\theta}_{\bar{a}}^*,\boldsymbol{\theta}_\star^*$ and the definition of $\boldsymbol{\omega}(T)$, in the second one, we have used Lemma C.14, and in the last one the facts that, thanks to definition $\boldsymbol{\theta}_{\bar{a}}^*,\boldsymbol{\theta}_\star^*$, there exists some problem dependent constant $C_{\boldsymbol{\mu}}$ such that $d(\mu_{i,m},\theta_{i,m}^*)$ is bounded. $\qquad\square$

**Theorem 4.1.** *For any multi-fidelity bandit model $\boldsymbol{\mu}\in\mathcal{M}_{\mathrm{MF}}$, Algorithm 1 using the threshold $\beta_{t,\delta}$ given in (31) is $\delta$-correct and satisfies*

$$\limsup_{\delta\to0}\frac{\mathbb{E}_{\boldsymbol{\mu}}[c_{\tau_\delta}]}{\log(1/\delta)}\leq C^*(\boldsymbol{\mu}). \tag{5}$$

*Proof.* The proof of the $\delta$-correctness is from Proposition C.13.

To prove the optimality, we first proceed by upper bounding the following quantity on the good event $\mathcal{E}_\epsilon(T)$:

$$F(\boldsymbol{\omega}^*,\boldsymbol{\mu})-F(\boldsymbol{\omega}(T),\hat{\boldsymbol{\mu}}(T)). \tag{36}$$

Define, for brevity, $\widetilde{T} := T - h(T) + 1$ and $\tilde{c}(s) := \sum_{a,m} \lambda_m \tilde{\pi}_{a,m}(s)$. Then, we start by analyzing $F(\boldsymbol{\omega}^*, \boldsymbol{\mu})$. On $\mathcal{E}_\epsilon(T)$ we have that:

$$F(\boldsymbol{\omega}^*, \boldsymbol{\mu}) = \frac{\sum_{a,m} C_{a,m}(T)}{\sum_{a,m} C_{a,m}(T)} F(\boldsymbol{\omega}^*, \boldsymbol{\mu}) - \frac{\sum_{s=1}^T \tilde{c}(s)}{\sum_{a,m} C_{a,m}(T)} F(\boldsymbol{\omega}^*, \boldsymbol{\mu}) + \frac{\sum_{s=1}^T \tilde{c}(s)}{\sum_{a,m} C_{a,m}(T)} F(\boldsymbol{\omega}^*, \boldsymbol{\mu})$$

$$\leq \frac{\sum_{s=1}^T \tilde{c}(s)}{\sum_{a,m} C_{a,m}(T)} F(\boldsymbol{\omega}^*, \boldsymbol{\mu}) + \frac{F(\boldsymbol{\omega}^*, \boldsymbol{\mu})}{\sum_{a,m} C_{a,m}(T)} 2\ln(KM)\sqrt{T}$$

$$\leq \frac{\sum_{s=1}^T \tilde{c}(s)}{\sum_{a,m} C_{a,m}(T)} F(\boldsymbol{\omega}^*, \boldsymbol{\mu}) + \frac{2\ln(KM)F(\boldsymbol{\omega}^*, \boldsymbol{\mu})}{\lambda_{\min}\sqrt{T}}$$

$$\leq \frac{\sum_{s=h(T)}^T \tilde{c}(s)}{\sum_{a,m} C_{a,m}(T)} F(\boldsymbol{\omega}^*, \boldsymbol{\mu}) + \frac{h(T)}{\lambda_{\min}T} F(\boldsymbol{\omega}^*, \boldsymbol{\mu}) + \frac{2\ln(KM)F(\boldsymbol{\omega}^*, \boldsymbol{\mu})}{\lambda_{\min}\sqrt{T}}$$

$$\leq \frac{\sum_{s=h(T)}^T \tilde{c}(s)F(\boldsymbol{\omega}^*, \hat{\boldsymbol{\mu}}(t)))}{\sum_{a,m} C_{a,m}(T)} + \frac{\lambda_M \widetilde{T}\epsilon}{\lambda_{\min}T} + \frac{h(T)}{\lambda_{\min}T} F(\boldsymbol{\omega}^*, \boldsymbol{\mu}) + \frac{2\ln(KM)F(\boldsymbol{\omega}^*, \boldsymbol{\mu})}{\lambda_{\min}\sqrt{T}},$$

where in the first inequality we have used Lemma C.14, while in the last step we have used Lemma C.12 together with the event $\mathcal{E}_\epsilon(T)$. At this point, we focus our analysis on $\frac{\sum_{s=h(T)}^T \tilde{c}(s)F(\boldsymbol{\omega}^*, \hat{\boldsymbol{\mu}}(t))}{\sum_{a,m} C_{a,m}(T)}$. Define, for brevity, $g_s = \tilde{c}(s)\nabla F(\tilde{\boldsymbol{\omega}}(s), \hat{\boldsymbol{\mu}}(s))$; then, we have that:

$$\frac{\sum_{s=h(T)}^T \tilde{c}(s)F(\boldsymbol{\omega}^*, \hat{\boldsymbol{\mu}}(s)))}{\sum_{a,m} C_{a,m}(T)} = \frac{\sum_{s=h(T)}^T \tilde{c}(s)\left(F(\boldsymbol{\omega}^*, \hat{\boldsymbol{\mu}}) \pm F(\tilde{\boldsymbol{\omega}}(s), \hat{\boldsymbol{\mu}}(s)))\right)}{\sum_{a,m} C_{a,m}(T)}$$

$$\leq \frac{\sum_{s=h(T)}^T \tilde{c}(s)F(\tilde{\boldsymbol{\omega}}(s), \hat{\boldsymbol{\mu}}(s)))}{\sum_{a,m} C_{a,m}(T)} + \frac{\sum_{s=h(T)}^T g_s \cdot (\boldsymbol{\omega}^* - \tilde{\boldsymbol{\omega}}(s))}{\sum_{a,m} C_{a,m}(T)}$$

$$\leq \frac{\sum_{s=h(T)}^T \tilde{c}(s)F(\tilde{\boldsymbol{\omega}}(s), \hat{\boldsymbol{\mu}}(s)))}{\sum_{a,m} C_{a,m}(T)} + \frac{C_r}{\lambda_{\min}\sqrt{T}}$$

$$\leq \frac{\sum_{s=h(T)}^T \tilde{c}(s)F(\tilde{\boldsymbol{\omega}}(s), \boldsymbol{\mu})}{\sum_{a,m} C_{a,m}(T)} + \frac{C_r}{\lambda_{\min}\sqrt{T}} + \frac{\lambda_M \widetilde{T}\epsilon}{\lambda_{\min}T},$$

where in the first inequality we have used the concavity of $F$, in the second one we have used Lemma C.17, and in the last one Lemma C.12 and the definition of $\mathcal{E}_\epsilon(T)$.

Finally, we have that:

$$\frac{\sum_{s=1}^T \tilde{c}(s)F(\tilde{\boldsymbol{\omega}}(s), \boldsymbol{\mu})}{\sum_{a,m} C_{a,m}(T)} = \frac{\sum_{s=1}^T \min_{a \neq \star} \inf_{\boldsymbol{\theta}_a, \boldsymbol{\theta}_\star \in \mathrm{MF}} \sum_{i \in \{\star, a\}} \sum_m \tilde{\pi}_{i,m}(s)d(\mu_{i,m}, \theta_{i,m})}{\sum_{a,m} C_{a,m}(T)}$$

$$\leq \frac{\min_{a \neq \star} \inf_{\boldsymbol{\theta}_a, \boldsymbol{\theta}_\star \in \mathrm{MF}} \sum_{s=1}^T \sum_{i \in \{\star, a\}} \sum_m \tilde{\pi}_{i,m}(s)d(\mu_{i,m}, \theta_{i,m})}{\sum_{a,m} C_{a,m}(T)}$$

$$\leq F(\boldsymbol{\omega}(T), \boldsymbol{\mu}) + \frac{4\ln(KM)MC_{\boldsymbol{\mu}}}{\lambda_{\min}\sqrt{T}}$$

$$\leq F(\boldsymbol{\omega}(T), \hat{\boldsymbol{\mu}}(T)) + \frac{4\ln(KM)MC_{\boldsymbol{\mu}}}{\lambda_{\min}\sqrt{T}} + \epsilon,$$

where (i) in the first equality we used the definition of $\tilde{c}(s)$, $\tilde{\boldsymbol{\omega}}(s)$ and $F$, in the second one we used Lemma C.18, and in the last one Lemma C.12.

Given this analysis, let us define:

$$B_T := \frac{2\lambda_M \widetilde{T}\epsilon}{\lambda_{\min}T} + \frac{h(T)}{\lambda_{\min}T} F(\boldsymbol{\omega}^*, \boldsymbol{\mu}) + \frac{2\ln(KM)F(\boldsymbol{\omega}^*, \boldsymbol{\mu})}{\lambda_{\min}\sqrt{T}} +$$

$$+ \frac{C_r}{\lambda_{\min}\sqrt{T}} + \frac{4\ln(KM)MC_{\boldsymbol{\mu}}}{\lambda_{\min}\sqrt{T}} + \epsilon.$$

Consider $T$ such that:

$$T \geq \max\left\{\left(\frac{2\ln(KM)F(\boldsymbol{\omega}^*, \boldsymbol{\mu})}{\lambda_{\min}\epsilon}\right)^2, \left(\frac{C_r}{\lambda_{\min}\epsilon}\right)^2, \left(\frac{4\ln(KM)MC_{\boldsymbol{\mu}}}{\lambda_{\min}\epsilon}\right)^2\right\} := T_\epsilon.$$

Then, it holds that:

$$B_T \leq \frac{2\lambda_M\epsilon}{\lambda_{\min}} + 5\epsilon = \bar{B}_\epsilon,$$

and, consequently, we have that

$$F(\boldsymbol{\omega}(T), \hat{\boldsymbol{\mu}}(T)) \geq F(\boldsymbol{\omega}^*, \boldsymbol{\mu}) - \bar{B}_\epsilon.$$

Combining this result with Lemma C.15 and C.16, we obtain:

$$\mathbb{E}[c_{\tau_\delta}] \leq \lambda_M T_\epsilon + C_0(\delta, \epsilon) + 1 + \sum_{t=0}^{+\infty} \mathbb{P}_{\boldsymbol{\mu}}(\mathcal{E}_\epsilon(t)^c).$$

By Lemma C.14 and Lemma 19 in [8], we obtain that:

$$\limsup_{\delta \to 0} \frac{\mathbb{E}[c_{\tau_\delta}]}{\log(1/\delta)} \leq \frac{1}{C^*(\boldsymbol{\mu})^{-1} - \bar{B}_\epsilon}.$$

Letting $\epsilon \to 0$ concludes the proof. $\qquad\square$

## D  Experiment details and additional results

In this section, we provide experimental details and additional results. For the experiments we relied on a server with 100 Intel(R) Xeon(R) Gold 6238R CPU @ 2.20GHz cpus and 256GB of RAM. The time to obtain all the empirical results is less than a day.

This section is structured as follows.

- First, we provide an additional details and results on the experiment presented in Section 5 (Section D.1 and Section D.2).
- Secondly, we provide results on additional $4 \times 5$ multi-fidelity bandits (Section D.3).
- Then, we analyze a typical trick that is used to improve the performance of gradient-based methods, that is using a constant rate against using the learning rate that the theory prescribes. (Section D.4).
- We then present results using very small value of $\delta$ w.r.t. to the one that has been considered in the main text (Section D.5). In particular, we verify that the performance difference amplifies.
- Finally, we discuss the approaches of [31].

### D.1  Further details on the experiments presented in Section 5

**First instance**  We begin by providing further details on the $4 \times 5$ multi-fidelity bandit model that we used in Figure 1 and Figure 2. First, Table 2 reports the $4 \times 5$ bandit model of Figure 1.

**Table 2:** Multi-fidelity bandit model presented in Figure 1.

|         | $\mu_1$ | $\mu_2$ | $\mu_3$ | $\mu_4$ | $\xi$ | $\lambda$ |
|---------|---------|---------|---------|---------|-------|-----------|
| $m = 1$ | 0.9465  | 0.8526  | 0.8162  | 0.9099  | 0.1   | 0.05      |
| $m = 2$ | 0.7727  | 0.8708  | 0.9050  | 1.0594  | 0.08  | 0.1       |
| $m = 3$ | 0.8812  | 0.8515  | 0.8209  | 1.0083  | 0.05  | 0.2       |
| $m = 4$ | 0.8284  | 0.8374  | 0.8353  | 0.9745  | 0.025 | 0.4       |
| $m = 5$ | 0.8494  | 0.8401  | 0.8495  | 0.9856  | 0.0   | 5         |

All arms, both for the bandit model of Figure 1 and 2 are Gaussian distributions with variance $\sigma^2 = 0.1$. The bandit model in Figure 1 has been generated according to a procedure that has been

used to generate MF instances in [25] (see their Appendix D.1). Specifically, first, two $M$-dimensional vectors are specified, which we refer to as $\boldsymbol{a}$ and $\boldsymbol{b}$. Specifically, $\boldsymbol{a}$ and $\boldsymbol{b}$ are such that $a_m \geq a_{m+1}$ and $b_m \geq b_{m+1}$ for all $m \in [M-1]$. Then, we first sample the means of the arm at fidelity $M$[13], and once this is done we sample $\mu_{i,m} \in \left[\mu_{i,M} - a_m - \frac{b}{2}, \mu_{i,M} + a_m + \frac{b}{2}\right]$. Then, $\xi$ is computed as $\xi_m = a_m + \frac{b_m}{2}$. In this sampling procedure, we have used $\boldsymbol{a} = [0.075, 0.06, 0.04, 0.02, 0]$ and $\boldsymbol{b} = [0.05, 0.04, 0.02, 0.01, 0]$.

**Second instance** We now recall the $5 \times 2$ example of Section 5:

**Table 3:** Multi-fidelity bandit model presented in Figure 2.

|         | $\mu_1$ | $\mu_2$ | $\mu_3$ | $\mu_4$ | $\mu_5$ | $\xi$ | $\lambda$ |
|---------|---------|---------|---------|---------|---------|-------|-----------|
| $m = 1$ | 0.4     | 0.4     | 0.4     | 0.4     | 0.5     | 0.1   | 0.5       |
| $m = 2$ | 0.5     | 0.5     | 0.5     | 0.5     | 0.6     | 0     | 5         |

We prove that, in this instance, the oracle weights are given by $\omega_i^* = [0.09621, 0]$ for all $i \in [4]$, and $\omega_5^* = [0, 0.61516]$ (this number have been rounded to the fourth decimal precision). In order to prove this, we first notice that in the considered domain the optimal fidelity for $i \in [4]$ is $m = 1$. This is direct from the fact that $\mu_{i,m} = \mu_{i,M} - \xi_m$ (see, e.g., Proposition B.5). Furthermore, we recall the expression $f_{5,i}$, for any $i \in [4]$:

$$f_{5,i}(\boldsymbol{\omega}, \boldsymbol{\mu}) = \inf_{\eta \in [\mu_{i,M}, \mu_{5,M}]} \sum_{m \in [M]} \omega_{5,m} \frac{d^-(\mu_{5,m}, \eta + \xi_m)}{\lambda_m} + \omega_{i,m} \frac{d^+(\mu_{i,m}, \eta - \xi_m)}{\lambda_m}.$$

Then, since $\eta_{5,i}^* \in [\mu_{i,M}, \mu_{5,M}]$, $\mu_{i,M} = \mu_{5,m} = \mu_{5,M} - \xi_m$, we have that the optimal fidelity for arm 5 is $m = 2$. At this point, consider the oracle weights $\boldsymbol{\omega}^*$. We notice that, due to the symmetry of the problem, $\omega_{i,1}^*$ is equal for all $i \in [4]$. Then, we can rewrite $f_{5,i}(\boldsymbol{\omega}^*, \boldsymbol{\mu})$ as a function of a single variable, that is:

$$f_{5,i}(\boldsymbol{\omega}, \boldsymbol{\mu}) = \inf_{\eta \in [\mu_{i,M}, \mu_{5,M}]} (1 - 4\omega_{i,1}) \frac{d^-(\mu_{5,2}, \eta)}{\lambda_M} + \omega_{i,1} \frac{d^+(\mu_{i,1}, \eta - \xi_m)}{\lambda_m},$$

and, consequently, we obtain that $C^*(\boldsymbol{\mu})^{-1}$ can be expressed as a convex optimization of a single variable, that is $\omega_{i,1}$. Taking the derivative of $F(\boldsymbol{\omega}, \boldsymbol{\mu})$ w.r.t. $\omega_{i,1}$ we obtain that the following equality should be satisfied at the optimum:

$$4\frac{d(\mu_{1,M}, \eta_{5,i}^*)}{\lambda_M} = \frac{d(\mu_{i,1}, \eta_{5,i}^* - 1)}{\lambda_m}.$$

Solving for $\eta_{5,i}^*$ gives a unique solution in the range $[0.5, 0.6]$, which is 0.539. Then, using Lemma C.3 and solving for $\omega_{i,1}$, we obtain $\omega_{i,1} = 0.09621$, and consequently, $\omega_{5,2} = 0.61516$.

**Thresholds** To conclude, we comment on the thresholds $\beta_{t,\delta}$ used by the algorithms. For the stopping rule in MF-GRAD we used $\beta_{t,\delta} = \log(K/\delta) + M \log(\log(t) + 1)$, which is a simplification of its theoretical value (31) that retains the same scaling in $K$ and $M$ (up to constants). In GRAD, we used $\beta_{t,\delta} = \log(K/\delta) + \log(\log(t) + 1)$, which is a similar simplification of the usual threshold for BAI, which instead of concentrating a sum of $2M$ KL terms (see the proof of Proposition C.13) only requires to concentrate a sum over 2 KL terms. Finally, in the confidence intervals that are used in IISE we have used the confidence bonuses $\sqrt{\frac{2\sigma^2(\log(KM/\delta) + \log(\log(t)))}{N_{a,m}(t)}}$[14], which compared to their original form is replacing some crude union bound over $t$ with a stylized version of the threshold that would follows from using tight time-uniform concentration. These choices were adopted consistently in all the experiments presented in this appendix, and they ensured the $\delta$-correctness requirement in all cases.

---

[13]For this step, we constrained the minimum gap between arms at fidelity $M$ is at least 0.1.

[14]Notice, indeed, that ISEE requires a union bound both on $K$ and $M$.

## D.2 Empirical cost proportions of MF-GRAD

In this section, we analyze the behavior of MF-GRAD by analyzing the evolution of the empirical cost proportions. Specifically, we repeat on the $4 \times 5$ bandit model of Table 2, the same experiment that we presented in Section 5 for the $5 \times 2$ bandit model. Figure 4 reports the result. Interestingly, we highlight how the sparsity pattern emerges also in this domain.

## D.3 Results on additional domains

In this section, we present results on additional $4 \times 5$ multi-fidelity bandit models. Specifically, we generate another random instance according to the procedure of [25], and we report the model in Table 4. Furthermore, we created an additional bandit model where means of some arms are slightly increasing on displaying a stationary trend over fidelity and we report the model in Table 5. In both cases, we considered Gaussian distribution with $\sigma^2 = 0.1$. Empirical results of MF-GRAD, IISE and GRAD for $\delta = 0.01$ can be found in Figure 5 and 6 respectively. As we can appreciate, MF-GRAD maintains the most competitive performance across both domains.

**Table 4:** Additional random multi-fidelity bandit model.

|  | $\mu_1$ | $\mu_2$ | $\mu_3$ | $\mu_4$ | $\xi$ | $\lambda$ |
|---|---|---|---|---|---|---|
| $m = 1$ | 0.6944 | 0.5080 | 0.4153 | 0.3564 | 0.1 | 0.05 |
| $m = 2$ | 0.5634 | 0.3723 | 0.4132 | 0.4570 | 0.08 | 0.1 |
| $m = 3$ | 0.6178 | 0.4322 | 0.3817 | 0.4065 | 0.05 | 0.2 |
| $m = 4$ | 0.6323 | 0.4225 | 0.3838 | 0.3582 | 0.025 | 0.4 |
| $m = 5$ | 0.6171 | 0.4216 | 0.3831 | 0.3783 | 0.0 | 1 |

**Table 5:** Additional multi-fidelity bandit model.

|  | $\mu_1$ | $\mu_2$ | $\mu_3$ | $\mu_4$ | $\xi$ | $\lambda$ |
|---|---|---|---|---|---|---|
| $m = 1$ | 0.41 | 0.35 | 0.51 | 0.41 | 0.1 | 0.1 |
| $m = 2$ | 0.45 | 0.37 | 0.56 | 0.39 | 0.08 | 0.125 |
| $m = 3$ | 0.47 | 0.38 | 0.64 | 0.40 | 0.04 | 0.25 |
| $m = 4$ | 0.48 | 0.36 | 0.62 | 0.42 | 0.02 | 0.5 |
| $m = 5$ | 0.5 | 0.35 | 0.61 | 0.42 | 0.0 | 1 |

## D.4 Improving performance with constant learning rate

As reported in [22], using constant learning rate can improve the identification performance in standard BAI settings. In the following, we analyze the performance difference of MF-GRAD that uses the theoretical learning rate, and MF-GRAD that uses a constant learning rate of $\alpha = 0.25$. We will refer this second version as MF-GRAD-CONST. Figure 7 and 8 reports the performance of the algorithms in the two bandit models of Section 5. As we can see, in both cases, MF-GRAD-CONST outperforms MF-GRAD.

We further investigate this behavior by showing the evolution of the empirical cost proportions of MF-GRAD-CONST during the learning process. Figure 9 and 10 reports the evolution of the empirical costs, over 100000 iterations. Comparing the results with Figure 4 and 9 we can appreciate as MF-GRAD-CONST move away from the initial cost proportions way sooner than MF-GRAD, which explains its superior performance in the moderate regime of $\delta$.

## D.5 Smaller value of $\delta$

Finally, we repeat the experiments that we presented in the previous section using smaller values of $\delta$. Specifically, we consider $\delta = 10^{-10}$. Figure 11 reports the performance of the $4 \times 5$ bandit model of Section 5, Figure 12 reports the performance of the $5 \times 2$ bandit model of Section 5, Figure 13 and 14 reports the performance of the additional bandit models presented in Appendix D.3. As one can notice the performance gap between MF-GRAD and the considered baseline increases.

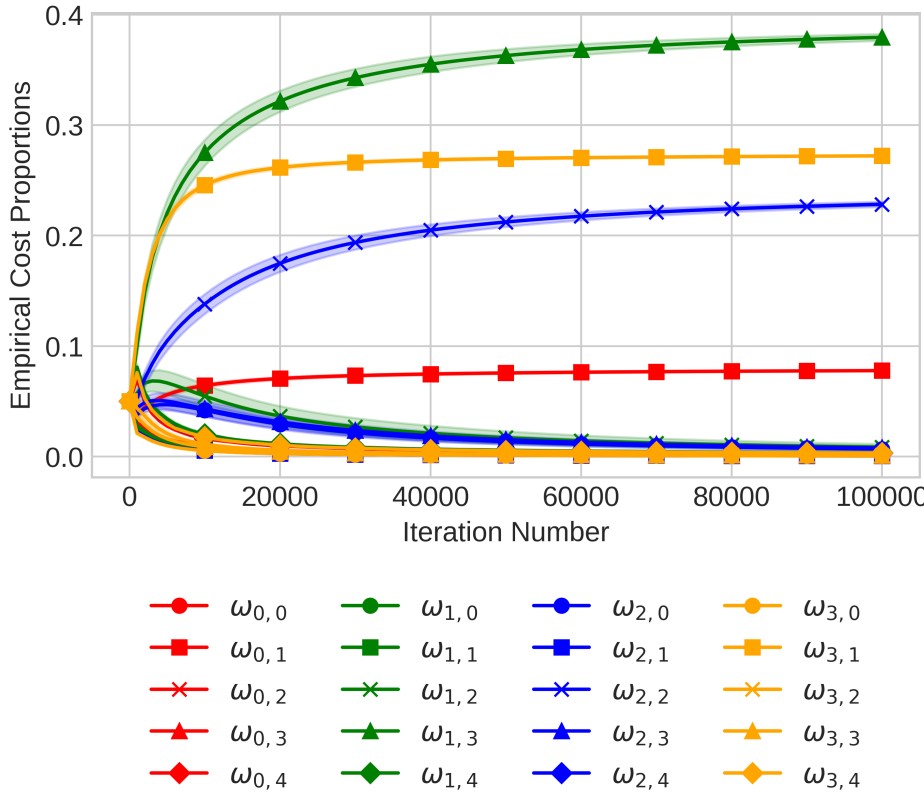

**Figure 4:** Empirical cost proportions of MF-GRAD for 100000 iterations on the $5 \times 2$ bandit model of Section 5. Results are average over 100 runs and shaded area report $95\%$ confidence intervals. Empirical cost proportions of each arm are plotted with the same color. Cost proportions at fidelity $1, 2, 3, 4$ and $5$ are visualized with circle, squared, cross, triangle, and diamond respectively.

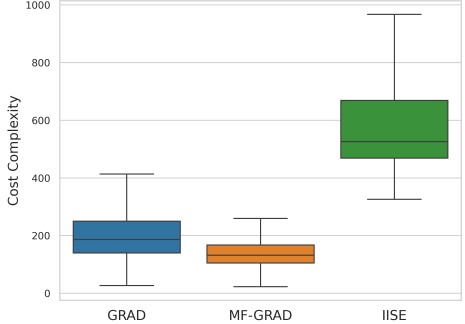

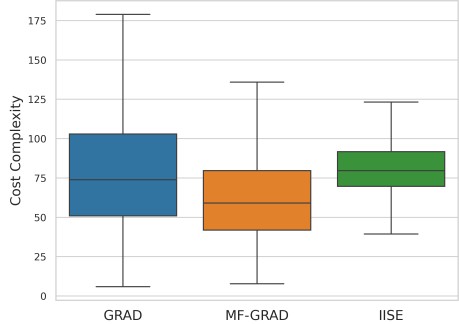

**Figure 5:** Empirical cost complexity for 1000 runs times with $\delta = 0.01$ on the multi-fidelity bandit of Table 4.

**Figure 6:** Empirical cost complexity for 1000 runs times with $\delta = 0.01$ on the multi-fidelity bandit of Table 5.

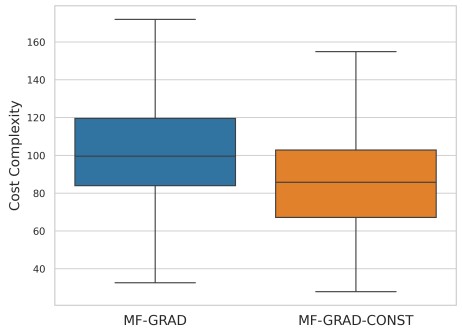
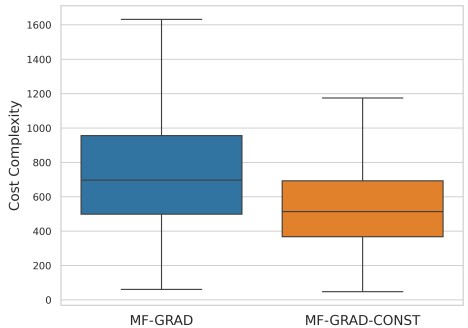

**Figure 7:** Empirical cost complexity for 1000 runs times with $\delta = 0.01$ on the $4 \times 5$ multi-fidelity bandit of Section 5.

**Figure 8:** Empirical cost complexity for 1000 runs times with $\delta = 0.01$ on the $5 \times 2$ multi-fidelity bandit of Section 5.

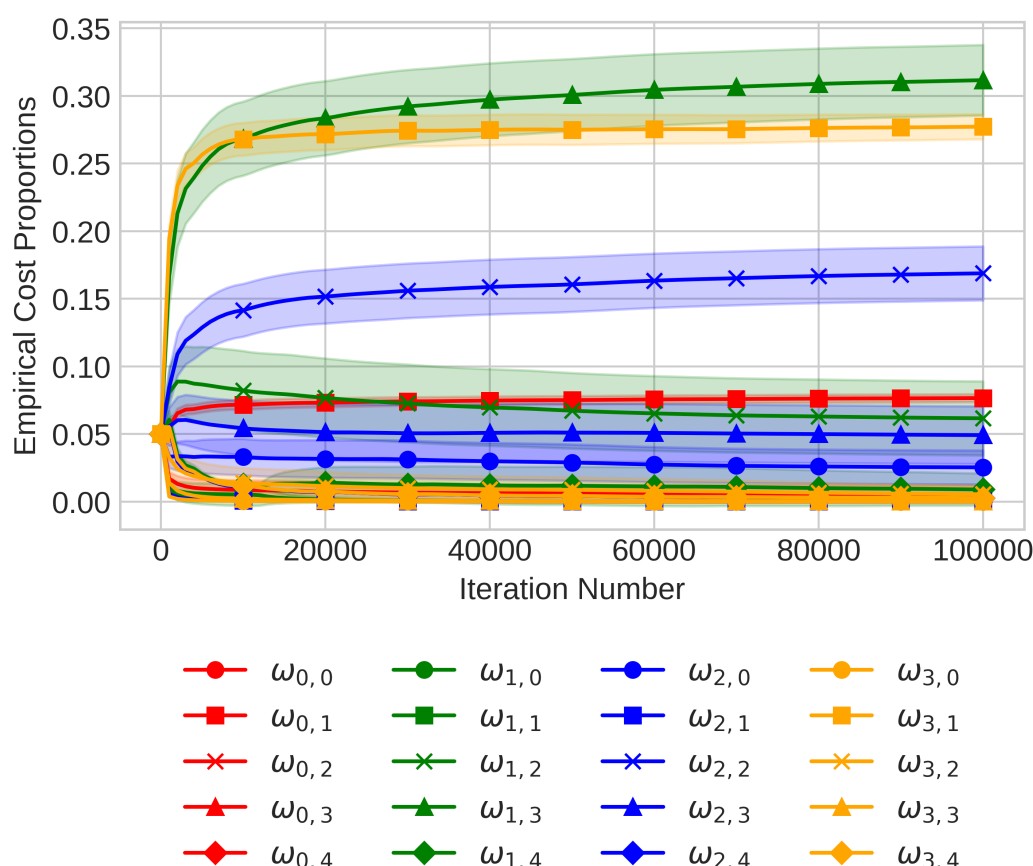

**Figure 9:** Empirical cost proportions of MF-GRAD-CONST for 100000 iterations on the $4 \times 5$ bandit model of Section 5. Results are average over 100 runs and shaded area report 95% confidence intervals. Empirical cost proportions of each arm are plotted with the same color. Cost proportions at fidelity 1, 2, 3, 4 and 5 are visualized with circle, squared, cross, triangle, and diamond respectively.

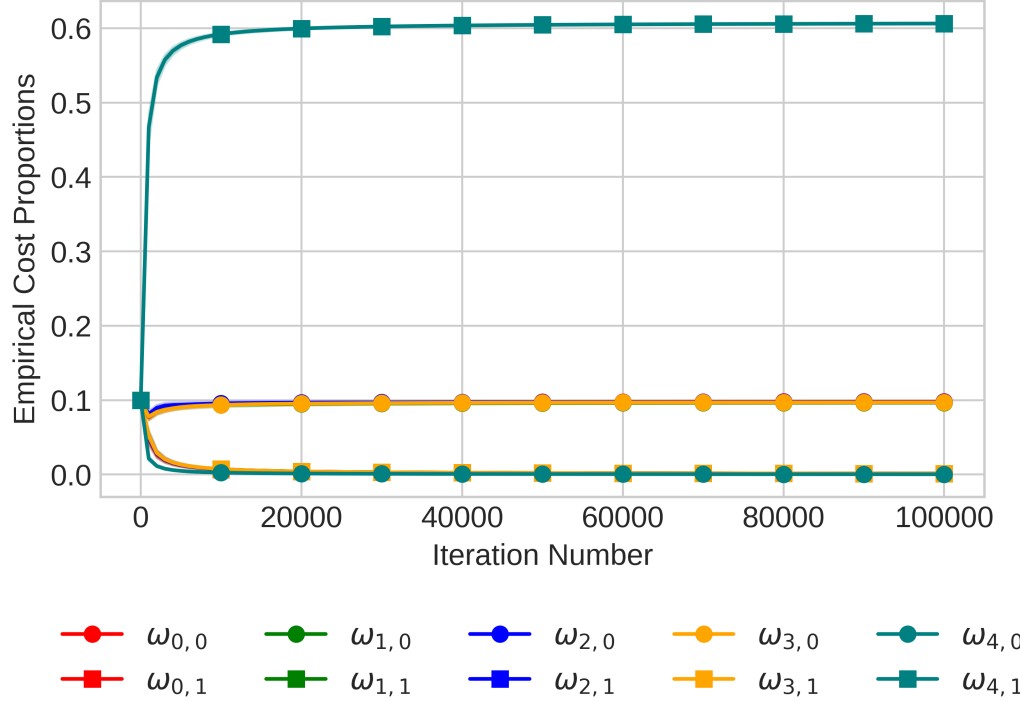

**Figure 10:** Empirical cost proportions of MF-GRAD-CONST for 100000 iterations on the $5 \times 2$ bandit model of Section 5. Results are average over 100 runs and shaded area report $95\%$ confidence intervals. Empirical cost proportions of each arm are plotted with the same color. Cost proportions at fidelity 1 and 2 are visualized with circle and squared respectively.

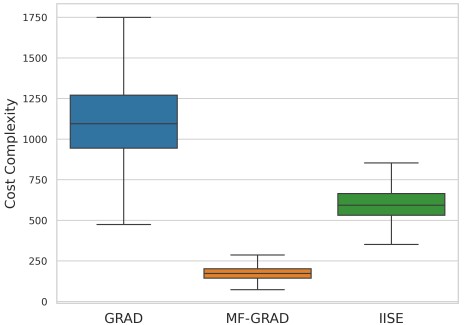

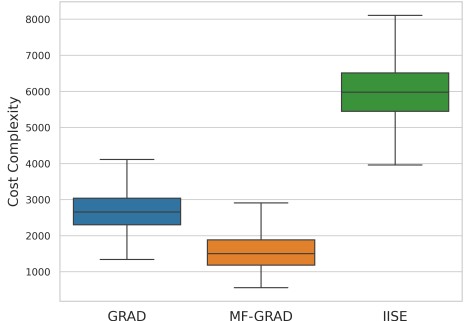

**Figure 11:** Empirical cost complexity for 1000 runs times with $\delta = 10^{-10}$ on the $4 \times 5$ multi-fidelity bandit of Table 2.

**Figure 12:** Empirical cost complexity for 1000 runs times with $\delta = 10^{-10}$ on the $5 \times 2$ multi-fidelity bandit of Table 3.

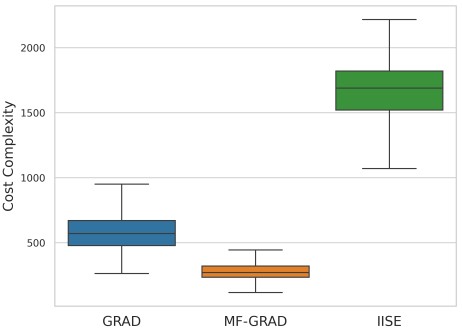
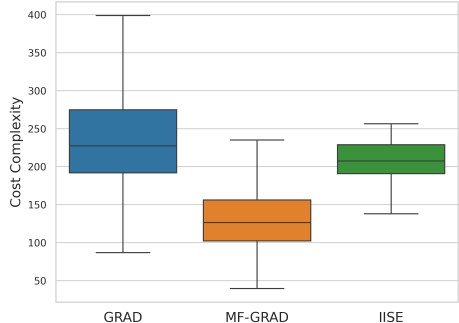

**Figure 13:** Empirical cost complexity for 1000 runs times with $\delta = 10^{-10}$ on the multi-fidelity bandit of Table 4.

**Figure 14:** Empirical cost complexity for 1000 runs times with $\delta = 10^{-10}$ on the multi-fidelity bandit of Table 5.

### D.6 On LUCB-ExploreA and LUCB-ExploreB

In this section, we present in detail the main issue behind the algorithms presented in [31], i.e., LUCBExploreA and LUCBExploreB, that is the fact that these algorithm might fail at stopping in some specific multi-fidelity bandit models. First, we provide numerical evidence of this phenomena by running both methods in a specific instance (Section D.6.1). Then, in Section D.6.2, we point out an error in the analysis of [31] that highlights how both algorithms fails at stopping when considering instances such as the one that has been considered in Section D.6.2.

#### D.6.1 Experimental issues

When experimenting with the algorithms proposed in [31], namely LUCBExploreA and LUCB-ExploreB, we have faced stopping issues. Specifically, both algorithms were not terminating in any reasonable number of steps on some specific instances. We now report an illustrative example of such scenarios. Consider the following Gaussian multi-fidelity bandit model: $\mu_1 = [0.64, 0.6]$, $\mu_2 = [0.46, 0.5]$, $\lambda = [0.1, 5]$, $\xi = [0.1, 0]$ and $\sigma^2 = 1$. In this scenario, the well-known LUCB algorithm [12] which only uses samples at fidelity $M$, stops soon (iteration $\approx$ 100k) paying a total cost of roughly 500k. When running LUCBExploreA and LUCBExploreB, instead, we faced termination issues. We let both algorithms run for a maximum number of $10^8$ samples (reaching a total cost which is approximately $10^7$), and the stopping criterion was never met for LUCBExploreA, while 70% of LUCBExploreB runs did not stop. LUCBExploreB explores more fidelities at the beginning, and that initial exploration can be enough to trigger the stopping test on some runs, but many continue until we artificially stop the experiment. Figure 15 reports the results of this experiment.

As a final remark, we notice that both LUCBExploreA and LUCBExploreB require additional knowledge in order to run, that is an upper bound on $\mu_{1,M}$ and a lower bound on $\mu_{2,M}$ (assuming arms to being ordered according to $\mu_{1,M} > \mu_{2,M} \geq \cdots \geq \mu_{K,M}$). The result presented in this section have been presented running their algorithms in the most favorable scenario, that is the situation in which the agent has perfect knowledge on the values $\mu_{1,M}$ and $\mu_{2,M}$.

#### D.6.2 Theoretical issues

The general idea of the LUCBExplore algorithms of [31] is to identify for each arm the "optimal fidelity" and pull the arm at that fidelity. In Appendix B.5, we described how that "optimal fidelity" can differ from the fidelity selected by our lower bound. Since our lower bound can be matched by an algorithm and thus describes the actual cost complexity of the problem, it betters represent the notion of optimal fidelity. We will thus call the fidelity used by the LUCBExplore algorithms *target fidelity* instead. The two variants ExploreA and ExploreB differ in the mechanism used to look for the target fidelity.

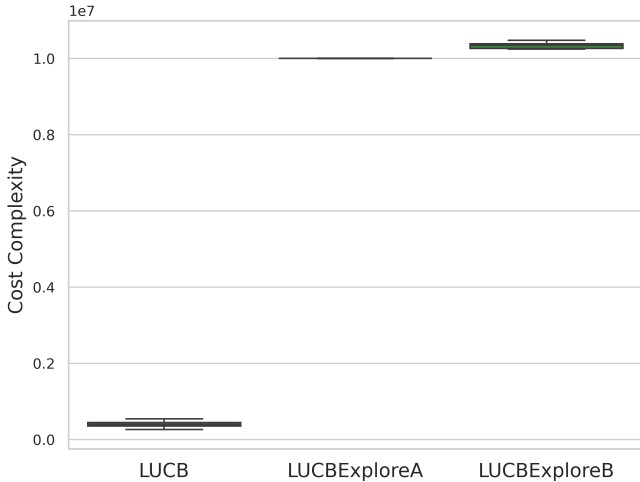

**Figure 15:** Visualization of the non-stopping behavior of LUCBExploreA and LUCBExploreB.

We first show that even if their algorithm used an oracle for the fidelity exploration mechanism that returns the target fidelity for all arms, it would still not be able to stop on some examples. We then highlight an issue with the proof of [31].

**Failure to stop with an oracle**  Consider the bandit instance from Appendix B.5. Recall that this is a $2 \times 2$ example of multi-fidelity BAI problem with $\xi_1 = 0.1$, $\xi_2 = 0.0$, $\mu_{1,M} = 0.6$, $\mu_{1,m} = 0.65$, $\mu_{2,M} = 0.5$, $\mu_{2,m} = 0.45$ (we write $M = 2$ and $m = 1$). All distributions are Gaussian with variance 1. We choose $\lambda_M > 4\lambda_m$, which means that the target fidelity for that problem are $m_1^* = 1$ and $m_2^* = 1$ (see details in Appendix B.5). LUCBExplore with an oracle that always selects that fidelity is the following algorithm:

- Initialization: $\hat{\mu}_{k,m}(t) = 0$, $N_{k,m}(t) = 0$, $UCB_k(t) = 1$, $LCB_k(t) = 0$ for all arms $k$ and fidelity $m$. $\ell_t = 1$, $u_t = 2$.
- While $LCB_{\ell_t}(t) \leq UCB_{u_t}(t)$
    - $\ell_t = \arg\max_k UCB_k(t)$, $u_t = \arg\max_{k \in [k] \setminus \{\ell_t\}} UCB_k(t)$
    - Pull arms $\ell_t$ and $u_t$ at their target fidelity.
- Output $\ell_t$

The indices are

$$LCB_k(t) = \max_m \left(\hat{\mu}_{k,m}(t) - \xi_m - \beta(N_{k,m}(t), t, \delta)\right)$$
$$UCB_k(t) = \min_m \left(\hat{\mu}_{k,m}(t) + \xi_m + \beta(N_{k,m}(t), t, \delta)\right)$$

where $\beta(n, t, \delta) = \sqrt{\log(Lt^4/\delta)/n}$ for some constant $L > 0$.

In the two-arms example here, the algorithm simplifies greatly: it always pulls both arms alternatively, always at fidelity $m = 1$. It stops when the LCB of one arm surpasses the LCB of the other.

We show that it can't stop and return the best arm 1, unless a confidence interval is not valid, which happens with small probability. If $\hat{\mu}_{1,1}(t) \leq \mu_{1,1} + \beta(t/2, t, \delta)$,

$$LCB_1(t) = \max\{0, \hat{\mu}_{1,1}(t) - \xi_1 - \beta(t/2, t, \delta)\}$$
$$\leq \mu_{1,1} - \xi_2$$
$$= 0.55 .$$

On the other hand, if $\min\{1, \hat{\mu}_{2,1}(t) \geq \mu_{2,1} - \beta(t/2, t, \delta)$,

$$\begin{aligned} UCB_2(t) &= \min\{1, \hat{\mu}_{2,1}(t) + \xi_1 + \beta(t/2, t, \delta)\} \\ &\geq \mu_{2,1} + \xi_1 \\ &= 0.55 \ . \end{aligned}$$

We get that we always have $LCB_1(t) \leq UCB_2(t)$, unless one of the two concentration inequalities on the empirical means are not true. The confidence width $\beta$ is designed to make those inequalities true for all $t \in \mathbb{N}$ with probability close to 1. We can similarly get that $LCB_2(t) \leq UCB_1(t)$ (which is expected since 2 is a worse arm) unless some concentration inequality is false.

We obtain that this algorithm with an oracle selection for the target fidelity cannot stop fast: the only way it can stop is if unlikely deviations occur.

**Issue with the proof**   There is an issue with the proof of the cost complexity upper bound of [31]. The issue is in the first 3 steps of their appendix E.2, pages 17 and 18. They identify a threshold $c$ (with value $0.55$ in our example of the last paragraph) and prove the following.

- Step 1: if the algorithm does not terminate and confidence intervals hold, then either both $LCB_{\ell_t}(t) \leq c$ and $UCB_{\ell_t}(t) \geq c$ or both $LCB_{u_t}(t) \leq c$ and $UCB_{u_t}(t) \geq c$.
- Step 2: confidence intervals are likely to hold.
- Step 3: if a sub-optimal arm $k$ satisfies $LCB_k(t) \leq c$ and $UCB_k(t) \geq c$, then its target fidelity cannot be pulled much.

They conclude that for all arms, the number of pulls at the target fidelity is upper bounded, with large probability.

Let's see the issue with that proof, on the same example as in the last paragraph.

In the example above with the oracle choice for the target fidelity, we saw that if confidence intervals hold and $\ell_t = 1$ (which is the most likely), then $LCB_{\ell_t}(t) \leq c$ and $UCB_{\ell_t}(t) \geq c$. That is, step 1 gives a condition on arm 1 only (and nothing on arm 2). But then we get nothing from step 3, since arm 1 is not a sub-optimal arm.

We only get an upper bound on the number of pulls for sub-optimal arms if we can say that they satisfy $LCB_k(t) \leq c$ and $UCB_k(t) \geq c$ at some point, but it might not be the case. Indeed, when the algorithm does not terminate, steps 1 and 2 together give that with large probability either both $LCB_{\ell_t}(t) \leq c$ and $UCB_{\ell_t}(t) \geq c$ or both $LCB_{u_t}(t) \leq c$ and $UCB_{u_t}(t) \geq c$. It is possible that we always have this property for $\ell_t = 1$ (the optimal arm), and that we can never apply step 3.

