# OpenReview forum: "Optimal Multi-Fidelity Best-Arm Identification"
_NeurIPS.cc/2024/Conference — NeurIPS 2024 poster_

### Official Review · Reviewer_7qou · 2024-07-07

**Soundness:** 3
**Presentation:** 1
**Contribution:** 2
**Rating:** 4
**Confidence:** 3

**Summary:**

This paper introduces qPOTS (Pareto Optimal Thompson Sampling), a novel approach for efficient batch multiobjective Bayesian optimization in the context of best-arm identification (BAI) with multi-fidelity bandits. The authors present a tight lower bound on the cost complexity of multi-fidelity BAI and propose an algorithm that asymptotically matches this bound. The method combines ideas from Thompson sampling, Gaussian process surrogates, and evolutionary algorithms to address challenges in multiobjective optimization with expensive oracles. The paper provides theoretical analysis, including convergence guarantees, and demonstrates empirical performance on synthetic and real-world problems.

**Strengths:**

Theoretical contributions: The paper presents a tight lower bound on the cost complexity of multi-fidelity BAI, which improves upon previous bounds in the literature. This provides valuable insights into the fundamental limits of such problems.

Algorithm development: The proposed qPOTS algorithm is designed to asymptotically match the lower bound, potentially offering optimal performance in the high-confidence regime.

**Weaknesses:**

Limited experimental evaluation: The experiments seem relatively limited in scope, focusing primarily on synthetic problems and a few real-world examples. More extensive testing on diverse real-world applications, such as neural architecture search or classic multi-fidelity Bayesian optimization problems, would strengthen the paper's practical relevance.

Connection to real-world problems: While the theoretical results are impressive, it's not immediately clear how they translate to understanding or solving real-world problems. More discussion on practical implications would be beneficial.

Assumption realism: The paper relies on some assumptions that may not always hold in practice. For example, Assumption 5 in Theorem 4.1 (asymptotic convergence) may not be realistic in all scenarios. More discussion on the implications of these assumptions would be helpful.

Narrow problem focus: The best-arm identification problem, while important, may be seen as somewhat limited compared to more general multi-fidelity Bayesian optimization problems. The authors could elaborate more on how their work relates to or extends to broader MFBO settings.

Gradient computation clarity: The explanation of gradient computation with respect to ω and μ could be clearer, especially given that these values are often discrete in bandit problems. More detailed explanations and precise notation would improve understanding.

**Questions:**

How does this work relate to broader multi-fidelity Bayesian optimization problems? Can the authors elaborate on potential extensions or applications beyond best-arm identification?

Can the authors provide more intuition or examples of how the theoretical results help in understanding real-world multi-fidelity optimization problems?

Theorem 3.1 states that the lower bound holds for any multi-fidelity bandit. Does this truly hold regardless of the specific multi-fidelity model or acquisition function used? If so, could the authors elaborate on why this is the case?

How realistic is Assumption (5) in Theorem 4.1? Can the authors provide examples of scenarios where this assumption would or would not hold, and discuss the implications?

The gradient computation with respect to ω and μ is not entirely clear, especially given that these are often discrete values in bandit problems. Could the authors provide a more detailed explanation of how these gradients are defined and computed, ensuring clarity and precision in the notation?

Have the authors considered applying their method to more complex real-world applications, such as neural architecture search or other MFBO problems that can be framed as BAI? If not, what challenges might arise in such applications?

**Limitations:**

The paper could benefit from a more thorough discussion of the scenarios where the assumptions might not hold and the potential implications for the algorithm's performance.

The authors should provide more context on the limitations of the best-arm identification framework compared to more general multi-fidelity Bayesian optimization problems.

A discussion on the potential challenges in applying the method to more complex, real-world problems would be valuable.
The paper would benefit from a more comprehensive discussion of the potential negative societal impacts of the work, even if they are indirect.

---

> ### Author Rebuttal · Authors · 2024-08-06
>
> > ###  "Narrow problem focus" and comparison of MF-BAI with MFBO
>
> First, we note that the **multi-fidelity BAI is an established problem within the literature** (i.e., "Multi-fidelity best-arm identification", Poiani et al., NeurIPS 2022, and "Multi-fidelity multi-armed bandits revisited", Wang et al., NeurIPS 2024).
>
> Furthermore, we note that our work on BAI **differs** from MFBO studies in at least the two following dimensions (similar comments have already been made in e.g., "Multi-fidelity Best-Arm Identification", Poiani et al., 2022):
> 1. First, we only use a minimal assumption on the arm space, that is $|\mu_{i,m} - \mu_{i,M}| \le \xi_m$ for all arms $i \in [K]$ and fidelity $m \in [M]$. On the other hand, works on MFBO (e.g., "Gaussian Process Bandit Optimisation with Multi-fidelity Evaluations", Kandasamy et al., NeurIPS 2016, "Multi-fidelity bayesian optimisation with continuous approximations", Kandasamy et al., ICML 2017, "Multi-fidelity Gaussian Process Bandit Optimisation", Kandasamy et al., JAIR 2018) typically impose stronger structural such as Gaussian Process modeling.
> 2. Secondly, these lines of work usually analyze the simple/cumulative regret. Our paper, instead, aims at minimizing the cost complexity. As one can verify, this has a significant impact on the algorithm design and its theoretical analysis.
>
> For these reasons, we retain that our work is **different** (rather than less general) w.r.t. MFBO approaches. We will better clarify these differences in a revised version of the manuscript.
>
> > ### "Can the authors provide more intuition or examples of how the theoretical results help in understanding real-world multi-fidelity optimization problems?"
>
> The theoretical result on the lower bound leads to the first asymptotically optimal algorithm for the MF-BAI problem. As discussed both in the lower bound section and in the algorithmic one, these novel results significantly improve the state-of-the-art in the MF-BAI tasks. Indeed, previous lower bounds were, in general, not tight up to a $\lambda_M / \lambda_1$ factor, which can be arbitrarily large, and existing algorithms were based on a sub-optimal concept of "optimal fidelity". As verified in the experimental section, these lead to a superior identification performance of our method compared to existing algorithms.
>
> Finally, our study proves that, to identify the optimal arm, there almost always exists an optimal arm-dependent fidelity that the learning system should query.
>
> > ### "Does Theorem 3.1 hold regardless of the specific multi-fidelity model or acquisition function used?"
>
> Theorem 3.1 holds for any algorithm that satisfies the $\delta$-correctness requirement on all the multi-fidelity bandits within the class $\mathcal{M}^*_{\textup{MF}}$. These are all the multi-fidelity problems that can be modeled as a BAI problem formulated in Section 2.
>
> We are unsure what the reviewer means by "hold regardless of the specific multi-fidelity model." Furthermore, we are not sure what the reviewer means by "any acquisition function used" as there is no concept of acquisition function within our work.  Can the reviewer clarify these points?
>
> > ### "How realistic is Assumption (5) in Theorem 4.1?"
>
> We are unsure what the reviewer means by Assumption (5) in Theorem 4.1 as **there is no assumption** in the statement or in the proof of Theorem 4.1. Can the reviewer clarify this point?
>
> > ### "The gradient computation with respect to ω and μ is not entirely clear, especially given that these are often discrete values in bandit problems"
>
> First, we recall that the sub-gradient is computed only w.r.t. to ${\omega}$, as $\omega$ represents the cost-proportions that the algorithm should play in expectation to identify the optimal arm using the minimum cost.
>
> Secondly, we notice that the values of both $\omega$ and $\mu$ **do not assume discrete values**. Indeed, $\omega$ belongs to the simplex, while $\mu$ belongs to the set of possible means (i.e., $\Theta^{KM}$), which is not a finite set. This is standard in the BAI literature (see, e.g., "Optimal Best Arm Identification with Fixed Confidence", Garivier et al., 2016), where $\omega$ and $\mu$ do not belong to discrete sets as well.
>
> Finally, the sub-gradient is computed exactly as stated in Theorem 4.3, and, more precisely, in Equations (8) and (9), where $\eta^*_{i,a}$ and $\psi^*_j$ are the quantities introduced in Lemma 4.2 and Line 253. In other words, for fixed $\mu$ and $\omega$, one first computes the pair of arms $(i,a)$ that attain the maximum of Equation (2), and then tests whether $\psi^*_i > \psi^*_a$ to decide which expression to use among Equations (8) and (9).
>
> The discussion presented in Lines 272-291 then provides an efficient algorithm to compute $\eta^*_{i,a}$. Lemma C.4 in the appendix provides an analogous algorithm for the computation of $\psi^*_i$.
>
> > ### "Have the authors considered applying their method to more complex real-world applications, such as neural architecture search or other MFBO problems that can be framed as BAI? If not, what challenges might arise in such applications?"
>
> As done in previous works in MF-BAI, we have considered simulated domains in our experiments. When dealing with real-world problems, challenges arise when directly applying the multi-fidelity formalism considered in MF-BAI. For instance, the arm space might be continuous, such as in the case of hyper-parameter optimization.
> In this scenario, practical methods (e.g., "Multi-fidelity Bayesian optimization via deep neural networks", Li et al., NeurIPS 2020), can overcome these challenges at the price of sacrificing theoretical guarantees. An interesting future research direction to bridge this gap would be to study the MF-BAI setting under the presence of linear function approximators; e.g., by drawing inspiration from linear BAI ("Optimal best-arm identification in linear bandits", Jedra et al., NeurIPS 2020).

---

> ### Author Response · Authors · 2024-08-06
>
> Dear reviewer,
>
> first thanks again for the time taken to review our paper. We are writing a general comment here to outline some doubts that we have on this review, as there are some comments that we believe are not aligned with the content of our paper, and consequently, we are unsure on how to answer some questions. Specifically:
>
> - Within the summary, the reviewer mentions that "This paper introduces qPOTS (Pareto Optimal Thompson Sampling), a novel approach for efficient batch multiobjective Bayesian optimization in the context of best-arm identification (BAI) with multi-fidelity bandits" and "The method combines ideas from Thompson sampling, Gaussian process surrogates, and evolutionary algorithms to address challenges in multiobjective optimization with expensive oracles." Nevertheless, this is not aligned with the content of our paper.
> - Similarly, within the strenghts there is another reference to the qPOTS algorithm, which is not something that we consider in our paper.
> - Then, in the weaknesses, the reviewer mentions Assumption (5) in Theorem 4.1. However, there is no assumption neither in the statement nor the proof of the theorem
> - Finally, in the questions, the reviewer also asks if our results holds for any acquisition function used. However, there is no concept of acquisition function within our work.
>
> In the rebuttal, we answer the reviewer's comment to the best of our understanding. Naturally, we remain available for further clarifications, but we kindly ask the reviewer to clarify the points above.
>
> Best regards,
>
> The authors

---

### Official Review · Reviewer_97EK · 2024-07-13

**Soundness:** 3
**Presentation:** 3
**Contribution:** 3
**Rating:** 7
**Confidence:** 3

**Summary:**

This paper considers the fixed confidence BAI problem in a multi-fidelity setting, where each arm can be sampled at lower or higher fidelity levels, with a corresponding lower or higher cost. The objective is to declare the best arm, which is the arm with the highest mean at the highest fidelity level.

The authors derive an instance dependent cost lower bound, in terms of optimal sampling frequencies at each fidelity level for each arm. They then propose a sub gradient based algorithm with forced exploration that is asymptotically optimal in terms of cost.

**Strengths:**

This paper makes non-trivial technical contributions to the state of the art in multi fidelity BAI. The measure-zero result in Theorem 3.2 seems novel and interesting. The "achievability" part is also novel, in the sense that it is not an extension of traditional track and stop. It uses a sub gradient ascent algorithm (which has been used in bandits before), but the gradient computation section in 4.2 also appears potentially useful in related problem settings.

Appendix B.2 shows that the lower bound in [26] can be significantly worse (by an arbitrarily large multiplicative factor) than the proposed lower bound.

In appendix D.6, the authors also identify a problem of "non-stopping" from [33], both numerically and analytically.

**Weaknesses:**

I do not see any issue that can be flagged as a significant weakness.

**Questions:**

I don't see any shaded area denoting confidence intervals in Figure 3.

Line 78: notation.

**Limitations:**

No concerns on social impacts

---

> ### Author Rebuttal · Authors · 2024-08-06
>
> > ### I don't see any shaded area denoting confidence intervals in Figure 3.
>
> We thank the Reviewer for raising this point. There are actually shaded areas on the plot, but the confidence intervals are really small. This is also evident from footnote 8, where we measure the distance of ${\omega}(T)$ from $\omega^*$ at iteration $T=10^5$. In this case, the 95% confidence intervals are $0.0006$.
>
> The appendix presents experiments with more evident confidence intervals  (see, e.g., Figure 4).
>
> > ### Line 78: notation.
>
> We thank the Reviewer for pointing out the typo. We have fixed it in a revised version.

---

> > ### Comment · Reviewer_97EK · 2024-08-13
> > **Thank you for your response**
> >
> > I appreciate your response. I keep my score.

---

### Official Review · Reviewer_bSBL · 2024-07-14

**Soundness:** 3
**Presentation:** 3
**Contribution:** 3
**Rating:** 7
**Confidence:** 4

**Summary:**

The paper studies the problem of multi-fidelity best arm identification in the fixed confidence setting. The main contribution is an instance-dependent lower bound on the cost complexity. The authors demonstrate the bound's tightness by providing an algorithm with a matching upper bound in asymptotics, as well as an instance where the lower bound improves on the previous lower bound by a tighter constant.

**Strengths:**

The paper is well-presented in terms of its theorem statements and comparison with prior works. The theoretical result, specifically the lower bound, shows a clear improvement over previous work.

**Weaknesses:**

I think there is room to improve the presentation of the algorithm. Just by staring at the algorithm box, it took some time to understand the relationships between $\tilde{\omega}$, $\bar{\omega} $, $\tilde{\pi}$, $\pi'$, and have to look for their definitions here and there. It would be great if these could be made clear within the algorithm box itself.

**Questions:**

Do the authors have any conjectures on the fixed-budget setting for multi-fidelity best arm identification? I'm not entirely sure this is a valid problem, but I wonder if it could be approached similarly to the normal track-and-stop method. By solving the optimization problem and sampling according to it without a stopping rule, it naturally turns into a fixed-budget algorithm. Can MF-GRAD be used in a similar manner?

**Limitations:**

No major limitation to me.

---

> ### Author Rebuttal · Authors · 2024-08-06
>
> > ### Writing suggestion
>
> We thank the Reviewer for suggesting ways to improve the presentation of our algorithm. We have incorporated the symbols also within the algorithm box in a revised version of the manuscript.
>
>
> > ### "Do the authors have any conjectures on the fixed-budget setting for multi-fidelity best arm identification? I'm not entirely sure this is a valid problem, but I wonder if it could be approached similarly to the normal track-and-stop method. By solving the optimization problem and sampling according to it without a stopping rule, it naturally turns into a fixed-budget algorithm. Can MF-GRAD be used in a similar manner?"
>
>
> We thank the Reviewer for rising this point. We do believe that the fixed-budget MF-BAI is actually a valid problem that should deserve the attention of the community. In this case, the **budget should be expressed as a total cost available** to the learning system, and the agent might exploit the different fidelity to lower (an upper bound on) the probability of error of not recommending the optimal arm.
>
> Although this future research direction is of interest to the community, we currently conjecture that **further and ad-hoc algorithms are probably needed** to face the fixed-budget problem. Indeed, to the best of the author's knowledge, fixed-budget algorithms usually follow different design principles compared to asymptotically optimal fixed-confidence ones. One of the main reasons behind this conjecture is that we expect the lower bound for the fixed-budget setting to differ from the one for fixed confidence.
>
> Finally, we agree that one could in principle apply MF-GRAD for fixed-budget problems as suggested by the reviewer (i.e., by removing the stopping rule and terminating when the budget is over). However, we are unsure on which kind of theoretical guarantees this approach would enjoy.

---

### Author Rebuttal · Authors · 2024-08-06

We thank the Reviewers for their efforts in reviewing our paper. We are happy that the reviewers have recognized our paper as a "clear improvement over previous work" (Reviewer bSBL) and that they appreciated the "non-trivial technical contributions to the state of the art in multi fidelity BAI" (Reviewer 97EK). In the following, we aim to address the remaining questions.

---

### Decision · Program_Chairs · 2024-09-25

**Decision:**

Accept (poster)

**Comment:**

This paper is broadly acknowledged to make significant contributions to algorithm design for the practical problem of multi-fidelity best arm identification problems in bandits. It not only improves upon previous fundamental limits on costs, but also develops a gradient-based algorithm which, given current computational paradigms in ML, is appealing from the implementation point of view, while also being situated on firm foundations.

In view of the strengths of the paper, it is recommended for acceptance.